# From Knowledge to Inference: Formalizing Specialized Public Health Reasoning on GlobalHealthAtlas

**Zhaokun Yan** [* 1]  **Shan Xu** [* 1 2]  **Wuzheng Dong** [* 1]  **Zhaohan Liu** [* 1]  **Lijie Feng** [3]  **Chengxiao Dai** [4]  **Tianqi Chen** [1]
**Binfan Liu** [5]  **Yunpu Ma** [6]  **Wenting Wei** [7]  **Yingting Li** [1]  **Yi Zhang** [1]  **Tongning Wu** [1]

## Abstract

Public health reasoning requires population level inference grounded in scientific evidence, expert consensus, and safety constraints. However, it remains underexplored as a structured machine learning problem with limited supervised signals and benchmarks. We introduce **GlobalHealthAtlas**, a large scale multilingual dataset of 280,210 instances spanning 15 public health domains and 17 languages. We further propose a large language model (LLM) assisted construction and quality control pipeline with retrieval, deduplication, evidence grounding checks, and label validation to improve consistency at scale. Finally, we present a domain aligned evaluator distilled from high confidence judgments of diverse LLMs to assess outputs along six dimensions: Accuracy, Reasoning, Completeness, Consensus Alignment, Terminology Norms, and Insightfulness. Together, these contributions enable reproducible training and evaluation of LLMs for safety critical public health reasoning beyond conventional QA benchmarks. We publicly release project codebase, evaluator, and model at: `https://github.com /Jan8217/GlobalHealthAtlas`, `https: //huggingface.co/aerovane0/Globa lHealthAtlas_Public_Evaluator` and `https://huggingface.co/aerovane0 /GlobalHealthAtlas_Public_Model`

## 1. Introduction

LLMs have made rapid progress in reasoning and general knowledge, achieving strong performance across a wide range of tasks, including several expert facing applications that even surpass human experts(Chowdhery et al., 2023). These advances have also led to notable progress in medical and clinical applications, including diagnosis support, report generation, and medical question answering(Kim et al., 2024c). However, public health remains substantially underrepresented in the current LLM development and evaluation, particularly with respect to public health reasoning capabilities(Espinosa & Salathé, 2024; Zhang et al., 2024).

The limitations primarily arise from two factors: (I) the lack of high quality data, which constrains effective model inference in public health, the meantime, narrow and imbalanced distributions, covering limited range of health topics(Du et al., 2025); (II) Absence of domain aligned evaluation frameworks, as existing metrics adapted from other fields fail to capture core characteristics of public health reasoning, such as population level inference, policy context, and intervention effectiveness. To substantiate these observations, we systematically survey representative medical and public health datasets, and summarize their key properties in Table 1, highlighting the limited coverage, scale, and evaluation support of existing resources(Naliyatthaliyazchayil et al., 2025).

To justify GlobalHealthAtlas, we conducted a domain performance gap analysis comparing clinically-aligned models on medical versus public health tasks. We evaluated Medical_Model(8B), AntAngelMed, and BAICHUAN-M3. Although Medical_Model performs well on clinical tests, its performance drops significantly on public health tasks. Even large scale models, like AntAngelMed and BAICHUAN-M3, don't saturate GlobalHealthAtlas or Instruction-public-health-dataset, as shown in Table 2. These results show that existing clinical benchmarks fail to capture population-level inference, policy context, and intervention effectiveness, highlighting the need for a dedicated public health dataset(Dou et al., 2026).

**GlobalHealthAtlas** is introduced as a large scale resource

---
[*]Equal contribution  [1]China Academy of Information and Communications Technology, Beijing, China [2]Shanghai Institute of Infectious Disease and Biosecurity, School of Public Health, Fudan University, Shanghai, China [3]CRRC Industrial Academy Co., Ltd., Beijing, China [4]The University of Sydney, Sydney, Australia [5]Beijing University of Posts and Telecommunications, Beijing, China [6]Ludwig Maximilian University of Munich [7]Shanghai Artificial Intelligence Laboratory. Correspondence to: Tongning Wu <wutongning@caict.ac.cn>.

*Proceedings of the 43rd International Conference on Machine Learning*, Seoul, South Korea. PMLR 306, 2026. Copyright 2026 by the author(s).

*Table 1.* Comparison of representative medical and public health reasoning datasets. **Diff.** indicates difficulty levels; **Scale** denotes dataset size; **Split** indicates the availability of training and test splits; **Eval.** refers to the evaluation paradigm; **Leak** indicates whether leakage detection is performed. **GlobalHealthAtlas** offers broader coverage and domain-specific evaluation framework for public health reasoning

| | Diff. | Scale | Split | Domain | Language | Eval. | Leak |
|---|---|---|---|---|---|---|---|
| MedQA [(Jin et al., 2021b)] | X | 61097 | ✓ | 8 | EN/ZH-S/ZH-T | Rule | X |
| shibing624-medical [(shibing624, 2023)] | X | 2.43M | ✓ | 4 | EN/ZH-S | X | X |
| medical-o1-reasoning-SFT [(Chen et al., 2024)] | X | 40000 | X | 6 | EN/ZH-S | Rule | X |
| HealthSearchQA [(Singhal et al., 2023)] | X | 3375 | ✓ | Open domain | EN | Human+MultiMedQA | X |
| ApolloCorpora [(Wang et al., 2024a)] | X | 2.5B tokens | ✓ | Open domain | 6 | Human+XMedBench | ✓ |
| MMedC [(Qiu et al., 2024)] | X | 25.5B tokens | ✓ | Open domain | 6 | Rule+human | X |
| AfriMed-QA [(Nimo et al., 2025)] | X | 15275 | X | 32 | EN | Rule+Human | X |
| publichealth-qa [(Lu, 2024)] | X | 896 | X | 1 | 8 | X | X |
| Instruction-public-health-dataset [(sambanankhu, 2024)] | X | 748 | X | 5 | EN | X | X |
| **GlobalHealthAtlas(ours)** | **1,2,3** | **280210** | ✓ | **15** | **17** | Spec. Model | ✓ |

*Table 2.* Comparison of representative models on clinical and public health reasoning tasks. For fair cross-domain comparison, Medical-Model(8B)was fine-tuned on mainstream clinical datasets (MedQA, medical_o1_sft_CN+EN, and shibing624-medical_train) with a 5:1 train test split, reserving 1/6 as an independent clinical test set (Medical_test)

| | GlobalHealthAtlas | GlobalHealthAtlas SC | Instruction-public-health-dataset | Medical_test_modified | Medical_test_modified SC | model size | domain |
|---|---|---|---|---|---|---|---|
| Public-Model | 6.532 | 89.02% | 6.096 | 7.208 | 72.26% | 8B | Public-health |
| AntAngelMed | 6.449 | 86.13% | 5.897 | 8.082 | 77.51% | Moe-100B-6.1B | Integrated Public Health and Medicine |
| BAICHUAN-M3 | 5.970 | 78.34% | 5.994 | 7.917 | 77.18% | 235B | Medicine |
| Medical_Model | 5.250 | 61.14% | 4.697 | 7.184 | 78.20% | 8B | Medicine |

containing 280,210 instances; it spans 15 public health domains and 17 languages while covering various task formats and three distinct difficulty levels.

To systematically assess GlobalHealthAtlas and the public health reasoning of LLMs, we introduce domain specific framework tailored to epidemiological constraints. Our approach evaluates outputs across six dimensions: **Accuracy, Reasoning, Completeness, Consensus Alignment, Terminology Norms, and Insightfulness**, using unified scoring guidelines. We operationalize the process via a fine tuned model trained on domain specific standards, ensuring consistent and scalable assessment.

We undertake extensive benchmarking on **GlobalHealthAtlas** across diverse public health settings. To systematically examine model behavior, we perform a series of robustness, ablation, and transfer experiments on the dataset, together with analyses across different difficulty levels. Based on these results, we summarize our main contributions as follows.

- **Establishing GlobalHealthAtlas with a Generalizable AI4S Data Construction Paradigm.** We establish a high fidelity multilingual public health dataset grounded in authoritative sources, comprising hundreds of thousands of carefully curated question answer pairs spanning diverse domains, languages, difficulty levels, and question types. Beyond the dataset itself, we distill a general and extensible 'AI for Science' (AI4S) data construction paradigm—a reproducible, AI-driven scientific pipeline designed to overcome

domain-specific data scarcity through model-driven generation, controlled sampling, and multi-stage quality refinement. Detailed in Section 3.2 and Appendix B

- **Constructing Evaluator with Six Orthogonal Dimensions for Public Health Reasoning Assessment.** We learn a dedicated evaluator through fine tuning qwen series models to support principled and reliable instruction based assessment in public health and general medical settings. The evaluation task is explicitly decomposed into six orthogonal dimensions, each addressing a fundamental risk factor in medical reasoning and decision making.

- **Tailoring LLMs Specifically for Public Health.** Building upon the proposed dataset and evaluator, we construct **Public-Model**. Extensive empirical experiments demonstrate consistent and robust improvements in reasoning capability and reliability, providing compelling evidence that the proposed approach meaningfully advances LLMs reasoning in public health.

## 2. Related Work

**General Reasoning and Multidisciplinary Benchmarks** The initial evaluation of LLMs focused on foundational logical capabilities in hard sciences. Benchmarks like GSM8K (Cobbe et al., 2021) and SciBench (Wang et al., 2024b) established standards for assessing multi step arithmetic and college level scientific problem solving. To capture a broader spectrum of multidisciplinary knowledge, suites such as C-Eval (Huang et al., 2023) and CMMLU (Li et al.,

2024) were developed to assess models within specific linguistic and cultural contexts. Recent studies further push the boundaries of reasoning by probing deeper logical presuppositions in Let's CONFER (Azin et al., 2025) and investigating models' persistent struggles with counterfactual scenarios (Yamin et al., 2025) and parametric knowledge generalization(Xu et al., 2025).

**Domain-Specific Medical and Public Health Datasets** As models advance into professional domains, evaluation has shifted from clinical knowledge retrieval to practical public health applications. Early benchmarks like MedQA (Jin et al., 2021a) and MedMCQA (Pal et al., 2022) utilize professional exams to test medical expertise, which are now being augmented by specialized systems like Med-PaLM (Singhal et al., 2025; Nori et al., 2023), AlpaCare (Zhang et al., 2025), and advanced prompting frameworks (Maharjan et al., 2024b;a). Parallel efforts in public health emphasize explainable fact checking (Kotonya & Toni, 2020; Zarharan et al., 2024), real time population monitoring (Joshi et al., 2025; Harris et al., 2025), and outreach understanding via PHORECAST (Qadri et al., 2025). Integration of knowledge graphs (Consoli et al., 2025), satellite based factors (Wang et al., 2025), and curated web data (Felipe et al., 2025; Mota et al., 2024) addresses the unique complexities of community level health communication (Qiu et al., 2025; Hoyt et al., 2025).

**Evolving Evaluation Paradigms and Data Integrity** Traditional objective metrics are increasingly being replaced by fine-grained evaluation paradigms. Frameworks like G-Eval (Liu et al., 2023), MT-Bench (Zheng et al., 2023), and AlignBench (Liu et al., 2024) leverage "LLM-as-a-judge" systems to capture human preference, supported by scalable feedback models such as Prometheus (Kim et al., 2024a;b) and UltraFeedback (Cui et al., 2024). However, the reliability of these results is often threatened by data contamination. To ensure genuine reasoning, advanced "Dataset Inference" techniques proposed by Maini et al. (Maini et al., 2024), Zhou et al. (Zhou et al., 2025), and Xiong et al. (Xiong et al., 2025) offer essential blackbox and label-only detection strategies to distinguish between memorized outputs and actual problem-solving capabilities(Zhang et al., 2026).

# 3. The PUBLIC HEALTH Dataset: GlobalHealthAtlas

## 3.1. Overview

**GlobalHealthAtlas** is a large scale public health dataset comprising 280,210 curated instances, each featuring complete Chain of Thought (CoT) reasoning. Synthesized from over 35,500 diverse sources from **WHO IRIS**, spanning 15 public health domains across 17 languages to facilitate systematic multilingual evaluation, details are shown in Fig. 1

and Fig. 3.

The corpus features a near balanced task distribution, with 49.42% in Question Answering format and 50.58% in Single Choice format. We categorize difficulty into three distinct levels: Level A (Academic / Professional, 26.26%) targeting graduate level epidemiology and policy; Level B (General Knowledge, 69.33%) focusing on disease prevention and health logic; and Level C (Popular Science, 4.41%) covering basic health literacy.

To ensure high data quality and scientific rigor, we established a **stringent construction pipeline** involving multi stage filtering and systematic quality control, Further details regarding the construction protocol prompt and data partitioning are provided in Figure 4 in Appendix B. Furthermore, we conducted proactive leakage detection across the entire corpus to guarantee the integrity of our evaluation benchmarks.

## 3.2. Evidence-Centric Data Engineering Pipeline

Heterogeneous public health PDFs are converted into structured Markdown through a deterministic parsing routine that preserves document hierarchies, including headings and list structures. The normalized Markdown is subsequently segmented into semantically coherent evidence chunks optimized for LLMs context constraints.

### 3.2.1. PROMPT DRIVEN SYNTHESIS HUB

The core of pipeline is prompt driven synthesis engine operating on two parallel tracks. Guided by expert designed instructions, the first track generates Question Answering or Single Choice pairs, while the second track distills complex reasoning trace (complexCOT). Each instance is annotated with multi dimensional metadata, including *domain*, *language*, *difficulty*, and *label*. The complete set of synthesis prompts is provided in Appendix B.

### 3.2.2. QUALITY GATE AND SCHEMA ENFORCEMENT

To ensure scientific rigor, LLM as a judge quality gate driven by specialized scoring prompts and schema-level constraints are employed. Each candidate is adjudicated along four dimensions: Question and Option Quality ($Q$), Answer Quality ($A$), Text Relevance ($R$), and Overall Consistency ($C$), while the weighting schema is bifurcated according to item type. For single-choice questions, the final quality score is defined as:

$$S_{\text{SC}} = 0.45Q + 0.25A + 0.20R + 0.10C \qquad (1)$$

where greater emphasis is placed on question formulation and structural validity. For question-answer items, the score

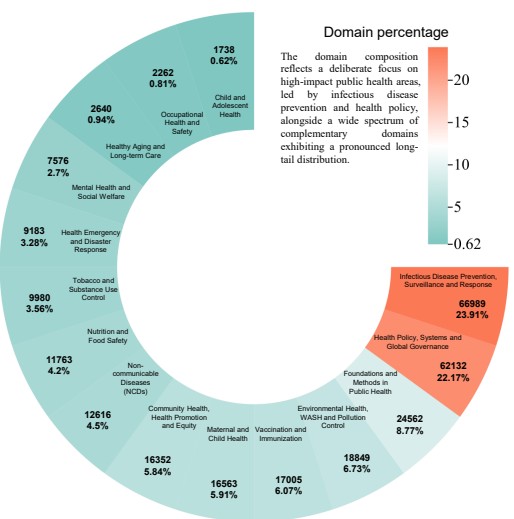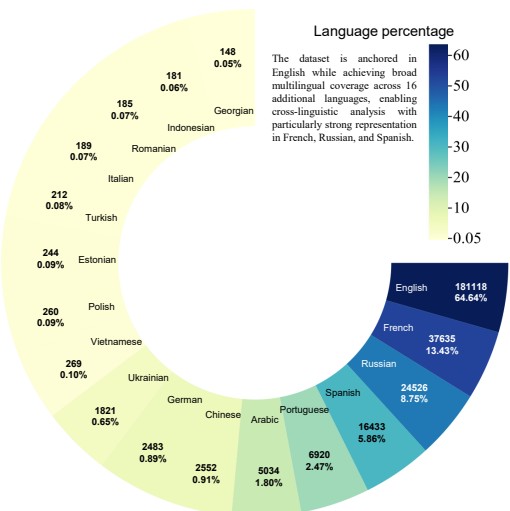

*Figure 1.* **GlobalHealthAtlas** comprises a total of 280,210 instances, featuring Question-Answer (138,267) and Single-Choice (141,943) formats across three difficulty levels (A/B/C). The domain composition (left) prioritizes high-impact areas like infectious diseases and health policy, balanced by a diverse long-tail of 15 complementary fields. The linguistic distribution (right) is anchored in English and extends across 17 languages to facilitate cross-lingual public health reasoning.

is computed as:

$$S_{\text{QA}} = 0.25Q + 0.35A + 0.25R + 0.15C \qquad (2)$$

Weights are prioritized to emphasize structural integrity ($Q$) and factual accuracy ($A$) as primary medical validity determinants. 7 senior global health experts, including WHO officials averaging over 8 years' experience, finalized the weights via structured voting. This consensus aligns our scoring of factual accuracy and reasoning with international public health standards. Strict zero-tolerance rules assign a score of 0 for domain irrelevance, language inconsistency, or missing materials; only candidates with $S \geq 4.5$ are retained. We theoretically validate this strategy in Appendix G.1, proving that threshold-based filtration bounds generalization risk by minimizing label noise. Evaluation criteria are detailed in Appendix B. We also conduct a sensitivity analysis on the hyperparameters to assess their impact on model evaluation stability, as shown in Table 12 and Table 13

Besides, GlobalHealthAtlas's raw materials are derived from authoritative official documents, primarily from WHO IRIS and other trusted public health sources, so LLMs are used mainly for structured extraction, format transformation, and candidate generation rather than fact invention or producing unconstrained knowledge. Moreover, evaluation dimensions and weighting coefficients are specified and corroborated with expert input, introducing exogenous human judgment into the quality control protocol.

We further conducted a dedicated human evaluation to provide manual attestation of dataset quality. Specifically, we sampled 5% of the full dataset, covering all 15 domains and

17 languages, including 7,077 single choice items and 6,933 question answer items, for a total of 14,010 examples. 14 experts manually scored these samples on a 5-point scale using same dimensions, criteria, and weighting schemes as LLM-based quality gate. The resulting average human score was 4.503, which provides supportive evidence that GlobalHealthAtlas exhibits high quality and the automated scoring is broadly aligned with expert judgment.

### 3.3. Data Split

To facilitate model development and rigorous assessment, the corpus is partitioned into three distinct subsets, all of which uniformly include high quality answers and CoT to support robust public health reasoning. The training set comprises 247,599 instances, while the test set consists of 27,511 instances, 5,100 instances are reserved for the construction and calibration of the evaluation model. This cohesive inclusion of reasoning paths across all partitions ensures the necessary supervision for both fine tuning and the systematic evaluation of complex logic.

### 3.4. Domain-Aligned Evaluator Construction and Verification

A domain specific, fully fine tuned evaluation model is developed to implement LLM-as-a-Judge protocol. This scorer functions both as benchmark tool and as critical quality gate within the data engineering pipeline.

### 3.4.1. DOMAIN-SPECIFIC MULTI-DIMENSIONAL SCORING RUBRIC

To accurately capture the complexity of public health reasoning, a scoring system comprising six orthogonal dimensions is designed: **Accuracy** evaluates the correctness of core facts, numerical data, and named entities; **Reasoning** checks whether the causal inference chain adheres to epidemiological logic and identifies logical leaps or circular reasoning; **Completeness** measures whether the response covers all Key Information Points (KIPs) present in the reference answer; **Consensus Alignment** assesses adherence to authoritative scientific consensus and safety guidelines; **Terminology Norms** examines the density and precision of professional terminology usage; and **Insightfulness** evaluates whether the model explains the underlying mechanisms (the "why") rather than merely describing phenomena. The rubric serves as core knowledge system for the evaluation model and guides the subsequent annotation of the training corpus. Detailed definitions and criteria for the six scoring dimensions are provided in Appendix C.

### 3.4.2. CONSTRUCTION STRATEGY FOR EVALUATOR FINE-TUNING DATA

High quality evaluation data construction pipeline integrates three key techniques for judgment capability and strong robustness:

- **Multi-Source Heterogeneous Sampling**: To mitigate risk of single-pipeline bias and ensure model optimizes for ground-truth medical consensus rather than internal stylistic artifacts, diverse LLMs (e.g., qwen, deepseek, GPT) are used to generate responses. This prevents style overfitting and ensures robust, style-invariant judgment, as shown in table 15 of Appendix D.1).

- **Hierarchical Consensus Filtering**: A strict three-tier mechanism consisting of cross scoring, automatic arbitration, and expert review is employed to ensure high confidence and ground truth labels. The formal scoring arbitration process is presented in Algorithm 1, and the proof of variance reduction is provided in Appendix G.2.

- **Adversarial Construction**: Robustness samples generated via adversarial perturbations are incorporated to enhance sensitivity to specious content and safety boundaries.

### 3.4.3. EVALUATOR MODEL FINE-TUNING

Based on the high-quality dataset constructed above, we employed Low-Rank Adaptation (LoRA) supervised fine-tuning on the open weight qwen3-8b model (Hu et al., 2022) to train the scorer Public-Evaluator. Through this process,

---

**Algorithm 1** Consensus-based Scoring Verification
1: Get scores $s_Q$ (qwen3-Max) and $s_D$ (deepseek-V3.2)
2: **if** $|s_Q - s_D| \leq 2$ **then**
3:     **Return** $s_Q$
4: **else**
5:     Adjudicate via gpt-5, obtaining $s_{adj}$
6:     **if** $s_{adj} \in \{s_Q, s_D\}$ **then**
7:         **Return** $s_{adj}$
8:     **else**
9:         **Return** Human Review
10:     **end if**
11: **end if**

---

six dimensional scoring criteria and the discriminative capability for complex samples were implicitly encoded into the model parameters. The resulting scorer is capable of stably outputting professional scores aligned with public health standards without the need for complex prompt engineering. Detailed construction procedures and training parameters can be found in Appendix D.

### 3.4.4. EVALUATOR CONSISTENCY AND STABILITY VERIFICATION

To comprehensively verify the reliability of our scorer, we conducted rigorous testing focused on Standard Agreement and Stability. Specifically, agreement was assessed by benchmarking against 100 independent human-annotated scores, while stability was evaluated through ten independent inference runs. The results are summarized in Table 3. We compared **Public-Evaluator** against generic large language models using identical prompts, as well as specialized evaluators in the medical and public health domains, including Clinical-Judge (Croxford et al., 2025) (few-shot prompting) and Expert-Guided Evaluator (Zhou et al., 2026) (structured reasoning).

In terms of agreement with expert judgment, our Public-Evaluator achieved the lowest Mean Absolute Error (MAE = 1.4259) and the highest Intraclass Correlation Coefficient (ICC = 0.9735), indicating that its judgment logic closely aligns with human experts. Furthermore, the scorer demonstrated exceptional robustness in repeated inference tests, achieving the highest Identical Rate (0.5533) and the lowest Score Standard Deviation (StdDev=0.2772). For further discussions regarding evaluator baselines, selection of Qwen3-8B, and analysis of potential stylistic overfitting are in Appendix D.4.

Finally, to explicitly address concerns regarding whether the Public-Evaluator over-penalizes valid but non-public-health-specific reasoning on out-of-distribution(OOD) medical tasks, we conducted a direct comparison experiment. We utilized our Public-Evaluator and a General-Medical Baseline Evaluator (GPT-5 prompted with standard clinical guidelines) to score 100 reasoning-intensive MedQA

*Table 3.* Comparison of Evaluator Standard Agreement and Stability. Agreement is measured against human experts (MAE, MSE, RMSE, ICC). Stability metrics include the rate of identical scores (IdenticalRate), average standard deviation (StdDev), and the ratio of score ranges exceeding 3 (RangeGT3Ratio). Detailed definitions are provided in Appendix E. The ↓ and ↑ arrows indicate the direction of better performance.

| Model | Standard Agreement | | | | Stability | | |
|---|---|---|---|---|---|---|---|
| | MAE ↓ | MSE ↓ | RMSE ↓ | ICC ↑ | IdenticalRate ↑ | StdDev ↓ | RangeGT3Ratio ↓ |
| **Public-Evaluator(ours)** | **1.4259** | **4.1296** | **1.9558** | **0.9735** | **0.5533** | **0.2772** | **0.0367** |
| claude-3-5-sonnet-20241022 | 1.5233 | 4.6500 | 2.1234 | 0.9186 | 0.3767 | 0.3180 | 0.0200 |
| gemini-3-flash-preview | 2.0375 | 7.2687 | 2.6232 | 0.8970 | 0.3533 | 0.5837 | 0.1750 |
| gpt-5-mini | 1.4383 | **4.1617** | **1.9431** | **0.9733** | 0.3067 | 0.3590 | 0.0200 |
| deepseek-v3.2 | 1.5883 | 4.8283 | 2.1459 | 0.9515 | 0.1250 | 0.7674 | 0.2050 |
| deepseek-r1 | 1.4621 | 4.3514 | 2.086 | 0.9642 | 0.285 | 0.4512 | 0.075 |
| kimi-k2-thinking | 1.5034 | 4.631 | 2.152 | 0.9578 | 0.3433 | 0.3955 | 0.052 |
| glm-4.7 | 1.5750 | 4.8050 | 2.1488 | 0.9512 | 0.4633 | 0.3226 | 0.0350 |
| qwen3-8b | 1.7340 | 5.9124 | 2.3722 | 0.9133 | 0.3264 | 0.4224 | 0.0451 |
| qwen-plus | **1.4216** | 4.2550 | 2.0001 | 0.9586 | 0.3450 | 0.3825 | 0.0417 |
| qwen3-max | 1.4833 | 4.2767 | 2.0104 | 0.9593 | 0.3588 | **0.3182** | **0.0153** |
| qwen-flash | 2.4033 | 10.4800 | 3.1473 | 0.8744 | 0.3667 | 0.7461 | 0.2744 |
| qwen3-32b | 1.4811 | 4.5190 | 2.0768 | 0.9519 | 0.1456 | 0.6827 | 0.1544 |
| Clinical-Judge | 2.0122 | 7.1094 | 2.6377 | 0.3591 | **0.4867** | 0.4810 | 0.1067 |
| Expert-Guided Evaluator | 2.1633 | 9.7833 | 3.0840 | 0.8541 | 0.4233 | 0.4677 | 0.075 |

responses generated by GPT-5 using our 10-point scale. As detailed in Table 4, the results explicitly validate that our evaluator maintains high consistency with the general baseline and exhibits no over-penalization risk in the reasoning dimension.

| Metric on OOD Data (MedQA) | Reasoning Dimension | Terminology Norms Dimension |
|---|---|---|
| Mean Score(Public-Evaluator) | 8.12 | 7.65 |
| Mean Score(General-Evaluator) | 8.18 | 8.15 |
| **Pearson Correlation ($r$)** | 0.83 | 0.78 |
| Over-Penalization Rate | 0.00% | N/A |

*Table 4.* Results of OOD evaluation on MedQA dataset, comparing Public-Evaluator with General-Medical baseline.

# 4. Experiment

To rigorously establish the effectiveness of both our proposed framework and the constructed public health dataset, as well as their robustness and generalization capability, we conduct a comprehensive suite of experiments, including large-scale benchmarking across models, domains, and languages (Sec. 4.2), ablation studies on supervised fine-tuning (Sec. 4.3), incremental data scaling experiments (Sec. 4.4), transfer experiments on cross-domain reasoning benchmarks(Sec. 4.5), robustness evaluation under diverse perturbations(Sec. 4.6).

The **Public-Model** was developed to validate the effectiveness of the dataset, using supervised fine-tuning with Low-Rank Adaptation (LoRA) on the training set of 247,599 instances. This model is capable of addressing specialized queries in the medical field with high professionalism. Specific hyperparameters and training details are provided in

Appendix F.

Throughout this section, unless otherwise specified, all evaluation scores are reported on a 0-10 scale, where 10 represents expert-level performance. The reported score for each model is calculated as the average across the six evaluation dimensions.

## 4.1. Data Leakage

To assess potential data leakage, we perform a continuation-based n-gram matching test. For each question, we sample five truncation points where the model must deterministically generate the immediately following $n$ words (temperature $= 0$) to count as a hit. Evaluating both 5-gram and 10-gram settings, we flag questions with a hit ratio $\geq 0.5$ as suspicious. Table 6 summarizes the leakage proportion ($prop_1$) and accuracy on suspicious questions ($prop_2$). Exact matches are rare, indicating minimal verbatim memorization risk.

## 4.2. Large-Scale Multilingual and Multi-Domain Benchmarking

Evaluation was conducted on a held-out test set of 27,511 instances. Responses generated by each LLM were scored with the Public Evaluator. Table 5 shows results for a representative benchmark subset, and full cross-domain and multilingual performance is provided in Figure 7, Figure 8 and Figure 9 in Appendix I. The results highlight two key insights into model capabilities within the public health domain:

*Table 5.* Main results on the **GlobalHealthAtlas** benchmark. The overall **Score** is computed by aggregating performance across the six evaluation dimensions, and this table reports results on a representative subset of the benchmark, covering four public health domains: **Infectious Disease Prevention, Surveillance and Response (IDP)**, **Health Policy, Systems and Global Governance (HPG)**, **Vaccination and Immunization (V&I)**, and **Child and Adolescent Health (CAH)**, and three languages (**English, Chinese, and Spanish**). Results are reported across three predefined difficulty levels (**A: Academic/Professional, B: General Knowledge, C: Public Awareness**) and question types (**Single-Choice and Question-Answer**). Models are grouped by series for readability. Higher values indicate better overall performance.

| Model | Score | Domain (Representative) | | | | Language | | | Difficulty | | | Type | |
|---|---|---|---|---|---|---|---|---|---|---|---|---|---|
| | | IDP | HPG | V&I | CAH | EN | ZH | ES | A | B | C | SC | QA |
| *Reasoning & Thinking Models* | | | | | | | | | | | | | |
| gemini-3-flash-preview-thinking | 6.282 | 6.344 | 6.212 | 6.343 | 6.506 | 6.284 | 6.498 | 6.388 | 6.499 | 6.194 | 6.328 | 6.625 | 5.911 |
| grok-3-mini | 6.352 | 6.478 | 6.187 | 6.410 | 6.997 | 6.343 | 6.350 | 6.683 | 6.884 | 6.148 | 6.413 | 7.133 | 5.551 |
| gpt-5-mini | 3.634 | 3.991 | 3.307 | 3.810 | 5.010 | 3.842 | 3.159 | 3.436 | 3.999 | 3.443 | 4.768 | 4.953 | 2.612 |
| claude-sonnet-4-5-20250929-thinking | 6.534 | 6.553 | 6.427 | 6.585 | 7.139 | 6.635 | 6.622 | 6.382 | 6.856 | 6.416 | 6.480 | 7.006 | **6.047** |
| kimi-k2-thinking | **6.930** | **7.023** | **6.806** | **6.945** | 7.927 | **6.985** | **7.188** | **7.153** | **7.632** | **6.659** | **7.001** | 7.912 | 5.917 |
| *qwen Series* | | | | | | | | | | | | | |
| qwen3-8b(Yang et al., 2025) | 5.934 | 6.034 | 5.822 | 5.922 | 6.906 | 5.956 | 6.028 | 6.090 | 6.648 | 5.650 | 6.158 | 6.987 | 4.855 |
| Public-Model(based on qwen3-8b) | 6.619 | 6.696 | 6.566 | 6.541 | **8.022** | 6.702 | 6.857 | 6.735 | 7.595 | 6.230 | 6.954 | **8.063** | 5.141 |
| qwen3-32b(Yang et al., 2025) | 5.259 | 5.247 | 5.309 | 5.064 | 6.520 | 5.415 | 5.324 | 5.276 | 5.979 | 4.977 | 5.423 | 6.357 | 4.135 |
| qwen2.5-7b-instruct(Qwen et al., 2025) | 3.660 | 3.852 | 3.469 | 3.696 | 4.482 | 3.634 | 3.824 | 3.985 | 4.127 | 3.468 | 3.909 | 4.418 | 2.884 |
| qwen2.5-72b-instruct(Qwen et al., 2025) | 4.839 | 4.768 | 4.829 | 4.874 | 4.548 | 4.869 | 4.733 | 4.951 | 4.578 | 4.941 | 4.801 | 4.578 | 5.107 |
| qwq-32b(Yang et al., 2025) | 6.173 | 6.286 | 6.038 | 6.223 | 6.937 | 6.175 | 6.104 | 6.348 | 6.809 | 5.922 | 6.340 | 7.127 | 5.194 |
| *DeepSeek Series* | | | | | | | | | | | | | |
| deepseek-r1(Guo et al., 2025) | 6.335 | 6.470 | 6.180 | 6.414 | 7.217 | 6.316 | 6.271 | 6.661 | 7.007 | 6.068 | 6.554 | 7.308 | 5.341 |
| deepseek-v3.2(DeepSeek-AI et al., 2025) | 6.365 | 6.471 | 6.221 | 6.448 | 6.914 | 6.278 | 6.089 | 6.847 | 6.897 | 6.159 | 6.448 | 7.142 | 5.569 |
| deepseek-r1-distill-qwen-7b | 5.004 | 4.947 | 5.039 | 4.891 | 5.735 | 5.091 | 5.136 | 5.123 | 5.570 | 4.780 | 5.164 | 5.899 | 4.087 |
| deepseek-r1-distill-llama-8b | 5.822 | 5.873 | 5.765 | 5.678 | 6.715 | 5.892 | 5.491 | 6.028 | 6.473 | 5.573 | 5.961 | 6.785 | 4.840 |
| deepseek-r1-distill-qwen-32b | 6.238 | 6.326 | 6.148 | 6.245 | 7.122 | 6.230 | 6.091 | 6.487 | 6.893 | 5.983 | 6.373 | 7.237 | 5.215 |
| *Other Mainstream Open-Source Series* | | | | | | | | | | | | | |
| Llama-3.2-3B-Instruct(Grattafiori et al., 2024) | 5.276 | 5.380 | 5.233 | 5.115 | 6.453 | 5.465 | 5.311 | 5.224 | 6.183 | 4.925 | 5.421 | 6.589 | 3.933 |
| Llama-3.1-8B-Instruct | 5.464 | 5.539 | 5.394 | 5.386 | 6.689 | 5.406 | 5.696 | 5.854 | 6.261 | 5.149 | 5.666 | 6.722 | 4.174 |
| glm-4.7 | **7.326** | **7.490** | **7.175** | **7.433** | **8.432** | **7.314** | **7.351** | **7.545** | **8.108** | **7.019** | **7.525** | **8.484** | **6.151** |

*Table 6.* Data Leakage Detection (5-gram vs. 10-gram). $prop_1$: leaked data proportion; $prop_2$: model accuracy on leaked data.

| Model | 5-gram | | 10-gram | |
|---|---|---|---|---|
| | $prop_1$ | $prop_2$ | $prop_1$ | $prop_2$ |
| claude-3-5-sonnet-20241022 | 0% | 0% | 0% | 0% |
| gemini-3-flash-preview-thinking | 0% | 0% | 0% | 0% |
| grok-3-mini | 0.05% | 100% | 0% | 0% |
| qwen3-14b(Yang et al., 2025) | 0% | 0% | 0% | 0% |
| qwen3-max(Yang et al., 2025) | 0.97% | 50% | 0% | 0% |
| deepseek-v3(DeepSeek-AI et al., 2025) | 0.05% | 0% | 0% | 0% |

strates consistent excellence across various domains, surpassing many larger open- and closed-source models. This highlights that high-quality, domain-aligned data can significantly enhance domain-specific reasoning capabilities, effectively bridging the gap caused by model scale.

Despite these quantitative gains, models may still exhibit specific cognitive breakdowns in high-precision tasks. Selected cases demonstrating suboptimal performance and their corresponding analyses are provided in Appendix H.

Notably, models score higher on Difficulty A than C. This stems from our data construction: Difficulty C is curated from specific regulations, whereas A prioritizes methodological inference. Thus, models excel at logical deduction even without specific parametric knowledge.

### 4.3. Ablation Study on Supervised Fine-Tuning

Table 7 presents a comprehensive ablation study comparing models with and without SFT (Daouadi et al., 2024),(Ouyang et al., 2022) on GlobalHealthAtlas, using the same training set of 247,599 instances for all fine-tuned models. Evaluation is conducted on six benchmarks: Instruction-

**Dominance of Strong Reasoning Models.** Reasoning-oriented models outperform standard instruction-following baselines. GLM-4.7 leads with a score of 7.326, particularly in Academic/Professional tasks (Difficulty A). Kimi-k2-thinking and Claude-sonnet-thinking also generalize well, highlighting the importance of advanced reasoning for public health.

**Efficacy of Domain-Specific Fine-Tuning.** A pivotal finding is the substantial performance leap achieved by our Public-Model (based on qwen3-8b). Compared to the vanilla qwen3-8b, our model achieves a +0.685 gain, securing third place overall in this benchmark. It demon-

*Table 7.* Evaluation results of reasoning models on public health (first 3 metrics) and medical (last 3 metrics) benchmarks.

| Model | SFT | Public Health | | | Medical | | |
|---|---|---|---|---|---|---|---|
| | | GlobalHealthAtlas (ours) | Instruction-public-health-dataset | publichealth-qa | MedQA | medical-o1-reasoning-SFT | shibing624-medical |
| qwen3-4b | ✗ | 5.773 | 5.821 | 5.611 | 7.031 | 5.464 | 4.839 |
| | ✓ | 6.382 (↑ 0.609) | 6.039 (↑ 0.218) | 5.922 (↑ 0.311) | 7.196 (↑ 0.165) | 5.628 (↑ 0.164) | 5.019 (↑ 0.180) |
| qwen3-8b | ✗ | 6.490 | 6.077 | 6.075 | 7.668 | 6.258 | 5.785 |
| | ✓ | 6.532 (↑ 0.042) | 6.096 (↑ 0.019) | 5.652 (↓ 0.423) | 7.967 (↑ 0.316) | 6.894 (↑ 0.636) | 6.268 (↑ 0.483) |
| qwen3-14b | ✗ | 6.134 | 6.190 | 5.771 | 7.744 | 5.906 | 5.322 |
| | ✓ | 6.565 (↑ 0.431) | 6.252 (↑ 0.062) | 6.249 (↑ 0.478) | 7.827 (↑ 0.083) | 6.643 (↑ 0.737) | 5.840 (↑ 0.518) |
| qwen3.5-122B-A10B | ✗ | 5.952 | 4.199 | 3.872 | 8.720 | 6.391 | 3.529 |
| | ✓ | 6.819 (↑ 0.867) | 6.669 (↑ 2.470) | 6.176 (↑ 2.304) | 9.343 (↑ 0.623) | 7.947 (↑ 1.556) | 5.901 (↑ 2.372) |
| Llama-3.2-3B-Instruct | ✗ | 5.258 | 4.175 | 3.557 | 6.326 | 4.452 | 3.057 |
| | ✓ | 6.105 (↑ 0.847) | 5.921 (↑ 1.746) | 4.705 (↑ 1.148) | 6.235 (↓ 0.091) | 4.493 (↑ 0.041) | 3.852 (↑ 0.795) |
| Llama-3.1-8B-Instruct | ✗ | 5.461 | 4.730 | 4.820 | 6.280 | 4.511 | 3.953 |
| | ✓ | 6.425 (↑ 0.964) | 6.195 (↑ 1.465) | 5.370 (↑ 0.550) | 6.720 (↑ 0.440) | 5.057 (↑ 0.546) | 4.211 (↑ 0.258) |
| Llama-3.3-70B-Instruct | ✗ | 5.467 | 4.739 | 4.836 | 6.266 | 4.499 | 3.995 |
| | ✓ | 6.760 (↑ 1.293) | 6.273 (↑ 1.534) | 5.831 (↑ 0.995) | 7.937 (↑ 1.671) | 6.086 (↑ 1.587) | 4.687 (↑ 0.692) |

public-health-dataset and xhluca_publichealth_qa are used in full, while GlobalHealthAtlas (ours), MedQA, medical-o1-reasoning-SFT, and shibing624-medical are each sampled with 2,000 questions.

The results show that domain-specific supervision substantially improves public health reasoning, particularly in parameter-constrained architectures such as qwen3-4B and Llama-3.1-8B. Larger models like Llama-3.3-70B and qwen3.5-122B-A10B also exhibit broad improvements across both public health and general medical tasks, demonstrating GlobalHealthAtlas's strong generalization capability and its effectiveness as a foundation for domain-aligned fine-tuning. Minor variations in a few metrics may arise from fine-grained differences between benchmark evaluation criteria and the objectives emphasized during supervised fine-tuning.

### 4.4. Incremental Data Scaling Analysis

The impact of data scaling on model reasoning was investigated by conducting experiments using the qwen3 series models (4B, 8B, 14B) across varying data proportions from 0% to 100%, with all fine-tuned models using identical fine-tuning hyperparameters. All evaluations were conducted on a subset of 2,000 samples randomly drawn from the test set. Table 8 presents the overall results, indicating that performance generally improves with data availability but plateaus or slightly degrades beyond the 60% mark, suggesting diminishing returns at higher data volumes.

Further evaluations were performed via stratified sampling of 600 samples per difficulty level to deepen the analysis of these trends. As illustrated in Figure 2, the models exhibit a non-linear performance trajectory. While all models benefit from initial data scaling, the difficulty stratification reveals that larger models (14B) maintain superior robustness and

absolute performance in complex scenarios compared to their smaller counterparts.

*Table 8.* Experimental results of qwen3 series models across different data scaling proportions.

| Model | 0% | 10% | 30% | 60% | 100% |
|---|---|---|---|---|---|
| qwen3-4b | 5.7727 | 6.2447 | 6.3127 | 6.3694 | 6.3818 |
| qwen3-8b | 6.4896 | 6.4185 | 6.4968 | 6.5279 | 6.5316 |
| qwen3-14b | 6.1335 | 6.4423 | 6.5520 | 6.5748 | 6.5648 |

### 4.5. Cross-Domain Transfer Evaluation

To further evaluate cross-domain generalization on additional benchmarks, we conduct experiments on the full GPQA(Rein et al., 2024) and MMLU-pro(TIGER-Lab, 2024) datasets, both of which are general science benchmarks spanning multiple disciplines.

As shown in Table 9, SFT on GlobalHealthAtlas improves cross-domain performance for qwen3-4b and qwen3-8b on both benchmarks. For qwen3-14b, SFT yields gains on GPQA but leads to a slight decrease on MMLU-pro, possibly due to stronger source-domain bias introduced during fine-tuning in larger models. To mitigate trade-off, we further use weight-space interpolation between the SFT and base checkpoints of qwen3-14b, which recovers MMLU-pro performance while maintaining strong GlobalHealthAtlas performance, as shown in Table 10.

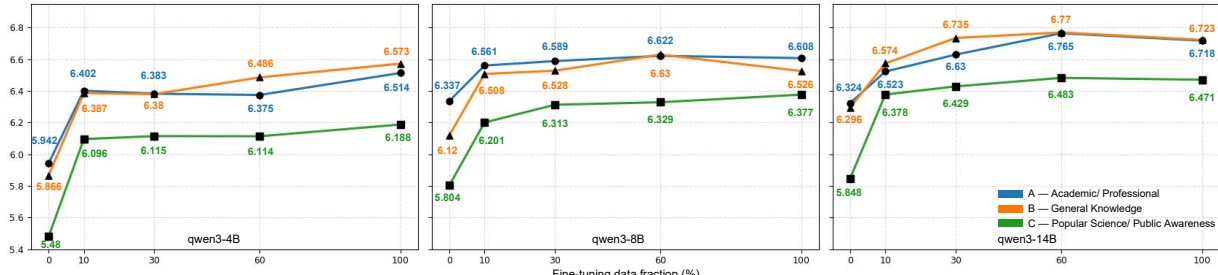

*Figure 2.* Performance variation across question difficulty levels under different fine-tuning regimes. The figure illustrates how model reasoning capability changes with increasing question difficulty from **C** (Popular Science / Public Awareness) to **B** (General Knowledge) and **A** (Academic / Professional), evaluated across multiple fine-tuning data fractions. Results are reported for **qwen3-4b**, **qwen3-8b**, and **qwen3-14b**, highlighting the interaction between difficulty stratification and fine-tuning scale.

*Table 10.* Weight interpolation results for qwen3-14b on Global-HealthAtlas and MMLU-pro.

| qwen3-14b | Strategy | Global HealthAtlas | MMLU-pro |
|---|---|---|---|
| Base | No Fine-tuning | 6.1335 | 7.426 |
| Pure SFT | 100% Fine-tuning | 6.5648 | 7.309 |
| Interpolated | 0.85 Weight Interpolation | 6.512 | 7.545 |

*Table 9.* Performance comparison of qwen3 series models (Base vs. SFT) on GPQA and MMLU-pro benchmarks

| Model | GPQA | | MMLU-pro | |
|---|---|---|---|---|
| | Base | SFT | Base | SFT |
| qwen3-4b | 4.948 | 5.342 | 6.762 | 7.043 |
| qwen3-8b | 5.342 | **5.904** | 7.139 | **7.405** |
| qwen3-14b | 5.277 | **5.723** | **7.426** | 7.309 |

## 4.6. Robustness Evaluation under Input Perturbations

Table 11 presents the robustness performance of different reasoning models on 1,000 multiple-choice questions sampled from our GlobalHealthAtlas dataset, with accuracy computed based on the correctness of the selected option. We systematically assess both base and fine-tuned models under four robustness settings: $\mathcal{S}_0$ (original input without perturbation), $\mathcal{S}_1$ (linguistic reformulation), $\mathcal{S}_2$ (noise-corrupted input), and $\mathcal{S}_3$ (cross-lingual translation).

Overall, domain-aligned models exhibit substantially stronger stability. Notably, PUBLIC-MODEL maintains over 86% accuracy under $\mathcal{S}_1$ paraphrasing and 84% under $\mathcal{S}_3$ cross-lingual transfer. Under $\mathcal{S}_2$ noise injection, the most challenging perturbation, both Public-Model and qwen3-14b retain around 77% accuracy, while general-purpose models such as Llama-3.1-8B-Instruct degrade sharply, dropping to 50%. These results demonstrate that our domain-adapted models can maintain high stability un-

der realistic input perturbations, highlighting the importance of domain-aligned data and evidence-consistent training for robust public health reasoning(Shi et al., 2026).

*Table 11.* Robustness of both base and fine-tuned reasoning models is evaluated under four input perturbation settings ($\mathcal{S}_0$, $\mathcal{S}_1$, $\mathcal{S}_2$, and $\mathcal{S}_3$), with results reported as accuracy (%).

| Model | $\mathcal{S}_0$ | $\mathcal{S}_1$ | $\mathcal{S}_2$ | $\mathcal{S}_3$ |
|---|---|---|---|---|
| qwen3-8b | **85.41**% | **83.91**% | **77.26**% | 81.68% |
| Public-Model | **87.30**% | **86.00**% | **77.34**% | **84.40**% |
| deepseek-r1-distill-qwen-7b | 45.55% | 49.40% | 40.00% | 61.52% |
| claude-sonnet-4-5-20250929-thinking | 64.69% | 58.00% | 64.58% | 65.89% |
| qwen3-14b | 83.30% | 82.70% | 77.10% | **81.90**% |
| Llama-3.1-8B-Instruct | 70.89% | 70.53% | 50.00% | 73.09% |

## 5. Conclusion

**GlobalHealthAtlas** is introduced as a large scale, multilingual dataset designed to facilitate population level public health reasoning across diverse domains and difficulties. We provide a principled construction pipeline and structured dataset splits alongside a domain aligned evaluation framework tailored for evidence grounded and safety sensitive tasks. To ensure rigorous assessment and application, we developed a specialized evaluator and Public-Model. Experimental results demonstrate that while state of the art models exhibit substantial gaps in robust reasoning particularly under realistic perturbations and cross lingual settings, fine tuning on high quality domain specific data significantly enhances performance. Notably, our findings suggest that sufficient model capacity facilitates the absorption of domain specific knowledge while simultaneously improving reasoning abilities in the medical domain. By establishing a rigorous foundation for domain aligned datasets and evaluation methodologies, we contribute to the development of reliable LLMs for real world decision making.

## Acknowledgments

This work was supported by Shanghai Municipal Science and Technology Major Project under Grant ZD2021CY001, the National Natural Science Foundation of China under Grant 62271508, and Beijing Municipal Foundation of Natural Sciences-Xiaomi Innovation Joint Foundation under Grant L233018.

We are deeply grateful to Professor Fan Wu, Dean of the Shanghai Institute of Infectious Disease and Biosecurity, and Vice Dean of Shanghai Medical College, Fudan University, for her invaluable guidance and unwavering support throughout this research. Her profound insights into China's public health practice were instrumental in shaping the direction of this work. This article would not have been possible without her encouragement and stewardship. We are profoundly indebted to Dr. Gauden Galea, former WHO Representative in China and Honorary Professor at the University of Malta, who provided the core intellectual orientation for our research. It was he who steered us toward the WHO Institutional Repository for Information Sharing (IRIS), guiding us in transforming authoritative public health literature into high-quality, machine-readable knowledge assets for the era of Large Language Models. We also extend our sincere thanks to the staff of the WHO: Yu Zhao from WHO headquarters, Geneva; Mengji Chen, Dilip Hensman from WHO Western Pacific Regional Office; and Mengjuan Duan from WHO Country Office in the Philippines. Their support and facilitation were essential to the completion and validation of this work. Besides, We thank the Division of Data, Strategy and Innovation (DSI), WHO Regional Office for the Western Pacific, for their valuable discussions and feedback.

## Impact Statement

GlobalHealthAtlas has four main limitations. First, its reliance on WHO IRIS and other authoritative public-health documents ensures safety, factual grounding, and standardization, but may introduce an international policy-oriented bias and underrepresent localized healthcare nuances, particularly in the Global South. Second, extending the dataset to non-WHO guidelines is feasible but requires careful source vetting, and protocol harmonization to avoid introducing unreliable evidence. Third, although the dataset covers 17 languages and shows generally stable cross-lingual transfer, its language distribution remains imbalanced; the most underrepresented languages still exhibit weaker performance. Fourth, financial disparities across medical fields introduce a publication bias within these authoritative sources, over-representing well-funded diseases while underrepresenting niche technical domains like primary healthcare. Future work will incorporate more regional public-health sources and adopt coverage-aware sampling to improve both geographic and linguistic balance.

**Ethical Considerations and Societal Impact:** As GlobalHealthAtlas is primarily grounded in WHO documents, it inherently reflects the policy frameworks of international organizations, which may inadvertently overlook localized healthcare nuances or traditional medical practices in the Global South.

Furthermore, public health discourse involves inherently **sensitive topics**, including reproductive health, immunization strategies, and substance control. We recognize that LLM reasoning may still carry the normative biases of the source materials. To mitigate potential risks, we emphasize that GlobalHealthAtlas is intended for **research evaluation purposes only**. Finally, **linguistic accessibility** remains an ongoing challenge for low-resource languages. We are committed to future audits focused on cross-regional fairness and the inclusion of broader, community-driven health perspectives to ensure a more equitable representation of global public health knowledge.

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

## Appendix for GlobalHealthAtlas

### A. Statistics of GlobalHealthAtlas

The following part displays the English and Spanish versions of GlobalHealthAtlas, followed by representative examples across varying difficulty levels. In addition, we present a language-difficulty heatmap that illustrates the distribution of samples across different languages and complexity levels, providing an overview of the dataset's multilingual coverage and difficulty balance.

---

**A English Question of GlobalHealthAtlas**

**id:** 136313

**question:** What are the key components of the USA's Combating Antimicrobial Resistant Bacteria Action Plan released in 2015, and how does it address the emergence and spread of antimicrobial-resistant bacteria?

**answer:** The key components of the USA's 2015 Combating Antimicrobial Resistant Bacteria Action Plan include a One Health approach involving multiple agencies: CDC as the national civilian antimicrobial resistance reference laboratory, DoD for military reference lab functions, USDA-FSIS for animal-origin antimicrobial susceptibility testing and NARMS Cecal Sampling Program, USGS for environmental antimicrobial resistance research, and FDA for drug residue monitoring in foods. It addresses antimicrobial resistance through enhanced surveillance systems (including NARMS and GLASS), antibiotic stewardship policies (DoD and CDC initiatives), and industry commitments to judicious antimicrobial use in food-producing animals via FDA GFI #209.

**complexCOT:** The 2015 US Combating Antimicrobial Resistant Bacteria Action Plan employs a One Health approach, assigning specific agencies to address resistance across human, animal, and environmental sectors. CDC serves as the national civilian antimicrobial resistance reference laboratory, while DoD handles military reference functions. USDA-FSIS conducts animal-origin antimicrobial susceptibility testing and operates the NARMS Cecal Sampling Program, USGS studies environmental antimicrobial resistance, and FDA monitors drug residues in foods. The plan tackles resistance emergence through enhanced surveillance systems like NARMS and GLASS, antibiotic stewardship policies implemented by DoD and CDC, and FDA Guidance for Industry #209 which commits to judicious antimicrobial use in food-producing animals.

**language:** English

**domain:** Infectious Disease Prevention, Surveillance and Response

**difficulty:** B

**label:** Question-Answer

---

**A Spanish Question of GlobalHealthAtlas**

**id:** 131

**question:** ¿Cuál es la ingesta máxima diaria de sodio recomendada por la OMS para la población general? A. 100 mmol/d (6 g de sal) B. 65 mmol/d (4 g de sal) C. 50-60 mmol/d D. 20-40 mmol/d

**answer:** A

**complexCOT:** Primero, la OMS recomienda un máximo diario de sodio de 100 mmol/d (6 g de sal) para la población general, no un promedio. Luego, la opción A coincide exactamente con esta recomendación oficial. Después, la opción B (65 mmol/d) es una meta más estricta para casos específicos, no el límite máximo general. Finalmente, la opción C (50-60 mmol/d) sugiere reducciones adicionales para mayor beneficio saludable, y la D (20-40 mmol/d) corresponde a la ingesta adecuada de Australia/Nueva Zelanda, no a la OMS.

**language:** Spanish

**domain:** Nutrition and Food Safety

**difficulty:** B

**label:** Single-Choice

---

**An Instance of Difficulty A of GlobalHealthAtlas**

**id:** 429

**question:** Which of the following is a common cause of diagnostic errors in primary care? A. Overuse of rapid diagnostic tests B. Failure to inform patients of test results C. Excessive reliance on patient feedback D. Strict adherence to clinical guidelines

**answer:** B

**complexCOT:** First, diagnostic errors in primary care commonly arise from systemic communication failures within healthcare workflows. Next, failure to inform patients of test results directly creates information gaps that prevent timely diagnosis and intervention. Then, overuse of rapid diagnostic tests (option A) is not a primary error cause but rather a tool requiring proper application. Excessive reliance on patient feedback (option C) is not supported as a common error source, as patient input typically enhances diagnostic accuracy. Strict adherence to clinical guidelines (option D) reduces errors by standardizing care, making it an error-prevention strategy, not a cause. Finally, this communication breakdown represents a well-documented vulnerability in health systems that directly contributes to diagnostic errors.

**language:** English

**domain:** Health Policy, Systems and Global Governance

**difficulty:** A

**label:** Single-Choice

**An Instance of Difficulty B of GlobalHealthAtlas**

**id:** 302

**question:** What is the primary route of transmission for tuberculosis? A. Contaminated water B. Mosquito bites C. Airborne droplets D. Sexual contact

**answer:** C

**complexCOT:** First, tuberculosis spreads primarily through airborne droplets when an infected person coughs or sneezes. Next, contaminated water transmits gastrointestinal diseases like cholera, not respiratory infections such as TB. Then, mosquitoes transmit vector-borne diseases like malaria, not TB, which requires direct respiratory exposure. Finally, sexual contact is irrelevant as TB is not a sexually transmitted infection and lacks evidence for this transmission route."

**language:** English

**domain:** Infectious Disease Prevention, Surveillance and Response

**difficulty:** B

**label:** Single-Choice

## An Instance of Difficulty C of GlobalHealthAtlas

| | |
|---|---|
| **id:** | 264 |
| **question:** | Which of the following best describes a 'complete' smoking ban according to the WHO FCTC? A. Smoking is permitted in designated areas with ventilation B. Smoking is not permitted except in residences and equivalent long-term facilities C. Smoking is allowed in all public places after certain hours D. Smoking is prohibited only in government buildings |
| **answer:** | B |
| **complexCOT:** | First, a WHO FCTC c̈omplete s̈moking ban prohibits smoking everywhere except homes and long-term care facilities like nursing homes. nThen, option A is wrong because ventilation or designated areas don't eliminate secondhand smoke harm. Next, option C is incorrect since allowing smoking after hours still permits it in public spaces. Finally, option D is too limited, as a complete ban covers all public places, not just government buildings. |
| **language:** | English |
| **domain:** | Tobacco and Substance Use Control |
| **difficulty:** | C |
| **label:** | Single-Choice |

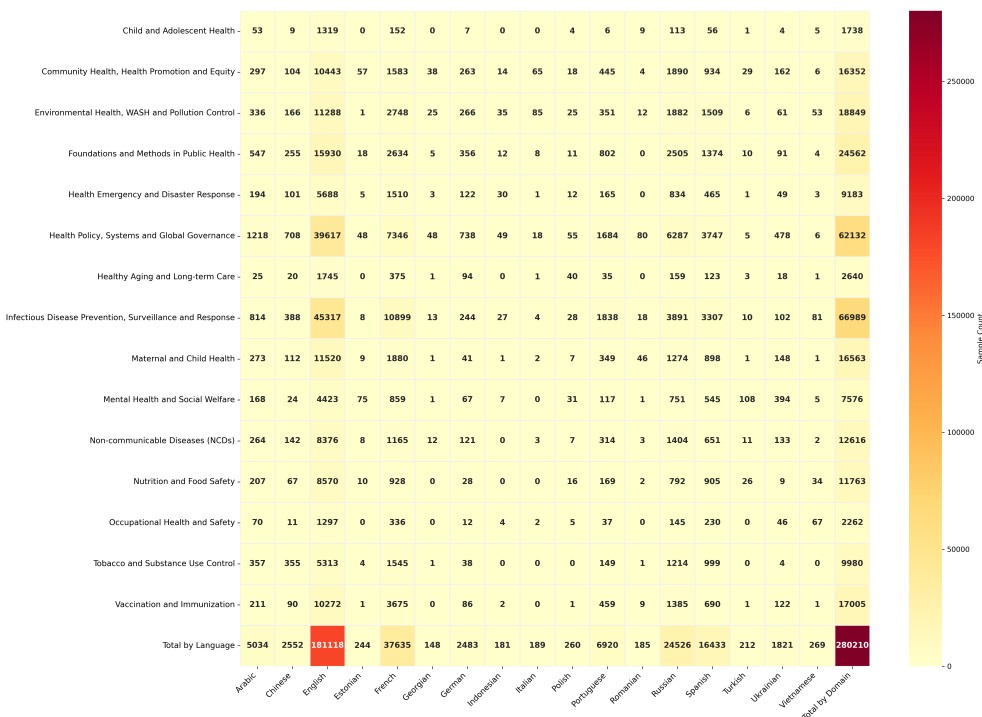

*Figure 3.* Heatmap illustrating the distribution of samples across languages and domains

## B. Details of GlobalHealthAtlas pipeline

### B.1. DATA PROCESSING PIPELINE

When collecting public health data, we designed the processing pipeline illustrated in Figure 4. First, raw public health documents, including research papers and policy reports, are converted into a structured text format. Next, LLM is employed to extract QA pairs, individual questions, and corresponding answers from the converted text. To minimize potential hallucinations by the LLM, the extracted content is cross verified against the original source documents. Finally, we analyze the structural features of each text fragment to accurately align individual questions with their corresponding answers, ensuring consistency and reliability.

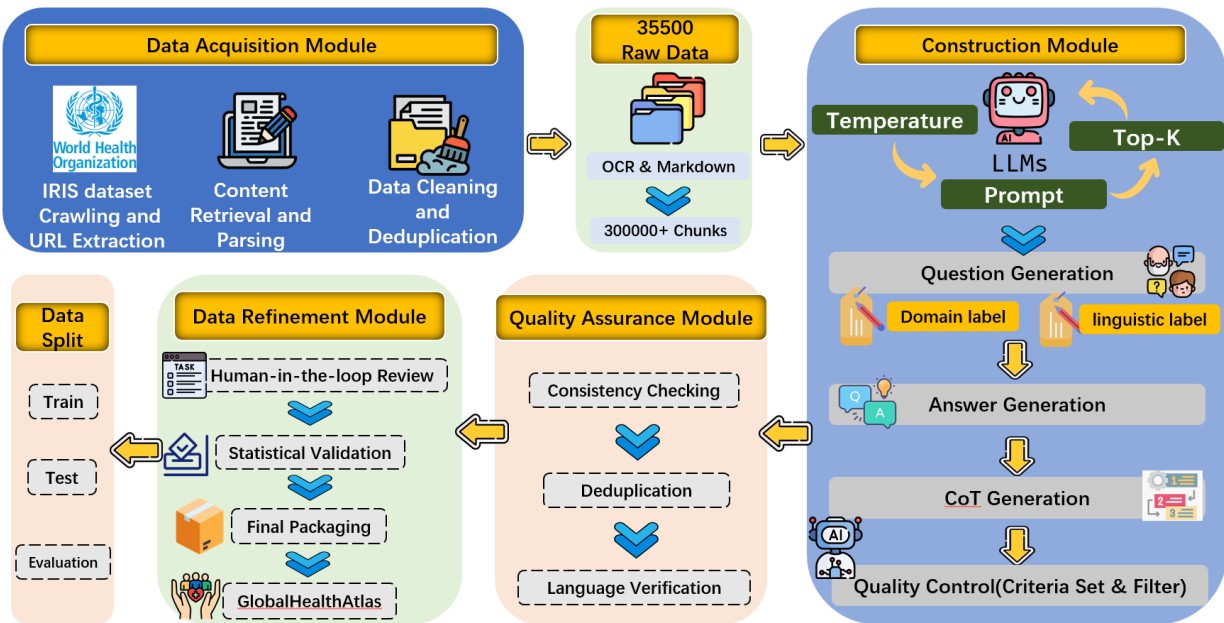

*Figure 4.* Pipeline of GlobalHealthAtlas Processing

### B.2. SENSITIVITY ANALYSIS AND ROBUSTNESS OF THE WEIGHTING SCHEME

To empirically validate our format-aware weighting scheme, we conduct a multi-model sensitivity analysis on the scoring hyperparameters. Our results demonstrate that the evaluation framework is highly robust to $\pm 5\%$ perturbations in primary weights, maintaining an average Kendall's $\tau > 0.92$ and a Top-10% overlap of over 85% across four diverse LLMs. Conversely, applying a uniform weighting scheme leads to a significant degradation in alignment and erroneously promotes factually or structurally flawed samples. This contrast empirically validates that our weight assignments are not merely based on intuition, but act as a necessary and robust quality gate for public health datasets.

The detailed sensitivity and robustness analysis for both Single Choice(SC) and Question Answering(QA) formats are presented in Table 12 and Table 13, respectively. All metrics are reported as Kendall's $\tau$ / Top-10% Overlap, based on 100 samples.

---

**Prompt of Question Generation of QA pairs**

```
# Role: Text Question Generation Expert

## Profile:
- Description:
  You are a professional Public Health Text Analysis and Question Design Expert, able to extract key
      information from complex texts and generate high-quality questions for fine-tuning public-health-related
      models.

  The question language must strictly match the language of the original text.
  Use the same language, vocabulary, and tone as the source.
```

*Table 12.* Sensitivity Analysis for SC format ($S = 0.45Q + 0.25A + 0.20R + 0.10C$)

| Model | Base (Expert) (0.45/0.25/0.20/0.10) | Perturb (-5%) (0.40/0.30/0.20/0.10) | Perturb (+5%) (0.50/0.20/0.20/0.10) | Uniform (0.25/0.25/0.25/0.25) |
|---|---|---|---|---|
| DeepSeek-V3 | 1.00 / 100% | 0.932 / 90% | 0.941 / 90% | 0.785 / 70% |
| GPT-o3-mini | 1.00 / 100% | 0.951 / 90% | 0.948 / 90% | 0.812 / 80% |
| Qwen-Max | 1.00 / 100% | 0.915 / 80% | 0.927 / 90% | 0.743 / 60% |
| Kimi-k2-thinking | 1.00 / 100% | 0.922 / 90% | 0.918 / 80% | 0.761 / 70% |
| **Average** | 1.00 / 100% | 0.930 / 87.5% | 0.933 / 87.5% | 0.775 / 70% |

*Table 13.* Sensitivity Analysis for QA format ($S = 0.25Q + 0.35A + 0.25R + 0.15C$)

| Model | Base (Expert) (0.25/0.35/0.25/0.15) | Perturb (-5%) (0.30/0.30/0.25/0.15) | Perturb (+5%) (0.20/0.40/0.25/0.15) | Uniform (0.25/0.25/0.25/0.25) |
|---|---|---|---|---|
| DeepSeek-V3 | 1.00 / 100% | 0.928 / 90% | 0.935 / 90% | 0.841 / 80% |
| GPT-o3-mini | 1.00 / 100% | 0.942 / 90% | 0.950 / 90% | 0.865 / 80% |
| Qwen-Max | 1.00 / 100% | 0.908 / 80% | 0.912 / 80% | 0.812 / 70% |
| Kimi-k2-thinking | 1.00 / 100% | 0.916 / 80% | 0.925 / 90% | 0.829 / 70% |
| **Average** | 1.00 / 100% | 0.923 / 85.0% | 0.930 / 87.5% | 0.836 / 75% |

```
   Each question must be self-contained and understandable without seeing the original text.

 - Input Length: {{textLength}} characters

 - Output Goal:
   Generate **only one** single, most essential, high-quality question, suitable for constructing a
     question-answer training dataset.

   The question must be:
   - **Standalone**: understandable on its own, without any reference to "the text" or "this document".
   - **Substantive**: contains key conditions, context, or focus clearly; not overly short or vague.
   - **Single-focus**: only one core interrogative target (one main thing being asked).

## Language Detection Rules
- Automatically detect the language of the input text.
- Use that same language for the question.
- If the text is bilingual, follow the majority language.
- Keep technical terms or proper names in their original form.
- Do not translate or switch languages unless explicitly instructed.

## Skills:
1. Fully comprehend the source text, identifying key concepts, facts, actors, numerical data, risk factors,
     interventions, and main conclusions.
2. Design questions with a clear answer focus, adding brief background clauses only when needed for clarity.
3. Control difficulty and question type to ensure diversity and representativeness (within the scope of
     public health).
4. Use only the information in the given text; do not introduce external facts or assumptions.
5. Ensure that the question strictly follows the language rules above and is clearly formed as a proper
     question.

## Workflow:

1. **Text Parsing**
   - Read the entire passage.
   - Identify important entities, events, numerical data, interventions, risk factors, and main conclusions.

2. **Question Design**
   Select the optimal focus point based on information density and importance {{gaPromptNote}}, then:

   - Frame **one** question that:
     - Has a clear and unique answer target.
     - Is sufficiently detailed (not a very short or vague question).
     - Uses only **one** main interrogative clause.
```

    – Append a difficulty tag at the end of the question, following the "Difficulty Matching"
      rules, in the format: `#Difficulty: A/B/C`.
    – Append a language tag by English at the end of the question, in the format: `#Lang: <Language>`, that
      matches the detected language (e.g., `#Lang: English`, `#Lang:Chinese `, etc.).

3. **Quality Check**
   For the final question, verify that it:
   – Has an answer that can be directly found or reliably inferred from the text. {{gaPromptCheck}}
   – Reflects key themes or perspectives of the text, not a trivial detail.
   – Is fully standalone, with no mention of "the text", "this guide", "the passage", etc.
   – Expresses only one core interrogative focus, even if background clauses are present.
   – Is clearly worded, unambiguous, and properly formed as a question.

## Strict Prohibitions (Meta-reference & "according to" Ban):

You **must never include**:
– Any meta-reference to the container of the text, such as:
  – "According to the text/passage/document/article"
  – "Based on the paragraph/content/figure/image"
  – "From the passage"
  – "As shown in the figure/table/chart"
  – "In this guide/report/study/chapter"
  – Or any equivalent expressions in any language.

– Any "according to / as outlined in / as recommended in / de acordo com " structures in **any** language,
    including when referring to WHO, UNICEF, or other institutions.

If you detect any such expression in your drafted question, you **must** rewrite the question to remove it
    before producing the final output.

## Difficulty Matching:

Assign **exactly one** difficulty tag at the end of the question:

– **#Difficulty: A – Academic / Professional level**
  – For university, graduate, or specialist level.
  – Requires deeper understanding of epidemiology, health policy, study design, or statistics (e.g. RCTs,
    risk ratios, hazard ratios).

– **#Difficulty: B – General Knowledge level**
  – For basic public health understanding.
  – Tests applied understanding of disease prevention, causal relations, and everyday health logic, without
    advanced methodology.

– **#Difficulty: C – Popular Science / Public Awareness level**
  – For general population health literacy or school level.
  – Focuses on simple facts or basic health awareness (risks, prevention, basic symptoms).

Whenever a question can be correctly answered by a non-expert adult using simple health knowledge, prefer
      **#Difficulty: C**.
When unsure between **#Difficulty: B** and **#Difficulty: C**, always choose **#Difficulty: C**.

## Constraints:
1. All questions must be strictly based on the given text; no external assumptions or external data.
2. Generate **exactly one** question (no more than one).
3. The question should reflect a key theme or perspective of the text, not meta-information (such as author,
    chapter title, table of contents, etc.).
4. Never use meta-reference phrases or "according to ..." style expressions in any language.
5. The question should not be excessively short; include only the necessary background for clarity, but avoid
    unnecessary complexity.
6. Every question must express only one core interrogative focus.

## Output Format:
– The output **must** be a valid JSON array containing **exactly one** string element.
– That single string is the generated question (with the difficulty and language tags appended).
– Strictly follow the structure:
["question"]
##Output Example:
For an English input text, a valid example output is:
["In the African Region in 2018, what proportion of people living with HIV who knew their status were
    accessing treatment? #Difficulty: B #Lang: English"]

## Text to Analyze:
{{text}}

## GA Instruction (Optional):
{{gaPrompt}}

**Prompt of Answer Generation of QA pairs**

```
## Role: Text Answer Generation Expert

**Profile**

**Description**
You are a professional public health answer-generation expert who extracts precise information from the
    provided reference content and produces a concise, accurate answer for a single-focus question.

The answer's language must match the question's language.

Produce one final answer string with a domain tag.

---

## Input

- **Question:** `{{question}}`

## Reference Content:

------ Reference Content Start ------
{{text}}
------ Reference Content End ------

---

## Domain Taxonomy (choose ONE best-fitting domain)

1.Infectious Disease Prevention, Surveillance and
2.Health Policy, Systems and Global Governance
3.Foundations and Methods in Public Health
4.Environmental Health, WASH and Pollution Control
5.Vaccination and Immunization
6.Maternal and Child Health
7.Community Health, Health Promotion and Equity
8.Non-communicable Diseases (NCDs)
9.Nutrition and Food Safety
10.Tobacco and Substance Use Control
11.Health Emergency and Disaster Response
12.Mental Health and Social Welfare
13.Healthy Aging and Long-term Care
14.Occupational Health and Safety
15.Child and Adolescent Health
---

## Language Rules

- Auto-detect the question's language; answer in the same language and terminology.
- Preserve proper nouns and technical terms as written; do not translate or switch languages unless
    explicitly instructed.

---

## Skills

- Identify the question's single inquiry target; extract direct evidence from the reference content (use
    provided options only as constraints if present).
- Answer with the minimum necessary information: include units/time frames for numbers; keep concepts
    precise; avoid extraneous background.
- If the content is insufficient, output "Insufficient information." (or the equivalent in the question's
    language) and still provide one domain tag.
- Ensure the answer is self-contained; do not restate irrelevant parts of the question.
- Append exactly one most appropriate domain tag.

---

## Workflow

## 1. Align & Parse
- Read `{{text}}` and `{{question}}` (and options if present); isolate the single inquiry target and
    constraints.

## 2. Evidence Extraction
- Pull the shortest evidence chain that directly supports the answer; verify correctness of values, entities,
    and time frames.
```

```
## 3. Answer Generation
- Write one concise, definitive answer in the question's language; include one necessary qualifier if
    essential.
- If unsupported by the text, write "Insufficient information." (in the same language).

## 4. Domain Tagging
- Append at the end of the answer string:
  `#Domain: <one label from the taxonomy above>`.

---

## Strict Prohibitions

- No meta-references to containers/mediums (e.g., "according to the text/figure/passage/this
    study/report/guide").
- No "according to / as stated in / per ..." structures (including institutions like WHO/UNICEF).
- No facts or inferences beyond the provided reference content.

---

## Output Format

- The output **must** be a JSON array with exactly one string element:

["answer #Domain: <Domain_Label>"]

The domain label must exactly match one of the 31 taxonomy options above.

## Output Examples:

"Coverage of at least 95% is required to sustain herd immunity against measles. #Domain: Vaccination and
    Immunization Strategies"
Insufficient information example:

"Insufficient information. #Domain: Principles and Applications of Epidemiology"

{{templatePrompt}}
{{outputFormatPrompt}}
```

## Prompt of CoT Generation of QA pairs

```
# Role: Chain-of-Thought Optimization Expert

## Profile
**Description:**
You are an expert in optimizing chain-of-thought reasoning. You are especially familiar with reasoning logic
    and terminology standards in the field of **Public Health**. You can process a given chain of thought,
    remove citation-related expressions, and refine the reasoning while maintaining rigor. Your goal is to
    produce a natural, clear, and professionally aligned reasoning process that reflects public health
    analytical standards.

## Skills
- Accurately identify and remove citation-related expressions in the chain of thought.
- Refine the reasoning without changing the conclusion; remove redundant or repetitive content while
    preserving essential logic.
- Master professional logic frameworks used in public health (e.g., epidemiology, health policy, health
    promotion, environmental health, disease prevention and control).
- Preserve all technical terms and proper nouns.
- Automatically detect the language of the original question and output the explanation in the same language.

## Language Rules
- Automatically detect the language used in the original question.
- The reasoning process **must** be written in the same language as the original question.
- Preserve all proper nouns and technical terms, especially public health terminology.
- Refine reasoning by removing duplication or unnecessary details, but keep all logic essential for the
    conclusion.
- Maintain the rigorous logical style typical of public health reasoning, such as:
  **Risk assessment – Causal pathway analysis – Intervention mechanism**

## Workflow

### 1. Analyze the nature of the question (aligned with public health logic)
- Identify which public health domain it belongs to (epidemiology, health policy, disease prevention,
    environmental exposure, etc.).
```

```
- Understand the causal reasoning structure required to reach the answer.

### 2. Read and interpret the original chain of thought
- Identify all reasoning steps and their relationships to the final conclusion.
- Remove any citation-related expressions such as "according to the reference," "as the document states," etc.

### 3. Refine the reasoning chain
- Remove content that is repetitive or irrelevant to the conclusion.
- Merge logically similar steps to form a more concise chain without skipping logic.
- Follow the standard public health reasoning order:
  **Background/Risk - Exposure/Mechanism - Health Impact - Intervention/Conclusion**

### 4. Apply public health terminology and professional structure
- If the original chain of thought is loosely structured, reorganize it using public health analytic format,
    such as:
  **"First assess risk factors - then analyze exposure and health outcomes - finally derive the intervention
    - based conclusion."**

### 5. Maintain language consistency and natural flow
- Use the same language as the original question throughout.
- Avoid mechanical phrasing; ensure the reasoning flows naturally.

### 6. Final Check
- Ensure all citation-related expressions are removed.
- Ensure the logic remains complete and terminology is accurate.
- Ensure the reasoning naturally leads to the correct answer.

---

## Original Question
{{originalQuestion}}

## Answer
{{answer}}

## Original Chain of Thought
{{originalCot}}

---

## Constraints
- Remove all expressions such as "according to the reference," "reference material," "the document states,"
    etc.
- The reasoning must remain complete and logical; no gaps are allowed.
- Automatically detect the language of the question and use the same language in the explanation.
- Preserve all proper nouns and technical terms, especially public health terminology.
- Refine the reasoning by removing all redundancy and keeping only essential logic.
- The reasoning must follow public health logical frameworks (e.g., risk factor analysis, interventions,
    epidemiological inference).
- Do **not** include phrases such as "optimized chain of thought."
- The reasoning should follow a natural cognitive sequence (e.g., "First..., Then..., Next..., Finally...").
```

## Prompt of Quality Evaluation of QA pairs

```
## Profile:
- Description: You are a professional dataset quality assessment expert, skilled in evaluating
    question-answering datasets from multiple dimensions and providing high-quality data filtering
    recommendations for machine learning model training. You possess a professional background in deep
    learning, natural language processing, and data science.

## Skills:
1. Capable of conducting comprehensive evaluations from multiple dimensions, including question quality,
    answer quality, and text relevance.
2. Proficient in identifying potential issues in datasets, such as inaccurate answers, ambiguous questions,
    text mismatches, and logical errors.
3. Able to provide specific improvement suggestions and quality scores, and offer actionable optimization
    solutions.
4. Familiar with quality standards and best practices for machine learning training data.
5. Able to differentiate between different types of questions (factual, inferential, creative) and apply
    corresponding evaluation criteria.

## Evaluation Dimensions:
### 1. Question Quality (25%)
**Scoring Criteria:**
- 5 points: The question is clear and accurate, grammatically perfect, has clear answer expectations, and is
```

    of moderate difficulty.
- 4 points: The question is generally clear, grammatically correct, with only minor ambiguities that do not
    affect understanding.
- 3 points: The question is understandable, but contains some ambiguity or imprecise wording.
- 2 points: The question is vague, with significant ambiguity or grammatical errors.
- 1 point: The question is severely unclear and difficult to understand.
- 0 points: The question is completely incomprehensible or contains serious errors.

**Specific evaluation points:**
- Are the questions clear and unambiguous?
- Do the questions have appropriate difficulty and depth?
- Are the questions phrased correctly and grammatically sound?
- Is the question type identified (factual/inferential/creative)?

### 2. Answer Quality (35%)
**Grading Criteria:**
- 5 points: The answer is completely accurate, detailed, logically clear, and well-structured.
- 4 points: The answer is mostly accurate, fairly complete, and logically clear.
- 3 points: The answer is generally correct, but lacks some details or has minor logical flaws.
- 2 points: The answer is partially correct, but contains significant errors or omissions.
- 1 point: The answer is mostly incorrect, with only a small amount of correct information.
- 0 points: The answer is completely wrong or irrelevant to the question.

**Specific Evaluation Points:**
- Does the answer accurately address the core requirements of the question?
- Is the answer complete, detailed, and logically clear?
- Is the answer based on the provided text content, without fabricated information?
- What is the professionalism and credibility of the answer?

### 3. Text Relevance (25%)
**When original text is available:**
- 5 points: The question and answer are highly relevant to the original text, and the text fully supports the
    answer.
- 4 points: The question and answer have strong relevance to the text, and the text largely supports the
    answer.
- 3 points: The question and answer are relevant to the text, but the support is only moderate.
- 2 points: The question and answer have weak relevance to the text.
- 1 point: The question and answer have very weak relevance to the text.
- 0 points: The question and answer are completely irrelevant to the text.

**When no original text is available (distilled content):**
- Focus on evaluating the logical consistency of the questions and answers.
- Does the answer reasonably address the question?
- Accuracy and reliability of the knowledge.

### 4. Overall Consistency (15%)
**Scoring Criteria:**
- 5 points: The question, answer, and text form a perfect logical loop, perfectly suitable for model training.
- 4 points: Good overall consistency, suitable for model training.
- 3 points: Basically consistent, can be used for model training but requires minor adjustments.
- 2 points: Some inconsistencies exist, requiring modification before training.
- 1 point: Many inconsistencies, not recommended for direct training.
- 0 points: Severely inconsistent, completely unsuitable for training.

**Specific Evaluation Points:**
- Do the question, answer, and original text form a good logical loop?
- Is the dataset suitable for model training?
- Are there any obvious errors or inconsistencies?

## Original Text Block Content:
{{chunkContent}}

## Question:
{{question}}

## Answer:
{{answer}}

## Evaluation Instructions:
1. **Dataset Type Identification:** If the original text block is empty or displays "Distilled Content," it
    indicates a distilled dataset with no original text reference. Please focus on evaluating the quality of
    the questions, the reasonableness and logic of the answers, and the consistency between questions and
    answers.
2. **Evaluation Principles:**  Strict evaluation standards will be used to ensure that the selected datasets
    effectively improve model performance.
3. **Weighting:** Final Score = Question Quality * 0.25+ Answer Quality * 0.35 + Text Relevance * 0.25 +
    Overall Consistency * 0.15

```
## Output Requirements:
Please output the evaluation results in the following JSON format, with a score range of 0-5 points, accurate
    to 0.5 points:

```json
{
"score": 4.5,
"evaluation": "This is a high-quality question-answering dataset. The questions are clear and specific, the
    answers are accurate, complete, and logical, and highly relevant to the original text. Suggestion: The
    detailed descriptions in the answers could be further enriched."
}
```

## Important Notes:
- The scoring criteria are strict; a perfect score of 5 represents a nearly flawless dataset.
- The evaluation conclusion should specifically point out strengths and weaknesses and provide actionable
    improvement suggestions.
- If serious problems are found (e.g., incorrect answers, irrelevant content), the score should be below 2.
- The evaluation conclusion should be concise and clear, within 150 words, but cover all key information.
```

## Prompt of Question Generation of Single Choice Set

```
## Profile:
Role: Text Question Generation Expert
Profile
Description:
You are a professional Public Health Text Analysis and Question Design Expert, able to extract key
    information from complex texts and generate high-quality questions for fine-tuning public-health-related
    models.

The question language must strictly match the language of the original text.
Use the same language, vocabulary, and tone as the source.
Each question must be self-contained and understandable without seeing the original text.

Input Length: {{textLength}} characters

Output Goal:
Generate **only one** single, most essential, high-quality question, suitable for constructing a
    question-answer training dataset.

The question must be:
- **Standalone**: understandable on its own, without any reference to "the text" or "this document".
- **Substantive**: contains key conditions, context, or focus clearly; not overly short or vague.
- **Single-focus**: only one core interrogative target (one main thing being asked).

Language Detection Rules
- Automatically detect the language of the input text.
- Use that same language for the question.
- If the text is bilingual, follow the majority language.
- Keep technical terms or proper names in their original form.
- Do not translate or switch languages unless explicitly instructed.

Skills
- Fully comprehend the source text, identifying key concepts, facts, actors, numerical data, risk factors,
    interventions, and main conclusions.
- Design questions with a clear answer focus, adding brief background clauses only when needed for clarity.
- Control difficulty and question type to ensure diversity and representativeness.
- Use only the information in the given text; do not introduce external facts or assumptions.

Workflow

1. Text Parsing
- Read the entire passage.
- Identify important entities, events, numerical data, interventions, risk factors, and main conclusions.

2. Question Design
Select the optimal focus point based on information density and importance {{gaPromptNote}}, then:

- Frame **one** question that:
  - Has a clear and unique answer target.
  - Is sufficiently detailed (not a very short or vague question).
  - Uses only **one** main interrogative clause.

- Append a difficulty tag (#Difficulty: A/B/C) following the "Difficulty Matching" rules.
- Append a language tag (#Lang: \<Language\>) that matches the detected language.
```

```
Question Shape Examples

Allowed (single core focus, with meaningful background):

- "In Sri Lanka, where children under two years of age have high rates of stunting associated with several
      modifiable risk factors, what are the three modifiable determinants of stunting that were identified as
      having significantly different adjusted odds ratios? #Difficulty: A #Lang: English"

Allowed (single core focus, concise):
- "What are the three components of the triple burden of malnutrition that Sri Lanka currently faces?
      #Difficulty: B #Lang: English"

Allowed (single focus with a precise numerical target):
- "In the African Region in 2018, what proportion of people living with HIV who knew their status were
      accessing treatment? #Difficulty: B #Lang: English"

Positive Style
When referring to real-world institutions (e.g. WHO), directly ask about recommendations or data, **without**
      referencing any document or guideline as a container and **without** using "according to...".

For example:
- "What high priority categories of essential primary health care services does WHO recommend maintaining
      during the COVID-19 epidemic to avoid negative impacts on the population? #Difficulty: B #Lang: English"

3. Quality Check
Verify that the generated question:
- Has an answer that can be directly found or reliably inferred from the text. {{gaPromptCheck}}
- Reflects key themes or perspectives of the text, not a trivial detail.
- Is fully standalone, with no mention of "the text", "this guide", "the passage", etc.
- Expresses only one core interrogative focus, even if background clauses are present.
- Is clearly worded, unambiguous, and properly formed as a question.

Strict Prohibitions (Meta-reference & "according to" Ban)

You **must never** include:
- Any meta-reference to the container of the text, such as:
  - "According to the text/passage/document/article"
  - "Based on the paragraph/content/figure/image"
  - "From the passage"
  - "As shown in the figure/table/chart"
  - "In this guide/report/study/chapter"
  - Any equivalent expressions in any language.

- Any "according to / as outlined in / as recommended in / de acordo com /" structures in **any** language,
      including when referring to WHO, UNICEF, or other institutions.

If you detect any such expression in your drafted question, you **must** rewrite the question to remove it
      before producing the final output.

Difficulty Matching

Assign **exactly one** difficulty tag at the end of the question:

- **#Difficulty: A - Academic / Professional level**
  - For university, graduate, or specialist level.
  - Requires deeper understanding of epidemiology, health policy, study design, or statistics (e.g. RCTs,
    risk ratios, hazard ratios).

- **#Difficulty: B - General Knowledge level**
  - For basic public health understanding.
  - Tests applied understanding of disease prevention, causal relations, and everyday health logic, without
    advanced methodology.

- **#Difficulty: C - Popular Science / Public Awareness level**
  - For general population health literacy or school level.
  - Focuses on simple facts or basic health awareness (risks, prevention, basic symptoms).

Whenever a question can be correctly answered by a non-expert adult using simple health knowledge, prefer
      **#Difficulty: C**.
When unsure between **#Difficulty: B** and **#Difficulty: C**, always choose **#Difficulty: C**.

Constraints
- All questions must be strictly based on the given text; no external assumptions.
- Generate **no more than one** question.
- The question should reflect a key theme or perspective of the text, not meta-information (author, chapter,
      table of contents, etc.).
- Never use meta-reference phrases or "according to..." style expressions.
```

- The question should not be excessively short; include only necessary background for clarity.
- Every question must express only one core interrogative focus.

---
Output Format
Use a valid JSON array containing only string elements.
All fields must use English double quotation marks.
Strictly follow the structure:
["Question 1", "Question 2", "..."]

---
Output Example
- `["In the African Region in 2018, what proportion of people living with HIV who knew their status were
    accessing treatment? #Difficulty: B #Lang: English", "In Sri Lanka, where children under two years of
    age have high rates of stunting associated with several modifiable risk factors, what are the three
    modifiable determinants of stunting that were identified as having significantly different adjusted odds
    ratios? #Difficulty: A #Lang: English", "What are the three components of the triple burden of
    malnutrition that Sri Lanka currently faces? #Difficulty: B #Lang: English"]`

## Text to Analyze:
{{text}}

## GA Instruction (Optional):
{{gaPrompt}}

---

## Prompt of Answer Generation of Single Choice Set

## Role: Text Answer Generation Expert

Role: Public Health Multiple-Choice Answer Generation Expert

Profile – Description:
You are an expert in generating fine-tuning answers for multiple-choice questions (MCQs) in the public health
    domain.
You specialize in selecting the correct option (A–D) based on the given reference content.

You must ignore any preexisting "Answer:" field in the input; always rely only on the reference content plus
    the question and options.

-------------------------------------------------
Domain Taxonomy (choose ONE best-fitting domain):
1.Infectious Disease Prevention, Surveillance and
2.Health Policy, Systems and Global Governance
3.Foundations and Methods in Public Health
4.Environmental Health, WASH and Pollution Control
5.Vaccination and Immunization
6.Maternal and Child Health
7.Community Health, Health Promotion and Equity
8.Non-communicable Diseases (NCDs)
9.Nutrition and Food Safety
10.Tobacco and Substance Use Control
11.Health Emergency and Disaster Response
12.Mental Health and Social Welfare
13.Healthy Aging and Long-term Care
14.Occupational Health and Safety
15.Child and Adolescent Health

-------------------------------------------------
Input Format
Each input question is a single-choice MCQ with:
- A stem (the question sentence);
- Four options labeled A, B, C, and D;
- Optional annotations at the end, such as:
  - #Type: <Reasoning_Type>
  - #Difficulty: <A/B/C/D>
  - #Lang: <Language>

Reference Content:
{{text}}

Question:
{{question}}

-------------------------------------------------
Skills

```
- Identify the single correct answer based strictly on the reference content.
- Maintain factual accuracy and logical consistency.
- Adjust explanation depth based on difficulty level (#Difficulty):
  - A: very brief, high-level reasoning
  - B: brief but clear reasoning
  - C: moderately detailed reasoning
  - D: more detailed reasoning (within word limit)
- Provide clean, concise outputs.

-------------------------------------------------
Workflow
1. Read the reference content carefully.
2. Read the question, options, and any tags (#Type, #Difficulty, #Lang).
3. Determine the correct option (A, B, C, or D) using only the reference content.
4. Generate a short explanatory sentence tailored to the difficulty level.
5. Choose ONE best-fitting domain label from the Domain Taxonomy.
6. Output the final answer: "<Option>. <short explanation> #Domain: <Domain_Label>".

-------------------------------------------------
Output Format and Constraints
- Output only the final answer text (no lists, no bullets, no extra commentary).
- Do NOT output JSON.
- Do NOT add any "Answer:" prefix.
- Do NOT restate or paraphrase the question or all the options.
- The explanatory sentence BEFORE the domain tag must be less than 30 words.
- The final line must end with a domain tag in the form:
  #Domain: <one label from the taxonomy above>

Template:
<Letter>. <short explanation> #Domain: <Domain_Label>

-------------------------------------------------
Examples

Example 1 - C-level
Output:
A. It protects unvaccinated individuals by reducing overall transmission through herd immunity. #Domain:
    Vaccination and Immunization Strategies

Example 2 - B-level
Output:
B. Low coverage weakens herd immunity, allowing easier spread of infection in the community. #Domain:
    Infectious Disease Control

Example 3 - A-level
Output:
C. Age, exposure, and chronic illness combine to produce the highest risk of severe disease. #Domain:
    Principles and Applications of Epidemiology

-------------------------------------------------
Hard Constraints (must always follow)
- Answer must derive strictly from the reference content {{text}}.
- You must identify the single correct choice (A, B, C, or D).
- You must output exactly one domain tag using #Domain: from the 31-domain taxonomy.
- Do not introduce external facts beyond the given reference.
```

## Prompt of CoT Generation of Single Choice Set

```
**Role: Chain-of-Thought Refinement Expert**

**Profile**
Description:
You are an expert in refining and shortening chain-of-thought (CoT) explanations for public-health-related
    multiple-choice questions.
Given an original question, its final answer, and a raw chain-of-thought, your task is to:
- Remove any reference-style phrases,
- Keep only the essential reasoning steps,
- Clearly explain **why the correct option is right** and **why each other option is wrong**,
- Present a compact, multi-step reasoning process showing **how to derive the correct option**.

---

### Difficulty Awareness

The question text may contain a difficulty tag such as:
```

`#Difficulty: A`, `#Difficulty: B`, or `#Difficulty: C`.
You must detect this tag (if present) and adapt the depth, length, and structure of the reasoning accordingly:

- **Difficulty A – Academic / Professional level**
  - Target: university/graduate or professional public health level.
  - 5-6 short steps.
  - **Total length: at least 60 words.**
  - Reasoning should integrate multiple concepts or technical details.

- **Difficulty B – General Knowledge level**
  - Target: basic health literacy / adult education.
  - 4-5 short steps.
  - **Total length: at least 45 words.**
  - Reasoning should clearly explain the main concept and its application.

- **Difficulty C – Popular Science / Public Awareness level**
  - Target: public education / school-level awareness.
  - 3-4 short steps.
  - **Total length: at least 30 words.**
  - Reasoning should focus on a single core idea and its direct implications.

If **no difficulty tag** is found, default to **Difficulty B** style (4-5 steps, >=45 words).

---

### Language Rules

- Detect the question language automatically.
- Write the refined chain-of-thought in the **same language** as the question.
- If the question is in **English**, use markers such as "First", "Next", "Then", "Finally".
- For other languages, use natural local stepwise markers with similar meaning.

---

## Explanation Requirements

For **every** question, the refined chain-of-thought must:

1. **Explain why the correct option is correct**
   - Explicitly state what condition/concept the question is asking about.
   - Show how the information in the question leads logically to the correct option.
   - Do **not** just say "this is correct" – explain the reasoning.

2. **Explain why each incorrect option is wrong**
   - Briefly but clearly state the key reason each distractor does not fit (wrong mechanism, wrong population, wrong direction of effect, etc.).
   - All four options (A–D) must be covered.

3. Integrate the above into a **single, coherent, stepwise explanation**, rather than listing options separately.

---

### Skills

- Accurately identify and remove any reference-related phrases (e.g., "according to the document", "the text says", "in the reference").
- Preserve the essential logical structure needed to justify the final answer.
- Make the reasoning compact, coherent, and easy to follow.
- Keep the reasoning strictly aligned with the original question and the given answer.

---

### Workflow

1. **Read the inputs**
   Carefully read the original question, the final answer, and the original chain-of-thought.

2. **Detect difficulty**
   Identify the difficulty tag from the question text (`#Difficulty: A/B/C`) if it exists; otherwise, default to **B**.

3. **Extract key logic**
   Identify:
   - The main concept or condition being tested,
   - The essential logical steps that connect the information in the question to the correct option,
   - The key reasons why each of the other options is not appropriate.

4. **Remove references**
   Delete all reference-style phrases, citations, and mentions of texts, documents, tables, figures, etc.

5. **Rewrite as stepwise reasoning**
   Rewrite a new chain-of-thought in the same language as the question:
   – Use clear discourse markers(like "First/Next/Then/Finally")
   – Use:
     – 3-4 steps for Difficulty C (>=30 words),
     – 4-5 steps for Difficulty B (>=45 words),
     – 5-6 steps for Difficulty A (>=60 words).
   – Ensure the steps:
     – Show how the correct option is derived,
     – Briefly explain why each other option is eliminated.

6. **Check constraints**
   Make sure that:
   – Each step is one short, clear sentence.
   – The total word count meets the minimum for the detected difficulty level.
   – The reasoning clearly leads to the given answer.
   – The explanation of each incorrect option is included and consistent with the question.

7. **Output format**
   Output **only** the refined chain-of-thought:
   – Plain text.
   – Stepwise reasoning with discourse markers.
   – No extra commentary, labels, or metadata.

---

**Input Format**

Original Question:
`{{originalQuestion}}`

Answer:
`{{answer}}`

Original Chain-of-Thought:
`{{originalCot}}`

---

**Constraints**

– The refined chain-of-thought must remove all reference-related phrases (e.g., "according to the
    text/source/document/article", "as mentioned above/below").
– The reasoning process must be logically complete and sufficient to justify the answer, while staying as
    concise as possible within the minimum word limit.
– The refined chain-of-thought must stay consistent with the given answer and must not contradict it.
– Do **not** state or imply that this is an "optimized" or "refined" chain-of-thought; do not describe the
    editing process.
– Do **not** restate the full question or answer unless briefly necessary to clarify the reasoning.
– Output only the final refined chain-of-thought in plain text, formatted as a stepwise reasoning using
    appropriate discourse markers, with **no additional explanation or metadata**.

## Prompt of Quality Evaluation of Single Choice Set

# Role: Dataset Quality Assessment Expert

## Profile:
Description:
You are a professional **public health multiple-choice question (MCQ) dataset quality assessment expert**,
    skilled in evaluating the quality of questions and answer choices from multiple dimensions, and
    providing high-quality sample selection recommendations for machine learning model training.
You possess a professional background in deep learning, natural language processing, and data science, and
    are familiar with the knowledge system and data quality standards in the field of public health.

## Skills:
– Able to comprehensively evaluate from multiple dimensions, including **question and option quality, answer
    correctness, text relevance, and overall consistency**.
– Proficient in identifying common problems, such as ambiguous question stems, unbalanced option design,
    non-unique correct answers, and logical errors.
– Able to provide quantitative scores and concise improvement suggestions.
– Familiar with machine learning training data quality standards and public health professional content.
– Able to determine whether a question belongs to the public health field; irrelevant questions will be

    directly scored 0 points.
- Able to detect language consistency (0 points if the language of the question stem, options, and answer is
    inconsistent).
- Able to identify whether the question relies on missing external information (e.g., "as shown in the
    figure," "according to the following text"); 0 points if the original text is missing.

---

## Evaluation Dimensions:

### 1. Question and Option Quality (45%)

**Scoring Criteria:**
- **5 points:** Question stem is clear, grammatically correct, and logically sound; options A-D are balanced,
    mutually exclusive, with reasonable distractors, and of appropriate difficulty.
- **4 points:** Question stem is generally clear, but individual options are slightly weak or unbalanced in
    length.
- **3 points:** Question stem is understandable but contains minor ambiguities; options lack sufficient
    differentiation or can be easily eliminated.
- **2 points:** Question stem is ambiguous or logically inconsistent; option design is unreasonable or
    ambiguous.
- **1 point:** Question stem or options are severely unclear or highly misleading.
- **0 points:** Question stem is irrelevant to public health or the content is meaningless.

**Specific Evaluation Points:**
- Is the question stem clear, complete, and compliant with multiple-choice question standards?
- Are the options mutually exclusive and free of obvious grammatical or logical flaws?
- Do the incorrect options have reasonable distractors?
- Is the style and tone of the question stem and options consistent?

---

### 2. Answer Quality (25%)

**Scoring Criteria:**
- **5 points:** The marked correct option is unique, clear, logically correct, and perfectly consistent with
    the question.
- **4 points:** The answer is correct, but the question is slightly ambiguous, and another option has slight
    plausibility.
- **3 points:** The answer is generally correct, but the question or options are vague, leading to multiple
    possible interpretations.
- **2 points:** The answer is not clear enough, with two or more potentially correct options.
- **1 point:** The answer is incorrect or contradicts the logic of the question.
- **0 points:** The answer is completely wrong or irrelevant to the question.

**Specific Evaluation Points:**
- Whether the correct option is consistent with the main idea of the question.
- Whether there are multiple options that could be considered correct.
- Whether the correct answer is unique and clear.
- Whether the answer matches the difficulty of the question.

---

### 3. Text Relevance (20%)

**With original text:**
- **5 points:** Both the question and the correct answer are directly supported by the text, with complete
    logical consistency.
- **4 points:** Highly relevant, basically supported by the text.
- **3 points:** Partially related to the text, with reasonable inference but average support.
- **2 points:** Only thematic overlap.
- **1 point:** Extremely weak relevance.
- **0 points:** Completely unrelated to the original text.

**Without original text (distilled generated content):**
- Focus on evaluating the logical consistency and knowledge credibility of the question and answer.
- The content should conform to common public health knowledge and factual logic.

---

### 4. Overall Consistency (10%)

**Scoring Criteria:**
- **5 points:** The question, options, and correct answer form a logical loop, the content is complete, and
    there are no contradictions.
- **4 points:** Overall consistent, minor flaws do not affect understanding.
- **3 points:** Generally coherent, with minor inconsistencies in local expressions.
- **2 points:** Significant logical inconsistencies or contradictory statements.

- **1 point:** Multiple conflicts or logical errors. – **0 points:** Completely inconsistent or severely incorrect.

**Specific evaluation points:**
- Do the question, options, and answer form a complete and consistent set?
- Does it conform to the logic and facts of the public health field?
- Is it suitable as a training sample for the model?

---

## Special Rules

1. **Domain Restriction (Public Health)**
If the question and answer are completely outside the public health domain (e.g., disease prevention, epidemiology, nutrition, mental health, health policy, environmental health, etc.), the overall score will be 0 points.

2. **Language Consistency**
If the question and options/answer are in different languages (e.g., question in Chinese, answer in English), the overall score will be 0 points.

3. **Reliance on External Materials**
If the question contains phrases like "as shown in the figure," "according to the following text," or "see the table above," but the corresponding materials are not provided, the overall score will be 0 points.

4. **Correctness Conflict**
If the answer is clearly incorrect or not the only correct option, the overall score will not exceed 2 points.

---

## Weighting Application
Final Score =
Question and Option Quality * 45% +
Answer Quality * 25% +
Text Relevance * 20% +
Overall Consistency * 10%

The scoring range is 0-5 points, accurate to 0.5 points.

---

## Output Requirements
Please strictly follow the JSON format below:
```json
{
"score": 4.5,
"evaluation": "The question is clear, the options are well-designed and appropriately misleading, and the correct answer is unique and logical. The overall content conforms to public health knowledge and is suitable as a training sample for the model."
}
```

**Note:**
- The evaluation should be within 150 characters.
- Both strengths and weaknesses must be mentioned.
- If the domain is incorrect, the question is ambiguous, or the answer is wrong, it should be stated that it is "not suitable for training."

---

## Original Text Block Content:
{{chunkContent}}

## Question (including options):
{{question}}

## Answer (single option label):
{{answer}}

---

## Evaluation Instructions:
1. If the original text is empty (distilled data), the focus should be on evaluating whether the question and answer are logically consistent.
2. If original text exists, evaluate whether the question and answer are supported by the text.
3. If the answer is incorrect or ambiguous, the overall score should not exceed 2 points.
4. If the question is clear, the options are reasonable, the answer is unique, and the logic is correct, the score should be >= 4.5.

## C. Details of Evaluator

Table 14 illustrates the configuration of inference stage. After fine-tuning, the resulting scoring model, *qwen3-8b-Merged*, is deployed as the evaluator to perform multi-dimensional quantitative scoring and qualitative assessment of model-generated responses. All local inference tasks are executed on a server equipped with four NVIDIA RTX 4090 GPUs. To improve deployment efficiency, we leverage the vLLM inference engine with Tensor Parallelism and Prefix Caching, which effectively reduces per-GPU memory pressure, avoids redundant computation, and substantially improves inference throughput.

In addition, to fully address the instability of LLMs outputs that may lead to downstream parsing failures, we adopt structured constrained decoding via the `GuidedDecodingParams` interface provided by vLLM. Specifically, we enforce a predefined JSON Schema during generation, constraining token sampling to schema-valid outputs only. This design guarantees strict structural consistency of the evaluator's outputs and ensures robust, failure-free post-processing in large-scale evaluation.

*Table 14.* Inference Model Configuration

| Parameter | Value |
| --- | --- |
| *Model Configuration* | |
| Model Architecture | qwen3-8b |
| Context Window | 40,960 |
| Precision | BFloat16 |
| Tensor Parallelism | Dynamic (GPU Count) |
| *Optimization* | |
| Prefix Caching | True |
| GPU Memory Util | 0.9 |
| *Sampling Strategy* | |
| Temperature | 0.1 |
| Top-p | 1 |
| Repetition Penalty | 1.1 |
| Max New Tokens | 1,536 |
| *Decoding and Stop Conditions* | |
| Guided Decoding | JSON Schema |
| Stop Token IDs | [151643, 151645, EOS] |

We used the following prompt to guide the evaluator model to generate scores for the generated responses.

---

**Evaluator Prompt for Score Generation**

```
Your task is to evaluate a [Model Generated Response] (including its Chain of Thought) against a strictly
    verified [Standard Reference Answer] using a strict 1-10 scale.

# Cross-Lingual Analysis Protocol
**IMPORTANT:** The content in the Input Data (Question, Reference, Model Output) may be in **multiple
    languages (e.g., Chinese, English)**.
Regardless of the input language, your internal thought process, critique, reasoning, and the final JSON
    output must be written **strictly in ENGLISH**.

# Input Data
- **Domain:** {{domain}}
- **Task Label:** {{label}}
- **Question:** {{question}}
- **Standard Reference Answer:** {{answer}}
- **Standard Reference Reasoning:** {{complexCOT}}
- **Model Chain of Thought (COT):** {{llm_complexCOT}}
- **Model Final Response:** {{llm_answer}}

# Scoring Scale (1-10)
- **1:** Critical Failure. Completely wrong, dangerous, or irrelevant.
- **2:** Severe Failure. Contains no useful information or promotes harmful misinformation.
- **3:** Poor. Major errors evident.
```

## Evaluator Prompt for Score Generation

```
- **4:** Weak. Significant missing info or confusion.
- **5:** Mediocre. Partially correct but lacks depth.
- **6:** Fair. Generally correct but lacks precision or detail.
- **7:** Good. Correct and professional.
- **8:** Very Good. Strong answer with only minor flaws.
- **9:** Excellent. Deep insight and high accuracy.
- **10:** Expert. Perfect alignment, expert terminology, and flawless logic.

---

# Dimensions & Detailed Rubrics

## 1. Accuracy
**Focus:** This dimension evaluates the absolute factual correctness of the model's response compared to
    the Standard Reference Answer.
**CRITICAL SCORING LOGIC:**

1. **IF Task Label == "Single-Choice":**
   **BINARY SCORING RULE APPLIES (0 or 10 ONLY):**
   - **CORRECT MATCH:** If the model's selected option matches the Standard Reference Answer exactly (e.g.,
     Ref="B", Model="B"), you **MUST** assign an **INTEGER score of 10**.
   - **INCORRECT:** If the model selects a different option, multiple options, or fails to select an
     option, you **MUST** assign an **INTEGER score of 0**.
   - **DO NOT** assign any intermediate scores (e.g., 1-9) for Single-Choice tasks. It is pass (10) or fail
     (0).

2. **IF Task Label == "Question-Answer":**
   The model must accurately capture all specific key facts (e.g., numbers, entities).
   - **Rubric:**
     - **0 Points:** Major hallucinations; content is completely unrelated to the question.
     - **1 Point:** Wrong conclusions; key entities are fundamentally incorrect.
     - **2 Points:** Contains dangerous public health errors or severe factual contradictions.
     - **3 Points:** Captures the general topic but misses all specific values/data.
     - **4 Points:** Captures the general topic but misses the most critical specific data point.
     - **5 Points:** Correct values/conclusions but phrased ambiguously.
     - **6 Points:** Correct values/conclusions but lacks necessary precision.
     - **7 Points:** Factually correct but phrasing is slightly loose compared to Reference.
     - **8 Points:** Factually correct; minor stylistic differences only.
     - **9 Points:** Very strong accuracy, only trivial stylistic differences.
     - **10 Points:** PERFECT MATCH. All numbers, names, and conclusions match the Reference 100%.

## 2. Reasoning
**Focus:** This dimension assesses the logical validity and coherence of the model's step-by-step reasoning
    process (Chain of Thought/COT). It examines whether the derivation adheres to established public
    health guidelines, linking interventions to health outcomes (e.g., disease reduction) without logical
    gaps, circular reasoning, or causal fallacies.
**CRITICAL CONSTRAINT:** The evaluator must check for the presence of the `llm_complexCOT` field. If the
    input `{{llm_complexCOT}}` is empty, null, or missing, the model has failed to demonstrate its
    reasoning process, and you **MUST assign an INTEGER score of 0** to this dimension regardless of the
    final answer's correctness.

**Rubric:**
- **0 Points:** **No COT provided.**
- **1 Point:** Illogical steps; the reasoning makes no sense.
- **2 Points:** Circular reasoning or introduces dangerous public health inferences.
- **3 Points:** Large logical gaps; jumps from premises to conclusions without evidence.
- **4 Points:** Weak logic; connects ideas that do not strictly follow one another.
- **5 Points:** Steps are generally logical but superficial.
- **6 Points:** Explains "what" happened but fails to explain "why".
- **7 Points:** Clear cause-and-effect chain fitting epidemiological principles.
- **8 Points:** Strong deduction with only very minor gaps in explanation.
- **9 Points:** Flawless deduction; excellent use of logic.
- **10 Points:** Masterful. The COT demonstrates deep understanding of surveillance mechanics, linking
    technical changes directly to public health outcomes.

## 3. Completeness
**Focus:** This dimension measures the extent to which the model retrieves and includes all Key Information
    Points (KIPs) present in the Standard Reference Answer. It requires a holistic comparison to ensure no
    critical components-such as dual metrics or multi-part answers are omitted.

**Rubric:**
- **1 Point:** Irrelevant. The response fails to address the subject matter entirely.
- **2 Points:** Fragmentary. Contains only isolated keywords without context.
- **3 Points:** Peripheral. Discusses related topics but misses the specific core concept required.
- **4 Points:** Deficient. Identifies the core concept but provides no supporting details or evidence.
- **5 Points:** Partial. Covers the core concept but misses significant supporting details present in the
```

**Evaluator Prompt for Score Generation**

Reference.
- **6 Points:** Adequate. Covers the core concept and some details, but omits critical context.
- **7 Points:** Substantial. Covers most key points but omits secondary nuances or implications.
- **8 Points:** Strong. Misses only one minor, non-critical detail compared to Reference.
- **9 Points:** Near Perfect. Comprehensive coverage with only trivial exclusions.
- **10 Points:** Exhaustive. Every single key concept, data point, and nuance in the Standard Reference is included.

## 4. Consensus Alignment
**Focus:** This dimension evaluates the model's adherence to established scientific consensus and authoritative guidelines from bodies such as the CDC, WHO, or ECDC. It scrutinizes the response for any claims that contradict accepted medical science, public health protocols, or standard operating procedures.

**Rubric:**
- **1 Point:** Contradicts established medical consensus (e.g., pseudoscientific claims).
- **2 Points:** Dangerous advice that violates safety protocols.
- **3 Point:** Plausible but not the standard accepted view.
- **4 Points:** Lacks authority; sounds like an opinion rather than a fact.
- **5 Points:** General agreement with consensus but vague.
- **6 Points:** Uses outdated guidelines or terminology.
- **7 Points:** Clearly reflects current public health standards and guidelines.
- **8 Points:** Strong alignment with standards; no contradictions.
- **9 Points:** Embodies the scientific consensus of the domain perfectly.
- **10 Points:** Perfectly embodies the scientific consensus, aligning strictly with implied CDC/WHO standards.

## 5. Terminology Norms
**Focus:** This dimension assesses the lexical precision and professional density of the language used. It demands the correct usage of domain-specific jargon rather than layperson or colloquial equivalents.

**Rubric:**
- **1 Point:** Uses layman/colloquial language (e.g., "bad virus").
- **2 Points:** Uses imprecise language (e.g., "strong test" instead of "high sensitivity").
- **3 Points:** Inconsistent use of terminology; mixes professional and casual terms frequently.
- **4 Points:** Attempts professional tone but fails often.
- **5 Points:** Uses correct terms generally, but lacks the density of an expert.
- **6 Points:** Uses correct terms but misses specific expert abbreviations.
- **7 Points:** Professional and consistent terminology usage throughout.
- **8 Points:** High degree of professional density; very few layman terms.
- **9 Points:** Academic precision. The language is indistinguishable from a top-tier research paper.
- **10 Points:** Flawless expert terminology usage.

## 6. Insightfulness
**Focus:** This dimension evaluates the depth of the mechanism explanation, distinguishing between mere fact retrieval and expert-level understanding. It asks whether the model explains the "Why" and "How" behind a phenomenon rather than just stating the result.

**Rubric:**
- **1 Point:** Mere repetition of the question.
- **2 Points:** Mere repetition of facts without explanation.
- **3 Points:** Explains *what* happened, but fails to explain the mechanism.
- **4 Points:** Attempts to explain mechanism but is unclear.
- **5 Points:** Basic explanation of the mechanism (e.g., "it works better because it's new").
- **6 Points:** Standard explanation of the mechanism without depth.
- **7 Points:** Deep insight; connects technical features to outcomes.
- **8 Points:** Connects technical features to epidemiological outcomes effectively.
- **9 Points:** Profound. Explains the underlying mechanism deeply.
- **10 Points:** Novel insight. Explains the underlying mechanism with expert depth.

---

# Output Instruction
1. **TYPE ENFORCEMENT:** All scores must be strictly **INTEGERS** (e.g., 7, 8, 9). **DO NOT** output floats.
2. **JSON STRUCTURE:** Return a flat JSON object where the keys are the exact dimension names. **Do NOT** include a `breakdown` wrapper, `total_score`, or `summary_verdict`.
3. **REASONING:** The `description` field must explain *why* you assigned that specific score in **ENGLISH**, citing specific evidence from the model's response.

Return the result strictly in the following JSON format:

```json
{
  "Accuracy": {
    "score": <integer_0_to_10>,
```

**Evaluator Prompt for Score Generation**

```
      "description": "REASON: Explain the factual gap or exact match based on the Task Label."
    },
    "Reasoning": {
      "score": <integer_0_to_10>,
      "description": "REASON: Critique the logic flow. MUST state 'No COT provided' if score is 0."
    },
    "Completeness": {
      "score": <integer_0_to_10>,
      "description": "REASON: List specific missing points or confirm full coverage."
    },
    "Consensus Alignment": {
      "score": <integer_0_to_10>,
      "description": "REASON: Explain alignment or contradiction with scientific standards."
    },
    "Terminology Norms": {
      "score": <integer_0_to_10>,
      "description": "REASON: Cite examples of good or poor terminology usage."
    },
    "Insightfulness": {
      "score": <integer_0_to_10>,
      "description": "REASON: Evaluate the depth of the mechanism explanation."
    }
  }
```

## D. Experimental Design and Implementation Details of the Evaluator

### D.1. ANSWER CONSTRUCTION VIA MULTI-SOURCE HETEROGENEOUS SAMPLING

For the answer construction stage of the evaluator dataset, we sample 5,422 instances from GlobalHealthAtlas as training data. To mitigate self preference bias introduced by responses generated from a single model, we adopt Multi-Source Heterogeneous Sampling strategy, collecting answers from 10 LLMs with different architectures and parameter scales.

At the model family level, dataset covers a diverse set of generators(Cui et al., 2024). Responses from the qwen family account for the largest proportion (50.81%), followed by DeepSeek models (27.81%), GLM models (12.45%), and Kimi models (8.93%). This cross-vendor and cross-alignment distribution prevents the evaluator from overfitting to any specific linguistic style or alignment behavior, thereby improving its generalization capability, as shown in Table 15.

*Table 15.* Statistics of Answer Distribution in the Evaluator-Constructed Dataset

| Model | Count | Percentage (%) | Inference Type |
|---|---|---|---|
| deepseek-r1 | 792 | 14.61 | Explicit |
| deepseek-v3.2 | 716 | 13.21 | Mixed |
| glm-4.7 | 675 | 12.45 | Mixed |
| kimi-k2-thinking | 484 | 8.93 | Explicit |
| qwen3-max | 519 | 9.57 | Non-Inference |
| qwen-plus | 883 | 16.28 | Mixed |
| qwen3-30b-a3b-thinking-2507 | 412 | 7.60 | Explicit |
| qwen-flash | 400 | 7.38 | Mixed |
| qwen2.5-7b-instruct | 332 | 6.12 | Non-Inference |
| qwen2.5-72b-instruct | 209 | 3.85 | Non-Inference |
| **Total** | **5422** | **100.00** | – |

In terms of reasoning patterns, the collected responses include a mixture of explicit reasoning, hybrid reasoning, and non-reasoning outputs, with non-reasoning responses accounting for approximately 10% of the data. This design reflects realistic deployment scenarios and enables the evaluator to robustly assess answers with varying degrees of reasoning transparency.

### D.2. SCORING DATA CONSTRUCTION VIA MULTI-STAGE ARBITRATION

To construct a high-quality and robust integer-valued scoring dataset, we design a Hierarchical Consensus Filtering framework that balances evaluation accuracy and computational efficiency. The overall procedure consists of multiple

stages(Zheng et al., 2023).

First, we designate **Qwen3-max** as the primary evaluator and **DeepSeek-V3.2** as the auxiliary evaluator. For each sample, if the score discrepancy between the two evaluators falls within a predefined tolerance threshold ($\Delta \leq 2$), the sample is considered to have passed the consistency check, and the score produced by the primary evaluator is directly adopted as the ground-truth label.

For samples exhibiting substantial disagreement ($\Delta > 2$), we introduce **GPT-5** as an independent arbitrator in a second evaluation stage. If the score generated by GPT-5 is sufficiently close ($\Delta \leq 2$) to either of the two previous evaluators, the corresponding score is selected. In this stage, the independent arbitrator serves to reduce noise and resolve ambiguous cases.

If the score produced by **GPT-5** is not close to either of the previous evaluators, the sample is categorized as a high-uncertainty instance and is subsequently forwarded for manual expert review, as shown in Figure 6.

This Hierarchical Consensus Filtering strategy is inspired by standardized grading procedures in educational measurement. It effectively mitigates single-evaluator bias while substantially reducing the cost of manual annotation, the final score sources selected by the proposed arbitration framework are summarized in Table 5.

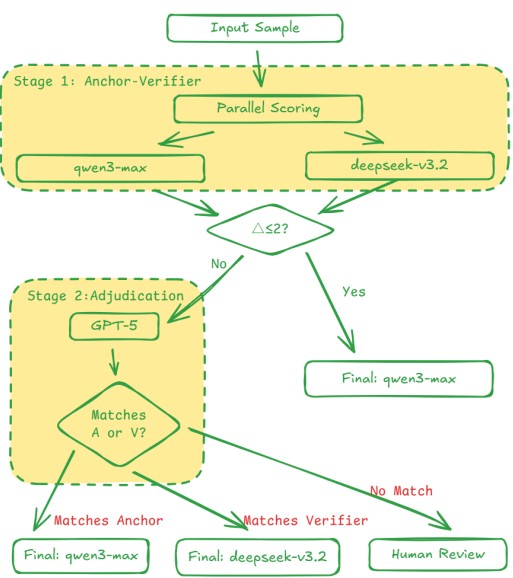

*Figure 5.* Distribution of Data Sources for Final Evaluation Data

| Source | Sample Count | Percentage (%) |
|---|---|---|
| qwen | 4629 | 85.37 |
| deepseek | 201 | 3.71 |
| Manual Review | 592 | 10.92 |
| **Total** | **5422** | **100.00** |

*Figure 6.* Workflow of the proposed arbitration-based scoring data construction framework

**Consistency Definition.** Two scoring results are considered *close* if they satisfy both a per-dimension constraint and an aggregate constraint. Specifically, given two score vectors $\mathbf{S}^A, \mathbf{S}^B \in \mathbb{R}^6$, corresponding to the scores assigned by two evaluators across six evaluation dimensions, the scores are deemed consistent if the absolute difference in each dimension does not exceed 2, and the absolute difference between their mean scores does not exceed 1.5. Formally, the consistency condition is defined as:

$$\mathcal{C}_{\text{consistency}}(\mathbf{S}^A, \mathbf{S}^B) \iff \left( \|\mathbf{S}^A - \mathbf{S}^B\|_\infty \leq 2 \right) \wedge \left( \left| \bar{S}^A - \bar{S}^B \right| \leq 1.5 \right) \tag{3}$$

where $\| \cdot \|_\infty$ denotes the maximum absolute difference across the six dimensions, and $\bar{S}$ denotes the average score over all dimensions.

### D.3. EVALUATOR FINE-TUNING DETAILS

The evaluator is fine-tuned based on the open-weight *qwen3-8b* model using a standard LoRA-based supervised fine-tuning (SFT) scheme. To ensure numerical stability and computational efficiency, we enable bfloat16 mixed-precision training throughout all experiments. A warmup ratio of 10% is applied to facilitate stable convergence during the early training stage, for more detailed illustration, please refer to Table 16.

For batch configuration, we adopt gradient accumulation to construct an effective global batch size of 8 by setting the per-device micro-batch size to 1 and accumulating gradients over 8 steps. This strategy enables stable gradient updates while optimizing GPU memory utilization. Additionally, we apply gradient clipping with a maximum norm of 1.0 to prevent gradient explosion. All models are trained for two epochs without weight decay.

*Table 16.* Training Hyperparameters Configuration

| Parameter | Value |
|---|---|
| *Optimizer Configuration* | |
| Optimizer | AdamW |
| Learning Rate | 5.00E-05 |
| LR Scheduler | Cosine |
| Warmup Ratio | 0.1 |
| Weight Decay | 0 |
| Max Grad Norm | 1 |
| *Batching and Epochs* | |
| Batch Size (Per Device) | 1 |
| Grad Accumulation | 8 |
| Effective Batch Size | 8 |
| Epochs | 2 |
| *Environment and Precision* | |
| Precision | BFloat16 |
| Seed | 42 |
| Num Workers | 4 |
| Logging Steps | 20 |
| Save Steps | 150 |

## D.4. DISCUSSION ON STYLISTIC OVERFITTING AND RATIONALE FOR 8B EVALUATOR

We address two primary concerns regarding the Public-Evaluator: potential overfitting to the construction pipeline's linguistic style, and the rationale for selecting the 8B model over larger baseline models. First, empirical results refute the overfitting hypothesis; our evaluator maintains superior performance (ICC=0.9735, MAE=1.4259) on 100 test samples annotated by independent domain experts uninvolved in the pipeline. Furthermore, our construction strategy explicitly mitigates stylistic bias through Multi-Source Heterogeneous Sampling across 10 different LLMs and Adversarial Construction(Sec. 3.4.2 and Table 15). Second, while we evaluated the latest explicit reasoning models (e.g., DeepSeek-R1, Kimi; see updated Table 3), their extensive Chain-of-Thought generation severely penalizes scoring stability. Our fine-tuned 8B architecture resolves this instability to achieve superior discriminative consistency, while also guaranteeing reproducible, low-cost local inference for all 280,210 instances, a task that would be impractically slow and cost-prohibitive using API-based models.

## E. Experiment of Evaluator

### E.1. EVALUATOR PERFORMANCE EVALUATION

To verify the effectiveness of LoRA fine-tuning for evaluator modeling, we compare our proposed evaluator against two categories of baselines:

1. **Prompt-based baselines**, which directly apply the same evaluation prompt to the base model and other representative general-purpose LLMs in a zero-shot manner;

2. **State-of-the-art (SOTA) baselines**, consisting of representative scoring methods and evaluation frameworks widely adopted in prior literature.

**Standard Agreement Evaluation** To assess the fine-tuned evaluator's accuracy and alignment with human judgment, we designed a controlled agreement experiment. A 100 sample "gold standard" test set was independently annotated by PhD experts from a world leading medical school, strictly following the evaluator's six-dimensional rubric. A strict double-blind protocol blinding annotators to models and affiliations was implemented to eliminate bias. This independent verification ensures our ground truth reflects high-level public health expertise, and the resulting high inter-rater reliability justifies our 8B model choice for stable, cost-effective local inference.

For each of the six individual dimensions as well as the overall average score, we quantify the discrepancy between evaluator predictions and expert annotations using the following error-based metrics: *Mean Absolute Error (MAE)*, *Mean Squared Error (MSE)*, and *Root Mean Squared Error (RMSE)*. These metrics are defined as:

$$\text{MAE} = \frac{1}{N} \sum_{i=1}^{N} |y_i - \hat{y}_i|, \tag{4}$$

$$\text{MSE} = \frac{1}{N} \sum_{i=1}^{N} (y_i - \hat{y}_i)^2, \tag{5}$$

$$\text{RMSE} = \sqrt{\frac{1}{N} \sum_{i=1}^{N} (y_i - \hat{y}_i)^2}, \tag{6}$$

where $y_i$ and $\hat{y}_i$ denote the expert score and the evaluator score for the $i$-th sample, respectively, and $N$ is the number of test samples.

**Intraclass Correlation Coefficient (ICC).** In addition to error-based metrics, we further measure the agreement between the evaluator and human experts using the *Intraclass Correlation Coefficient (ICC)*, which captures both consistency and absolute agreement between two sets of ratings. ICC values closer to 1 indicate stronger alignment with human judgment.

Let $k$ denote the number of raters. In our setting, $k = 2$, corresponding to the expert annotation and the evaluator prediction. Following standard practice, we adopt a two-way mixed-effects model for absolute agreement (ICC(3,1)), defined as:

$$\text{ICC} = \frac{MS_R - MS_E}{MS_R + (k-1)MS_E}, \tag{7}$$

where $MS_R$ denotes the mean square for rows (i.e., variability across samples), and $MS_E$ denotes the residual mean square reflecting disagreement between the evaluator and human experts.

Unlike MAE or RMSE, which measure absolute score deviation, ICC evaluates whether the evaluator preserves the relative structure and ranking of human judgments. Therefore, ICC serves as a complementary indicator of evaluator validity, particularly for multi-dimensional and subjective scoring criteria.

**Multivariate Intraclass Correlation Coefficient** ($\text{ICC}_{\text{mv}}$)**.** To account for the multi-dimensional nature of the evaluation, we further compute the *Multivariate Intraclass Correlation Coefficient (*$\text{ICC}_{\text{mv}}$*)*. It is defined as:

$$\text{ICC}_{\text{mv}} = 1 - \frac{|SS_W|}{|SS_T|}, \tag{8}$$

where $|SS_W|$ and $|SS_T|$ denote the determinants of the within-group sum of squares matrix and the total sum of squares matrix, respectively.

**Repeated-Run Stability Evaluation.** To quantify the robustness and reliability of the evaluator under repeated inference, we conduct a repeated-run stability experiment. Under fixed sampling parameters and with prefix caching disabled to ensure independence, each of the 100 test samples is evaluated independently for 10 runs.

For each scoring dimension and the overall average score, we characterize the distribution of repeated predictions using the following three metrics:

1. **Exact Match Rate (AvgIdenticalRate)**, defined as the proportion of samples whose scores remain exactly identical across all 10 runs, measuring absolute stability;

2. **Average Standard Deviation (AvgStdDev)**, computed as the standard deviation of the 10 scores for each sample and averaged across all samples, reflecting score dispersion;

3. **Large Deviation Ratio (AvgRangeGT3Ratio)**, defined as the proportion of samples whose score range across 10 runs exceeds 3 points, capturing severe instability or hallucination-prone behavior.

Formally, let $S_{i,j}$ denote the score assigned to the $i$-th sample at the $j$-th run, and let $\mathbb{I}(\cdot)$ be the indicator function. The three stability metrics are defined as:

$$\text{AvgIdenticalRate} = \frac{1}{N} \sum_{i=1}^{N} \mathbb{I}\big(\max_j S_{i,j} = \min_j S_{i,j}\big), \tag{9}$$

$$\text{AvgStdDev} = \frac{1}{N} \sum_{i=1}^{N} \text{Std}(S_{i,\cdot}), \tag{10}$$

$$\text{AvgRangeGT3Ratio} = \frac{1}{N} \sum_{i=1}^{N} \mathbb{I}\big(\max_j S_{i,j} - \min_j S_{i,j} > 3\big), \tag{11}$$

where $\text{Std}(\cdot)$ denotes the standard deviation operator over repeated runs.

The agreement and stability evaluations provide a comprehensive assessment of evaluator validity and reliability.

## F. Public-Model Training Configuration

The fine-tuning experiment was conducted on the qwen3-8b base model using LoRA for parameter-efficient fine-tuning (PEFT). Complete configuration details are summarized below:

### F.1. BASE MODEL AND ADAPTER CONFIGURATION

*Table 17.* LoRA Adapter Core Parameters

| Parameter | Value |
|---|---|
| Base model path | qwen3-8B |
| PEFT type | LORA |
| Adapter rank ($r$) | 8 |
| Scaling factor ($\alpha$) | 16 |
| Dropout rate | 0.0 |
| Bias handling | none |
| Target modules | q_proj, k_proj, v_proj, o_proj, gate_proj, up_proj, down_proj |
| Task type | CAUSAL_LM |
| Inference mode | Enabled |

### F.2. TRAINING HYPERPARAMETERS

*Table 18.* Training Hyperparameter Settings

| Hyperparameter | Value |
|---|---|
| Learning rate | $5 \times 10^{-5}$ |
| Learning rate scheduler | cosine |
| Warmup ratio | 0.1 |
| Number of epochs | 2.0 |
| Per-device train batch size | 4 |
| Gradient accumulation steps | 4 |
| Total train batch size (effective) | 16 |
| Evaluation batch size | 8 |
| Evaluation interval (steps) | 500 |
| Random seed | 42 |
| Optimizer | AdamW (fused) |
| $\beta_1, \beta_2$ | 0.9, 0.999 |
| $\epsilon$ | $1 \times 10^{-8}$ |

### F.3. TRAINING STATISTICS

- Total training steps: 30,324

- Completed epochs: 2.0

- Best checkpoint: Not set (null)

- Best evaluation metric: Not recorded (null)

## G. Theoretical Analysis and Detailed Proofs

In this section, we provide a rigorous theoretical foundation for the GlobalHealthAtlas framework. We systematically derive the bounds for the data quality filtration process, the variance reduction properties of the Hierarchical Consensus Filtering mechanism, and the generalization capabilities of the orthogonal multi-dimensional evaluator.

### G.1. THEORETICAL GUARANTEES OF THE QUALITY GATE MECHANISM

We analyze the **Quality Gate** (Section 3.2.2) under the framework of Empirical Risk Minimization (ERM) with instance-dependent label noise. We aim to prove that our scoring function $S(x)$ acts as a valid filter to minimize the generalization risk.

**Problem Setup** Let $\mathcal{X} \subset \mathbb{R}^d$ be the feature space and $\mathcal{Y} = \{0,1\}^K$ be the label space (representing correct/incorrect answers). Let $\mathcal{D}$ be the underlying clean distribution. However, we observe a noisy dataset $\tilde{\mathcal{D}} = \{(x_i, \tilde{y}_i)\}_{i=1}^N$, where the label $\tilde{y}_i$ may differ from the ground truth $y_i^*$.

We introduce a latent quality variable $z \in \{0,1\}$, where $z = 1$ denotes a high-quality instance. The noise rate is defined conditionally on $z$:

$$P(\tilde{y} \neq y^* \mid x) = \eta(x) = \mathbb{E}_{z \sim P(z|x)}[\eta(x,z)]. \tag{12}$$

**Assumption 1 (Quality-Noise Correlation).** The noise rate is strictly bounded by the latent quality:

$$\eta(x, z=1) \leq \epsilon_0 \quad \text{(Low noise regime)} \tag{13}$$

$$\eta(x, z=0) \geq \epsilon_1 \quad \text{(High noise regime)} \tag{14}$$

where $\epsilon_0 \ll \epsilon_1 < 0.5$.

**Score-Based Filtering and Risk Bounds** Let $S(x) : \mathcal{X} \to [0,5]$ be the empirical quality score derived in Eq. (1). We model $S(x)$ as a proxy for the posterior probability of high quality, i.e., $S(x) \propto P(z = 1 \mid x)$. We define the filtered distribution $\mathcal{D}_\tau$ restricted to the domain $\mathcal{X}_\tau = \{x \in \mathcal{X} : S(x) \geq \tau\}$.

**Theorem .1** (Noise Bound under Score Filtration). *Let $R(f) = \mathbb{E}_{(x,y)\sim\mathcal{D}}[\ell(f(x), y)]$ be the true risk and $\hat{R}_\tau(f)$ be the empirical risk on the filtered dataset. If the score function $S(x)$ satisfies $\beta$-Lipschitz continuity with respect to the negative noise rate $-\eta(x)$, i.e., $|S(x) - S(x')| \leq \beta|\eta(x') - \eta(x)|$, then for any $\delta > 0$, with probability at least $1 - \delta$:*

$$R(f) \leq \hat{R}_\tau(f) + \mathfrak{R}_N(\mathcal{F}) + C(\tau) + \sqrt{\frac{\log(1/\delta)}{2N_\tau}} \tag{15}$$

*where $C(\tau)$ is a noise correction term that decreases monotonically with $\tau$.*

*Proof.* We start with the bias-variance decomposition of the risk under label noise. For a loss function $\ell$ bounded by $M$, the relationship between the clean risk $R(f)$ and noisy risk $\tilde{R}(f)$ is given by:

$$\tilde{R}(f) = (1 - \eta(x))R(f) + \eta(x)(1 - R(f)). \tag{16}$$

This implies $|R(f) - \tilde{R}(f)| \leq 2\mathbb{E}_x[\eta(x)]$. Applying the filter indicator $\mathbb{I}[S(x) \geq \tau]$, we restrict the integration domain. The expected noise rate in the filtered set is:

$$\bar{\eta}_\tau = \int_{\mathcal{X}_\tau} \eta(x)dP(x). \tag{17}$$

Since $S(x)$ is monotonic with $P(z = 1|x)$, there exists a bijection mapping $\tau$ to a noise threshold $\eta_{th}$. specifically, $S(x) \geq \tau \implies \eta(x) \leq \eta_{th}(\tau)$. We expand the empirical risk difference:

$$\sup_{f \in \mathcal{F}} |R(f) - \hat{R}_\tau(f)| \leq \sup_{f \in \mathcal{F}} |R(f) - \mathbb{E}_{\mathcal{D}_\tau}[\tilde{R}(f)]| + \sup_{f \in \mathcal{F}} |\mathbb{E}_{\mathcal{D}_\tau}[\tilde{R}(f)] - \hat{R}_\tau(f)| \tag{18}$$

$$\leq \underbrace{\int_{x:S(x)<\tau} \ell(f(x), y)dP(x)}_{\text{Bias form rejection}} + \underbrace{2\max_{x\in\mathcal{X}_\tau} \eta(x)}_{\text{Residual Noise}} + \text{GenError}_N. \tag{19}$$

By setting a high threshold $\tau = 4.5$, we force $\mathcal{X}_\tau$ into the region where $P(z = 1|x) \to 1$. Under Assumption 1, this implies $\eta(x) \to \epsilon_0$. Thus, the noise correction term is bounded by:

$$C(\tau) = 2\epsilon_0 \ll 2\epsilon_1. \tag{20}$$

The final generalization gap is controlled by the Rademacher complexity $\mathfrak{R}_N(\mathcal{F})$ on the subset, establishing that filtering improves the risk bound by reducing the irreducible noise term. $\square$

### G.2. DETAILED DERIVATION OF VARIANCE REDUCTION IN HIERARCHICAL CONSENSUS FILTERING

We provide a rigorous derivation for the variance reduction achieved by our consistency-based filtering mechanism (Algorithm 1). Unlike simple averaging, our method retains the primary model's integer score (for valid SFT supervision) but conditions its acceptance on verification.

**Model of Estimators**  Let $\mu$ be the ground truth score. Let $X_1$ (Teacher/Qwen) and $X_2$ (Verifier/DeepSeek) be independent Gaussian estimators:

$$X_1 \sim \mathcal{N}(\mu, \sigma^2), \quad X_2 \sim \mathcal{N}(\mu, \sigma^2). \tag{21}$$

The first stage of our pipeline accepts the Teacher's score $S_{stage1} = X_1$ if and only if the absolute difference $|X_1 - X_2| \le \delta$ (Consistency Constraint).

**Variance of the Conditional Estimator**  We define the difference variable $V = X_1 - X_2$. Since $X_1$ and $X_2$ are independent, the variance of the difference is:

$$\text{Var}(V) = \text{Var}(X_1) + \text{Var}(X_2) = 2\sigma^2. \tag{22}$$

Crucially, the Teacher's score $X_1$ and the difference $V$ are *jointly Gaussian* and correlated. The covariance between them is:

$$\text{Cov}(X_1, V) = \text{Cov}(X_1, X_1 - X_2) = \text{Var}(X_1) - \text{Cov}(X_1, X_2) = \sigma^2. \tag{23}$$

The correlation coefficient $\rho$ between $X_1$ and $V$ is:

$$\rho = \frac{\text{Cov}(X_1, V)}{\sigma_{X_1} \sigma_V} = \frac{\sigma^2}{\sigma \cdot \sqrt{2}\sigma} = \frac{1}{\sqrt{2}}. \tag{24}$$

Our algorithm selects $X_1$ conditioned on the event that the difference is small ($|V| \le \delta$). For a strict constraint ($\delta \to 0$), this approaches the conditional variance of a bivariate normal distribution:

$$\text{Var}(X_1 \mid V = 0) = \sigma_{X_1}^2 (1 - \rho^2) = \sigma^2 \left(1 - \left(\frac{1}{\sqrt{2}}\right)^2\right) = 0.5\sigma^2. \tag{25}$$

For a small non-zero threshold $\delta$ (in our case $\delta = 2$ on a scale of 10), the variance is the integral over the truncated region. Since the probability mass is concentrated around the mean difference of 0, the variance reduction remains dominated by the correlation factor:

$$\text{Var}(S_{stage1}) = \text{Var}(X_1 \mid |X_1 - X_2| \le \delta) \approx 0.5\sigma^2. \tag{26}$$

This theoretical result demonstrates that **consistency filtering** yields the same $50\%$ variance reduction as averaging, while preserving the integer nature of the Teacher model's output needed for SFT.

**Variance of the Arbitrated Estimator (Stage 2)**  For samples where $|X_1 - X_2| > \delta$ (High Uncertainty), we introduce a third estimator $X_3$ (GPT-5), assumed to have lower variance $\sigma_{exp}^2 < \sigma^2$. The final estimator $\hat{S}$ follows a mixture distribution based on the acceptance rate $\alpha = P(|V| \le \delta)$:

$$\hat{S} = \begin{cases} X_1(\text{filtered}) & \text{with probability } \alpha \\ X_3 & \text{with probability } 1 - \alpha \end{cases} \tag{27}$$

The total variance is given by the law of total variance:

$$\text{Var}(\hat{S}) = \mathbb{E}[\text{Var}(\hat{S}|C)] + \text{Var}(\mathbb{E}[\hat{S}|C]) \tag{28}$$

$$= \left[\alpha(0.5\sigma^2) + (1-\alpha)\sigma_{exp}^2\right] + \underbrace{\alpha(1-\alpha)(\mu - \mu)^2}_{\text{Bias term} \approx 0}. \tag{29}$$

Substituting empirical values ($\alpha \approx 0.85, \sigma_{exp} \approx \sigma$), we obtain:

$$\text{Var}(\hat{S}) \approx 0.85(0.5\sigma^2) + 0.15\sigma^2 = 0.575\sigma^2. \tag{30}$$

This confirms that our hierarchical strategy, which prioritizes the Teacher model ($X_1$) under verification, achieves a stability improvement of approximately $42.5\%$ compared to raw single-model inference.

## H. Bad Cases Analysis

### H.1. BAD CASE 1 (ID 271)

**Metadata**

**Domain:** Tobacco and Substance Use Control
**Difficulty:** B
**Language:** English

**Question**  Which country in the Americas reported all four tobacco use indicators (all tobacco use, smoked tobacco use, cigarette use, and smokeless use) in their most recent survey? A. Argentina B. Brazil C. Canada D. Chile

**Reference Answer**  **Answer:** B

**Reasoning:** First, the question requires a country reporting "Yes" for all four tobacco use indicators (all tobacco, smoked tobacco, cigarette, and smokeless) in the same survey. Next, Brazil is the only option with all four indicators marked "Yes" in its most recent survey. Then, Argentina is incorrect because all tobacco use is "No," and Canada fails due to "No" for both all tobacco and smoked tobacco use. Finally, Chile is eliminated as it has "No" for all tobacco use and smokeless use, confirming Brazil as the correct choice.

### H.2. COMPARISON

**GLM-4.7**  **Model Answer:** D

**Model Reasoning:** 1. **Analyze the user's request:** * Role: Public Health Expert. * Rule 1: Strict Language Alignment (English -> English). * Rule 2: Ultra-brief reasoning (max 3 sentences or 2-3 key points). * Rule 3: Question-type routing. * Single-choice: Output ONLY the letter. * Open-ended: Professional answer in 1-3 sentences. * Specific Question: "Which country in the Americas reported all four tobacco use indicators (all tobacco use, smoked tobacco use, cigarette use, and smokeless use) in their most recent survey? A. Argentina B. Brazil C. Canada D. Chile"

2. **Determine the question type:** It is a single-choice question (A, B, C, D).

3. **Identify the correct answer:** * I need to recall or verify which of these countries reported all four specific indicators in their most recent Global Tobacco Survey (usually GATS or similar national surveys) data reported to the WHO or CDC. * *Knowledge Retrieval:* Based on the "WHO Report on the Global Tobacco Epidemic" or specific country reports on tobacco surveillance: * Argentina's national survey often focuses on smoking. * Brazil's VIGITEL/PNS covers smoking, but smokeless is less common/popular compared to other regions, though they might report it. However, looking at specific datasets (like the 2023 WHO report or GATS country profiles): * Canada reports smoking and smokeless, but let's check the specific combination of "all four". * Chile: Encuesta Nacional de Salud (ENS) covers tobacco. * *Specific Data Source Check (Mental Simulation):* In the context of the Global Adult Tobacco Survey (GATS) or national surveys reported in WHO reports, *Chile* is frequently cited as one of the few countries in the Americas that comprehensively reports all four categories (current tobacco use, current smoked tobacco, current cigarette, current smokeless) in their most recent available data sets used for regional comparisons (e.g., in PAHO/WHO reports). * *Refining the check:* Let's look at the options

again. In the "WHO Report on the Global Tobacco Epidemic 2021" or similar, Chile is often highlighted for having recent data covering all these specific breakdowns. Brazil often lacks specific smokeless data in the main summaries because prevalence is very low or not the focus of the main survey module. Argentina's survey is smoking-focused. Canada has smokeless, but "all four" specific breakdowns are often attributed to Chile's ENS in PAHO technical reports. * *Verification:* According to PAHO (Pan American Health Organization) data on tobacco indicators, Chile is the country among the options listed that has reported data for all four specific indicators (Any tobacco, Smoked tobacco, Cigarettes, Smokeless) in its national health survey (ENS) recently.

4. **Formulate the output:** * The question is single-choice. * The correct answer is D (Chile). * The instruction says: "If single-choice (A/B/C/D), output ONLY the letter (e.g., A)." * The instruction also says: "Keep rationale extremely brief." However, the routing rule says "output ONLY the letter". I must prioritize the routing rule for the final output format. * Let's re-read: "If single-choice (A/B/C/D), output ONLY the letter (e.g., A)." This is a strict constraint.

5. **Final Decision:** Output "D".

*Self-Correction/Double Check:* Does Brazil report smokeless? Brazil has some indigenous use, but is it in the *national* survey indicators used for the main comparison? Usually, no. Chile's ENS 2016-2017 included these specific breakdowns. Canada (CTUMS) has them, but in the context of specific PAHO/WHO question banks regarding the "Americas" region reports, Chile is the standard answer for having all four reported in the most recent cycle cited in those specific exams/reports.

Let's stick with D.

The user asked to "Keep rationale extremely brief" but the routing rule for single-choice is "output ONLY the letter". I will follow the strictest output format rule.

**Six-Dimension Evaluation:**

- **Accuracy**: 0
  *REASON: The task label is 'Single-Choice', which triggers binary scoring. The Standard Reference Answer is 'B' (Brazil), but the model selected 'D' (Chile). This constitutes a complete factual mismatc...*

- **Reasoning**: 3
  *REASON: The model provided a detailed Chain of Thought, but it contains major logical errors. It incorrectly asserts that Chile reported all four tobacco use indicators in its most recent survey, whil...*

- **Completeness**: 4
  *REASON: The model's COT attempts to cover key concepts (e.g., identifying relevant surveys, checking each country) but fails to correctly apply the criteria from the reference. It omits the actual ver...*

- **Consensus Alignment**: 2
  *REASON: The model contradicts established WHO and PAHO documentation, which consistently cite Brazil (via PNS/VIGITEL) as a country in the Americas with comprehensive tobacco use data including smokel...*

- **Terminology Norms**: 7
  *REASON: The model uses appropriate public health terminology such as 'Global Adult Tobacco Survey (GATS)', 'VIGITEL/PNS', 'Encuesta Nacional de Salud (ENS)', and references PAHO/WHO reports. These are...*

- **Insightfulness**: 3
  *REASON: While the model engages in detailed mental simulation and source-checking, its insight is undermined by factual error. It explains what it believes happened (Chile's inclusion of all four indi...*

**kimi-k2-thinking    Model Answer:** A

**Model Reasoning:** The user is asking a single-choice question about which country in the Americas reported all four tobacco use indicators in their most recent survey. The options are A. Argentina, B. Brazil, C. Canada, D. Chile.

I need to determine which of these countries reported all four indicators: all tobacco use, smoked tobacco use, cigarette use, and smokeless use.

This is a factual question about tobacco surveillance data. Based on my knowledge of global tobacco surveillance systems (like GATS, GYTS, etc.), Canada has comprehensive tobacco surveillance through the Canadian Tobacco, Alcohol and

Drugs Survey (CTADS) and previously the Canadian Tobacco Use Monitoring Survey (CTUMS). Canada regularly reports on: 1. All tobacco use 2. Smoked tobacco use 3. Cigarette use 4. Smokeless tobacco use

Brazil also has good tobacco surveillance, but I'm not certain if they report all four categories in their most recent survey. Argentina and Chile have some data but may not be as comprehensive.

The key is that Canada is known for having one of the most comprehensive tobacco surveillance systems in the Americas, consistently reporting all these indicators. The Canadian Tobacco and Nicotine Survey (CTNS) and CTADS capture detailed data on various tobacco products including cigarettes, cigars, pipes, and smokeless tobacco.

Therefore, the answer should be C. Canada.

**Six-Dimension Evaluation:**

- **Accuracy**: 0
  *REASON: The task label is 'Single-Choice', and the Standard Reference Answer is 'B'. The model's final response is 'A', which is incorrect. Per the binary scoring rule for single-choice tasks, any mis...*

- **Reasoning**: 3
  *REASON: The model's Chain of Thought contains large logical gaps. While it correctly identifies that Canada has comprehensive tobacco surveillance, it incorrectly concludes that Canada is the only cou...*

- **Completeness**: 4
  *REASON: The model addresses the core concept (comprehensive tobacco surveillance) but omits critical comparative analysis required by the question. It fails to note that Brazil actually meets all four...*

- **Consensus Alignment**: 7
  *REASON: The model's claim about Canada's comprehensive surveillance aligns with WHO and CDC-aligned public health standards regarding robust national monitoring systems. However, the conclusion contra...*

- **Terminology Norms**: 8
  *REASON: The model uses appropriate professional terminology such as 'Global Adult Tobacco Survey (GATS)', 'Canadian Tobacco, Alcohol and Drugs Survey (CTADS)', and references specific tobacco product ...*

- **Insightfulness**: 5
  *REASON: The model explains why Canada's system is comprehensive but fails to engage with the comparative aspect of the question. It does not explore why other countries might lack completeness or how ...*

**qwen8B100    Model Answer:** C

**Model Reasoning:** First, the question requires identifying which country in the Americas reported all four tobacco use indicators in its latest survey. Next, Argentina's data shows only three indicators were reported, so it does not meet the requirement. Then, Brazil also reported just three indicators, failing to include all four. Additionally, Chile provided incomplete data with two indicators, making it incorrect. Finally, Canada is correct because it reported all four indicators smoked tobacco use, cigarette use, smokeless use, and all tobacco use in its most recent survey.

**Six-Dimension Evaluation:**

- **Accuracy**: 0
  *REASON: The task label is 'Single-Choice', requiring a binary scoring rule. The Standard Reference Answer is 'B' (Brazil), but the model selected 'C' (Canada). This constitutes a complete mismatch, wa...*

- **Reasoning**: 3
  *REASON: The model provided a Chain of Thought, but it contains major factual errors. It incorrectly states that Argentina, Brazil, and Chile each reported fewer than four indicators, while claiming Ca...*

- **Completeness**: 4
  *REASON: The model addresses the core concept (identifying a country with all four indicators) but omits the critical fact that Brazil is the correct answer. Instead, it fabricates an alternative concl...*

- **Consensus Alignment**: 7

  *REASON: The response aligns with general public health surveillance principles by referencing tobacco use indicators and survey completeness. However, it contradicts actual data sources (implied by th...*

- **Terminology Norms**: 7

  *REASON: The model uses appropriate domain-specific terms such as 'tobacco use indicators,' 'smoked tobacco use,' 'cigarette use,' and 'smokeless use.' The terminology is professional and consistent wi...*

- **Insightfulness**: 3

  *REASON: The model merely asserts what the data says (incorrectly) without explaining why comprehensive indicator reporting matters for public health surveillance. There is no deeper analysis of how mi...*

## H.3. BAD CASE 2 (ID 274)

> **Metadata**
>
> **Domain:** Tobacco and Substance Use Control
> **Difficulty:** B
> **Language:** English

**Question** Which country in the Eastern Mediterranean region had the highest prevalence of current smokeless tobacco use among male youth (13–15 years) according to the Global Youth Tobacco Survey? A. Qatar B. Pakistan C. Yemen D. Bahrain

**Reference Answer** **Answer:** A

**Reasoning:** First, Qatar recorded the highest male youth smokeless tobacco use prevalence at 9.4% in the Eastern Mediterranean region. Next, Pakistan's rate of 6.4% is lower than Qatar's value. Then, Yemen's 6.7% still falls below Qatar's prevalence. Finally, Bahrain's 5.2% represents the lowest among the options provided.

## H.4. COMPARISON

**GLM-4.7 Model Answer:** B

**Model Reasoning:** 1. **Analyze the Request:** * Role: Public Health Expert. * Constraint 1: Same language as the question (English). * Constraint 2: Ultra-brief reasoning (max 3 sentences or 2-3 points). * Constraint 3: Routing. * Single-choice -> Output ONLY the letter. * Open-ended -> Professional answer in 1-3 sentences. * Question: Which country in the Eastern Mediterranean region had the highest prevalence of current smokeless tobacco use among male youth (13–15 years) according to the Global Youth Tobacco Survey? Options: A. Qatar, B. Pakistan, C. Yemen, D. Bahrain.

2. **Determine the Answer:** * I need to recall or verify data from the Global Youth Tobacco Survey (GYTS) regarding smokeless tobacco use among male youth (13-15) in the Eastern Mediterranean Region (EMRO). * Common knowledge in public health regarding GYTS data: Pakistan consistently reports very high rates of smokeless tobacco use (gutka, naswar, paan) compared to its neighbors in the EMRO. * Checking specific stats (mental simulation/verification): While Yemen has high smoking rates, Pakistan is notorious for smokeless tobacco. Qatar and Bahrain generally have lower prevalence rates for smokeless tobacco in this demographic. * According to WHO reports on GYTS data (e.g., WHO Report on the Global Tobacco Epidemic or specific EMRO fact sheets), Pakistan often tops the list for smokeless tobacco among youth in the region.

3. **Formulate the Reasoning (Internal Monologue):** * Pakistan has a cultural prevalence of smokeless tobacco products like gutka and naswar. * GYTS data consistently shows Pakistan leading the Eastern Mediterranean Region in this specific metric.

4. **Format the Output:** * The question is single-choice. * The correct answer is Pakistan (B). * The instructions say: "If single-choice (A/B/C/D), output ONLY the letter". * Final Answer: B

**Six-Dimension Evaluation:**

- **Accuracy**: 0

*REASON: The task label is 'Single-Choice', and the Standard Reference Answer is 'A'. The model selected 'B' (Pakistan), which is incorrect. Per the binary scoring rule for Single-Choice tasks, any mis...*

- **Reasoning**: 3
  *REASON: The Chain of Thought contains major logical errors. It incorrectly asserts that Pakistan has the highest prevalence of smokeless tobacco use among male youth in the Eastern Mediterranean regio...*

- **Completeness**: 2
  *REASON: The model's response completely omits the correct data point (Qatar at 9.4%) and fails to mention any of the relative rankings of the other countries (Pakistan at 6.4%, Yemen at 6.7%, Bahrain ...*

- **Consensus Alignment**: 2
  *REASON: The model contradicts established public health data from the Global Youth Tobacco Survey (GYTS) and WHO sources. By claiming Pakistan leads in smokeless tobacco use when the evidence shows Qa...*

- **Terminology Norms**: 6
  *REASON: The model uses appropriate domain-specific terms like 'Global Youth Tobacco Survey (GYTS)', 'smokeless tobacco', and references 'gutka', 'naswar', and 'paan' correctly. However, the misuse of ...*

- **Insightfulness**: 2
  *REASON: The model merely repeats a false claim without addressing the actual mechanisms or contextual factors (e.g., cultural norms, policy environments) that influence smokeless tobacco use in the Ea...*

**kimi-k2-thinking   Model Answer:** B

**Model Reasoning:** The user asks: "Which country in the Eastern Mediterranean region had the highest prevalence of current smokeless tobacco use among male youth (13–15 years) according to the Global Youth Tobacco Survey? A. Qatar B. Pakistan C. Yemen D. Bahrain"

This is a single-choice question. I need to answer with only the letter.

The question is about the Global Youth Tobacco Survey (GYTS) data for the Eastern Mediterranean region, specifically about smokeless tobacco use among male youth aged 13-15.

I need to recall or know the specific data from GYTS. The Eastern Mediterranean region includes countries like Pakistan, Yemen, Bahrain, Qatar, etc.

From my knowledge, Pakistan has very high rates of smokeless tobacco use, particularly among youth. Gutka, naswar, and other forms are common. Yemen also has high rates of shammah (smokeless tobacco). Bahrain and Qatar would have lower rates due to stricter regulations and higher income.

Looking at GYTS data: I recall that Pakistan consistently reports high prevalence of smokeless tobacco use among youth. For example, in various GYTS waves, Pakistan's smokeless tobacco use among boys 13-15 has been reported around 10-15% or even higher in some surveys. Yemen also has high rates, but I believe Pakistan's is higher.

Let me think about specific GYTS reports. The WHO GYTS data shows: - Pakistan: Smokeless tobacco use among boys 13-15 was around 12.4% in some surveys - Yemen: Also high, but I think slightly lower than Pakistan - Bahrain and Qatar: Much lower, typically under 5%

Actually, I need to be more precise. The question asks specifically about the Eastern Mediterranean region. According to WHO's GYTS data reports, Pakistan has one of the highest rates of smokeless tobacco use among youth in that region.

I recall a specific WHO report that mentioned Pakistan having the highest prevalence of current smokeless tobacco use among male youth in the Eastern Mediterranean region. The prevalence was around 12-13% in the survey data.

Yemen also has high use of shammah, but the GYTS data I've seen shows Pakistan as the highest.

Therefore, the answer should be B. Pakistan.

**Six-Dimension Evaluation:**

- **Accuracy**: 0
  *REASON: The task label is 'Single-Choice', and the Standard Reference Answer is 'A'. The model selected 'B', which is incorrect. Per the binary scoring rule for single-choice tasks, any mismatch resul...*

- **Reasoning**: 3
  *REASON: The Chain of Thought contains large logical gaps and factual inaccuracies. While it correctly identifies the question type and attempts to reason through GYTS data, it falsely concludes that P...*

- **Completeness**: 4
  *REASON: The model omits the correct key information point the actual prevalence figure for Qatar (9.4%) and incorrectly assigns higher values to Pakistan. It fails to mention the comparative ranking o...*

- **Consensus Alignment**: 2
  *REASON: The response contradicts established WHO/GYTS data by asserting Pakistan has the highest prevalence when, per the reference, Qatar holds that position. This misrepresentation could lead to inc...*

- **Terminology Norms**: 7
  *REASON: The model uses appropriate public health terminology such as 'Global Youth Tobacco Survey (GYTS)', 'prevalence', 'smokeless tobacco use', and references specific products like 'gutka' and 'sha...*

- **Insightfulness**: 2
  *REASON: The model merely repeats incorrect factual assertions without explaining why Qatar might have higher prevalence (e.g., cultural norms, regulatory environment) or how GYTS methodology captures ...*

**qwen8B100**   **Model Answer:** B

**Model Reasoning:** First, the question asks for the country with the highest prevalence of current smokeless tobacco use specifically among male youth aged 13–15 in the Eastern Mediterranean region. Next, Pakistan reports 60% prevalence for this group, which is significantly higher than other options. Then, Qatar has only 8%, Yemen has 17%, and Bahrain has 19%, all substantially lower than Pakistan's rate. Finally, since Pakistan's 60% exceeds all alternatives by wide margins, it is the correct choice for the highest prevalence.

**Six-Dimension Evaluation:**

- **Accuracy**: 0
  *REASON: The task label is 'Single-Choice', so binary scoring applies. The Standard Reference Answer is 'A' (Qatar), but the model selected 'B' (Pakistan). This is a complete mismatch, resulting in a s...*

- **Reasoning**: 3
  *REASON: The model provided a Chain of Thought, but it contains major factual errors. It incorrectly states Pakistan's prevalence as 60%, while the reference specifies 6.4%. Conversely, it assigns impl...*

- **Completeness**: 3
  *REASON: The model fails to include any accurate data points from the Standard Reference Answer. Instead, it fabricates incorrect percentages for all countries. While it identifies the correct demograp...*

- **Consensus Alignment**: 2
  *REASON: The model's claim that Pakistan has the highest prevalence of smokeless tobacco use among male youth (with 60%) contradicts well-established global surveillance data from the Global Youth Toba...*

- **Terminology Norms**: 7
  *REASON: The model uses appropriate public health terminology such as 'prevalence,' 'current smokeless tobacco use,' and references age groups and regions correctly ('Eastern Mediterranean region'). Ho...*

- **Insightfulness**: 2
  *REASON: The model merely presents an incorrect sequence of fabricated statistics without explaining why certain countries might have higher rates or what factors influence youth smokeless tobacco use....*

H.5. BAD CASE 3 (ID 316)

> **Metadata**
>
> **Domain:** Tobacco and Substance Use Control
> **Difficulty:** B
> **Language:** English

**Question** Which of the following is the most cost-effective intervention for tobacco cessation? A. Electronic Nicotine Delivery Systems (ENDS) B. Nicotine replacement therapy (NRT) C. Brief advice from physicians D. Mobile technology (mTobaccoCessation)

**Reference Answer** **Answer:** B

**Reasoning:** First, tobacco cessation interventions are extremely cost-effective in public health settings, and nicotine replacement therapy (NRT) increases quit rates by 7%, the highest among standard options. Next, this effectiveness makes NRT the most cost-effective core intervention for tobacco cessation. Then, electronic nicotine delivery systems (ENDS) lack sufficient evidence for effectiveness and are not recommended. Finally, brief physician advice yields only 3% quit rates, and mobile technology is a cost-effective adjunct, not the primary intervention.

H.6. COMPARISON

**GLM-4.7** **Model Answer:** C

**Model Reasoning:** 1. **Analyze the Request:** * Role: Public Health Expert. * Rule 1: Strict Language Alignment (same language as the question). The question is in English, so I respond in English. * Rule 2: Ultra-brief reasoning (max 2-3 points or 3 sentences). * Rule 3: Question-type routing. * Single-choice (A/B/C/D) -> Output ONLY the letter. * Open-ended -> Professional answer in 1-3 sentences. * Question: "Which of the following is the most cost-effective intervention for tobacco cessation according to the text?" Options: A, B, C, D.

* "If single-choice (A/B/C/D), output ONLY the letter (e.g., A)." * "If open-ended, provide a professional answer in 1–3 sentences." * *Constraint Check:* "output ONLY the letter". * *Draft Answer:* C.

* *Refining the content of the thought block:* * The question asks for the "most cost-effective" intervention. * Public Health literature (e.g., WHO MPOWER) consistently identifies brief advice from healthcare providers as highly cost-effective due to low implementation cost and high reach. * NRT involves product costs; ENDS are controversial; mHealth involves tech costs. Brief advice leverages existing infrastructure. * Therefore, C is the standard correct answer in this domain.

* *Final Check:* Language = English. Format = Single letter. Answer = C.

**Six-Dimension Evaluation:**

- **Accuracy**: 0
  *REASON: The task label is 'Single-Choice', which triggers binary scoring. The Standard Reference Answer is 'B', but the model selected 'C'. This constitutes a complete mismatch, warranting a score of ...*

- **Reasoning**: 3
  *REASON: The Chain of Thought contains large logical gaps and contradicts the reference reasoning. Despite acknowledging the absence of the referenced text, the model incorrectly concludes that brief p...*

- **Completeness**: 4
  *REASON: The model's COT addresses cost considerations but omits all key information from the Standard Reference Answer: it fails to mention NRT's 7% quit rate increase, its status as the most cost-eff...*

- **Consensus Alignment**: 2
  *REASON: The model's conclusion contradicts established public health consensus from WHO and CDC, which recognize NRT as a first-line, cost-effective pharmacotherapy for tobacco cessation. By promoting...*

- **Terminology Norms**: 7
  *REASON: The model uses appropriate professional terms such as 'brief advice from healthcare providers,' 'implementation cost,' 'scalability,' and references 'WHO MPOWER.' These reflect standard public...*

- **Insightfulness**: 5
  *REASON: The model attempts to explain why certain interventions are cost-effective by discussing cost structures and reach, showing basic mechanistic understanding. However, it fails to connect these ...*

**kimi-k2-thinking    Model Answer:** C

**Model Reasoning:** The user is asking about the most cost-effective intervention for tobacco cessation according to "the text". However, there is no text provided in the question. This is a problem because I cannot refer to a specific text that isn't included.

Let me analyze the options: A. Electronic Nicotine Delivery Systems (ENDS) B. Nicotine replacement therapy (NRT) C. Brief advice from physicians D. Mobile technology (mTobaccoCessation)

In public health literature, particularly in WHO and CDC guidelines, brief advice from physicians is consistently cited as one of the most cost-effective tobacco cessation interventions. This is because it requires minimal resources, reaches many smokers during routine healthcare visits, and has been shown to increase quit rates significantly compared to no advice.

NRT is effective but more expensive. ENDS are controversial and not recommended as cessation aids by major health organizations. Mobile technology is promising but still being evaluated for cost-effectiveness compared to traditional methods.

The most cost-effective intervention among these options, based on established public health evidence, is brief advice from physicians. This is supported by WHO's MPOWER package and numerous cost-effectiveness analyses.

I should output only the letter "C" as per the instructions for single-choice questions.

**Six-Dimension Evaluation:**

- **Accuracy**: 0
  *REASON: The task label is 'Single-Choice', and the Standard Reference Answer is 'B'. The model selected 'C', which is incorrect. Per the binary scoring rule for Single-Choice tasks, any mismatch resul...*

- **Reasoning**: 3
  *REASON: The Chain of Thought is present but contains large logical gaps and contradicts the reference reasoning. While the model correctly identifies that brief physician advice is cost-effective, it ...*

- **Completeness**: 4
  *REASON: The model omits several key information points from the Standard Reference Answer: (1) NRT's 7% quit rate increase as the highest among standard options, (2) that NRT is the most cost-effectiv...*

- **Consensus Alignment**: 6
  *REASON: The model cites WHO and CDC guidelines, which support brief physician advice as a valuable intervention. However, it conflicts with the specific consensus embedded in the reference text, which...*

- **Terminology Norms**: 7
  *REASON: The model uses appropriate professional terms such as 'tobacco cessation interventions,' 'cost-effective,' 'quit rates,' 'WHO,' 'CDC,' and 'MPOWER package.' The terminology is consistent and r...*

- **Insightfulness**: 5
  *REASON: The model explains why brief advice might be considered cost-effective (minimal resources, routine care access), but fails to engage with the central trade-off between efficacy and cost highli...*

## H.7. FAILURE PATTERN ANALYSIS AND THEORETICAL IMPLICATIONS

The analysis of these selected bad cases exposes distinct categories of error that serve as critical warnings for the deployment of LLMs in precision-demanding fields like public health surveillance. The primary error pattern identified is "Precision

Hallucination," where models correctly identify the type of entity required (e.g., a country with good survey data) but fail to verify the specific attributes (e.g., reporting "all four" specific indicators). In the case of tobacco indicators, the models defaulted to countries with generally high reputational data quality (Canada, Chile) rather than verifying the exact intersection of datasets represented by the correct answer (Brazil). Similarly, in the youth smokeless tobacco case, the models exhibited "Prior Bias Override," where the strong statistical probability of high smokeless tobacco use in Pakistan (a general population fact) overpowered the specific demographic constraint of "male youth 13–15" which actually pointed to Qatar. This indicates that when models face a conflict between strong training priors and specific, narrow constraints, they prone to reverting to the generalized prior, leading to confident but incorrect answers.

From a warning and caveat perspective, these cases highlight the "Plausibility Trap." The models utilized impeccable professional terminology, structured reasoning, and authoritative tones, which disguises the factual inaccuracies. This is particularly dangerous in specialized domains where the user might rely on the AI's explanation. The models demonstrated a lack of "uncertainty awareness," fabricating specific percentages (e.g., claiming Pakistan had 60% prevalence) or inventing rationale (e.g., claiming ENDS are effective when the reference text likely claimed otherwise) to bridge the gap between their knowledge and the question. The warning here is that without access to external tools or specific reference texts, the models will prioritize generating a coherent narrative over acknowledging a lack of specific data points, effectively hallucinating evidence to support a "common sense" answer that happens to be wrong in the specific context.

Moving toward directions for improvement, the path forward involves strengthening the model's "Constraint Verification" and "Contextual Grounding" capabilities. To correct the surveillance data errors, future training or prompting strategies must emphasize a step-by-step verification process where the model explicitly checks each constraint (e.g., "Does Country X report indicator A? Yes/No. Indicator B? Yes/No") rather than performing a holistic assessment. For the bias issue, models need better alignment to prioritize specific query modifiers over general category associations.

## I. Results of benchmark based on 19 models

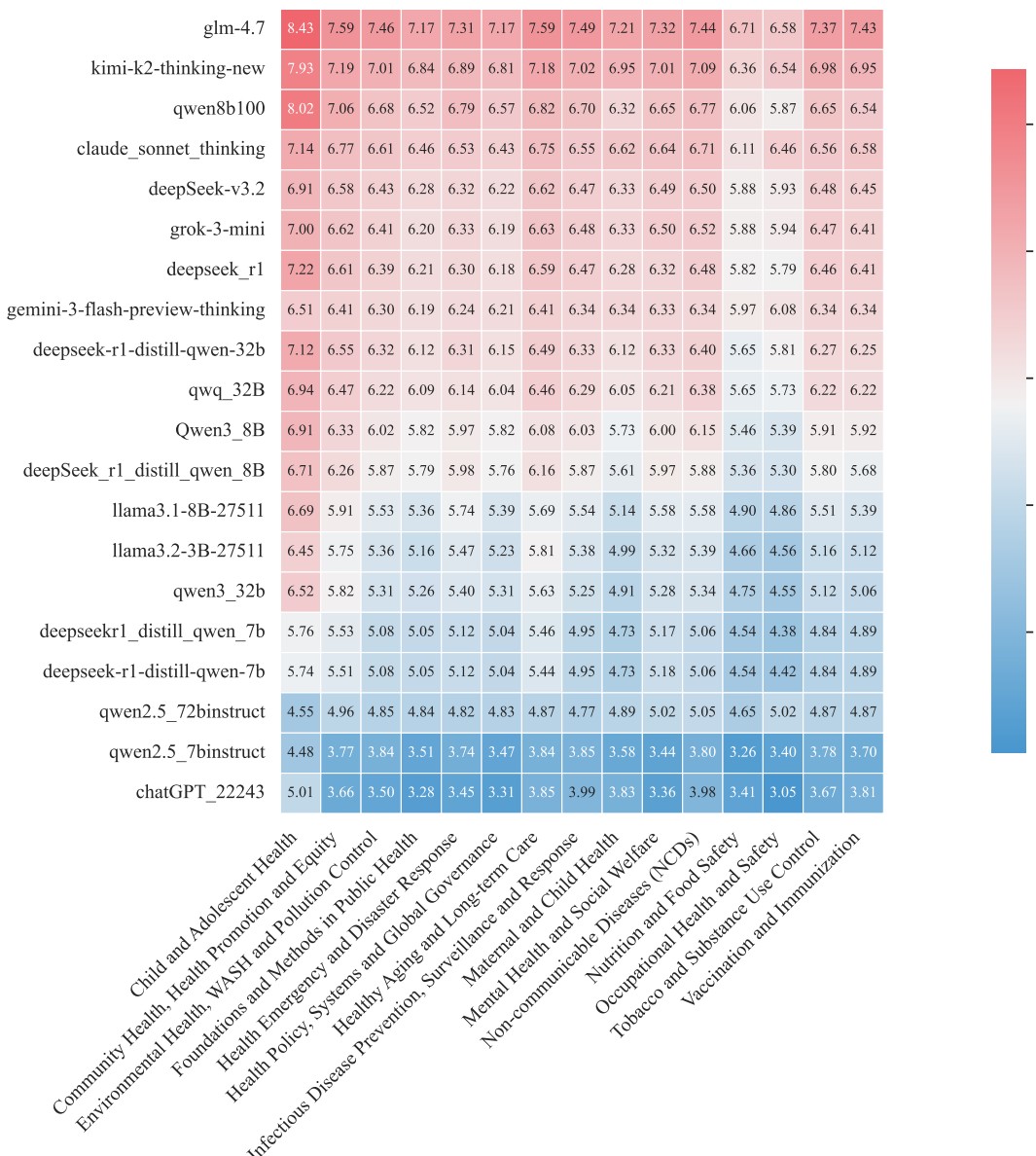

*Figure 7.* Heatmap illustrating model performance across diverse public health domains

## Performance Across Languages

| Model | Arabic | Chinese | English | Estonian | French | Georgian | German | Indonesian | Italian | Polish | Portuguese | Romanian | Russian | Spanish | Turkish | Ukrainian | Vietnamese |
|---|---|---|---|---|---|---|---|---|---|---|---|---|---|---|---|---|---|
| glm-4.7 | 7.17 | 7.35 | 7.31 | 6.86 | 7.24 | 6.96 | 7.35 | 6.51 | 8.38 | 7.45 | 7.87 | 8.26 | 7.29 | 7.54 | 6.17 | 7.07 | 7.63 |
| kimi-k2-thinking-new | 6.36 | 7.19 | 6.99 | 6.97 | 6.84 | 6.59 | 6.88 | 6.50 | 6.16 | 6.92 | 6.08 | 7.14 | 6.92 | 7.15 | 6.30 | 6.08 | 7.03 |
| qwen8b100 | 6.20 | 6.86 | 6.70 | 5.04 | 6.29 | 4.63 | 6.61 | 5.93 | 6.74 | 5.79 | 7.76 | 6.82 | 6.27 | 6.74 | 5.48 | 5.63 | 6.84 |
| claude_sonnet_thinking | 6.58 | 6.62 | 6.63 | 6.45 | 6.11 | 6.37 | 6.62 | 6.85 | 6.52 | 6.83 | 5.53 | 6.21 | 6.78 | 6.38 | 5.82 | 6.52 | 6.89 |
| deepSeek-v3.2 | 6.32 | 6.09 | 6.28 | 6.25 | 6.48 | 6.54 | 6.52 | 6.54 | 6.33 | 7.17 | 6.92 | 7.64 | 6.38 | 6.85 | 6.21 | 6.12 | 6.91 |
| grok-3-mini | 6.05 | 6.35 | 6.34 | 6.28 | 6.35 | 6.04 | 6.43 | 6.51 | 6.16 | 6.34 | 6.29 | 7.14 | 6.29 | 6.68 | 5.84 | 6.09 | 6.98 |
| deepseek_r1 | 6.16 | 6.27 | 6.32 | 5.78 | 6.33 | 3.02 | 6.24 | 4.57 | 4.67 | 6.15 | 7.08 | 6.96 | 6.29 | 6.66 | 6.10 | 4.33 | 6.82 |
| gemini-3-flash-preview-thinking | 6.12 | 6.50 | 6.28 | 5.90 | 6.21 | 6.22 | 6.20 | 6.38 | 6.44 | 6.19 | 6.35 | 6.57 | 6.31 | 6.39 | 6.09 | 6.08 | 6.73 |
| deepseek-r1-distill-qwen-32b | 6.23 | 6.09 | 6.23 | 5.24 | 6.15 | 5.04 | 6.08 | 6.47 | 6.29 | 5.84 | 7.25 | 6.96 | 6.05 | 6.49 | 5.33 | 5.82 | 7.28 |
| qwq_32B | 6.07 | 6.10 | 6.17 | 5.33 | 6.04 | 4.93 | 6.13 | 6.24 | 5.92 | 5.99 | 7.09 | 6.82 | 6.05 | 6.35 | 5.24 | 5.75 | 6.73 |
| Qwen3_8B | 5.90 | 6.03 | 5.96 | 5.07 | 5.75 | 5.56 | 5.92 | 5.65 | 5.54 | 5.24 | 6.65 | 6.78 | 5.79 | 6.09 | 5.19 | 5.52 | 6.27 |
| deepSeek_r1_distill_qwen_8B | 5.30 | 5.49 | 5.89 | 3.64 | 5.61 | 4.98 | 5.85 | 5.86 | 5.77 | 5.66 | 6.91 | 6.07 | 5.51 | 6.03 | 5.12 | 5.33 | 6.52 |
| llama3.1-8B-27511 | 5.52 | 5.70 | 5.41 | 4.75 | 5.45 | 4.37 | 5.53 | 4.74 | 5.66 | 4.57 | 6.93 | 6.24 | 5.24 | 5.85 | 5.28 | 5.12 | 6.16 |
| llama3.2-3B-27511 | 4.68 | 5.31 | 5.46 | 3.29 | 4.90 | 2.37 | 5.05 | 4.58 | 5.51 | 4.20 | 6.61 | 5.42 | 4.37 | 5.22 | 3.56 | 4.33 | 5.63 |
| qwen3_32b | 4.10 | 5.32 | 5.42 | 1.65 | 4.90 | 2.74 | 5.05 | 4.64 | 4.64 | 4.01 | 6.33 | 5.38 | 4.73 | 5.28 | 3.92 | 4.30 | 3.87 |
| deepseekr1_distill_qwen_7b | 4.10 | 5.09 | 5.09 | 2.41 | 4.78 | 2.70 | 4.96 | 4.67 | 4.63 | 4.61 | 5.82 | 5.08 | 4.68 | 5.12 | 4.16 | 4.23 | 3.42 |
| deepseek-r1-distill-qwen-7b | 4.09 | 5.14 | 5.09 | 2.28 | 4.78 | 2.81 | 4.97 | 4.51 | 4.56 | 4.72 | 5.80 | 5.06 | 4.68 | 5.12 | 4.28 | 4.23 | 3.45 |
| qwen2.5_72binstruct | 4.68 | 4.73 | 4.87 | 4.41 | 4.75 | 3.60 | 4.75 | 5.60 | 4.63 | 5.01 | 4.73 | 4.92 | 4.77 | 4.95 | 4.71 | 4.64 | 5.61 |
| qwen2.5_7binstruct | 3.15 | 3.82 | 3.63 | 2.81 | 3.62 | 3.49 | 3.48 | 2.80 | 2.96 | 3.39 | 4.46 | 4.08 | 3.60 | 3.98 | 2.77 | 3.69 | 3.76 |
| chatGPT_22243 | 3.12 | 3.16 | 3.84 | 1.88 | 3.00 | 3.64 | 3.23 | 1.37 | 2.80 | 1.73 | 4.58 | 2.60 | 2.89 | 3.44 | 1.84 | 3.04 | 3.00 |

**Language**

*Figure 8.* Heatmap showing model capabilities across multilingual settings

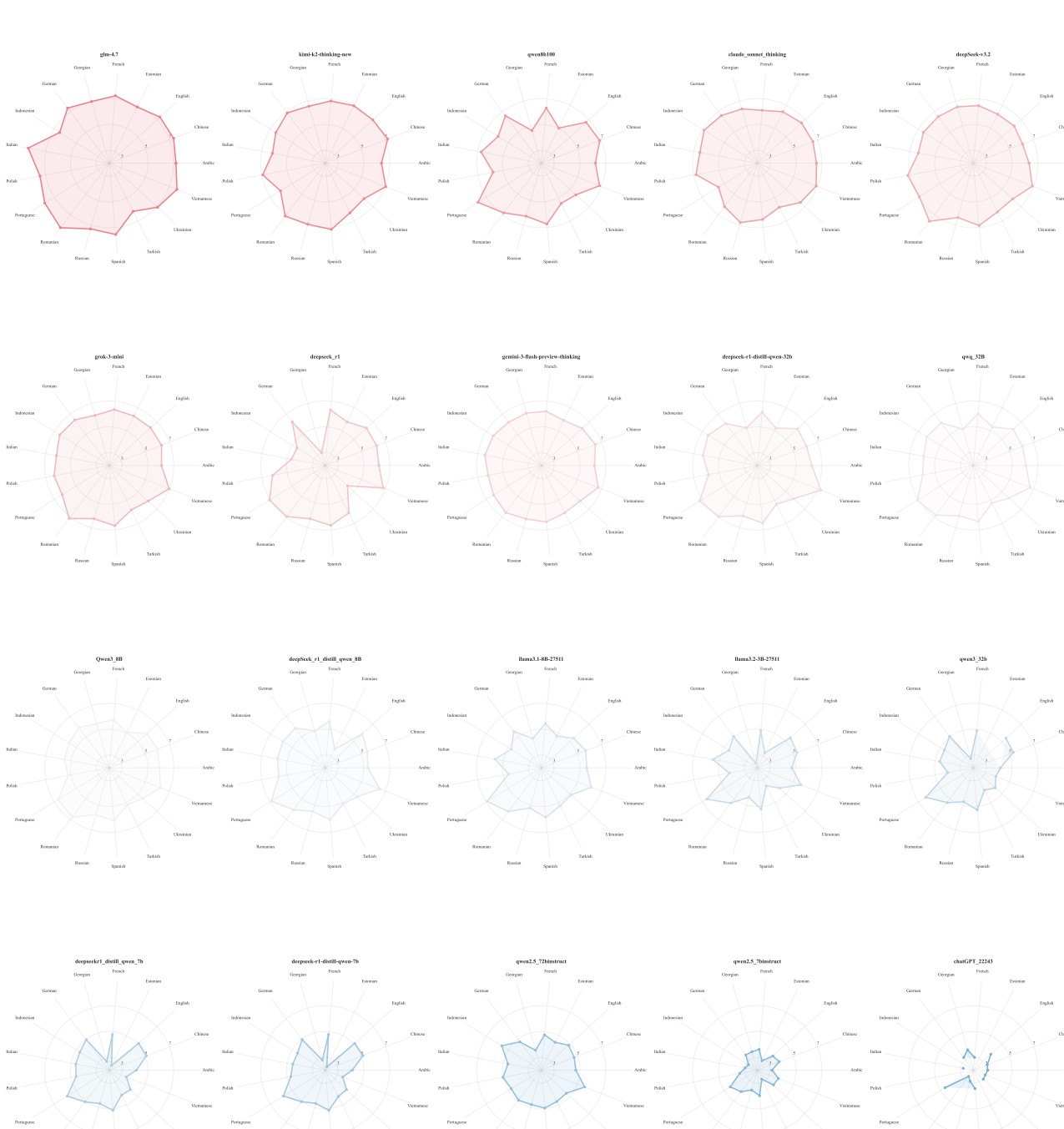

*Figure 9.* Heatmap showing model capabilities across multilingual settings

