# OpenReview forum: "From Knowledge to Inference: Formalizing Specialized Public Health Reasoning on GlobalHealthAtlas"
_ICML.cc/2026/Conference — ICML 2026 regular_

### Official Review · Reviewer_ppWD · 2026-03-11

**Soundness:** 3
**Presentation:** 3
**Significance:** 3
**Originality:** 2
**Overall Recommendation:** 4
**Confidence:** 4

**Summary:**

The paper introduces GlobalHealthAtlas, a large-scale multilingual public health reasoning dataset containing 280,210 instances across 15 public health domains and 17 languages, with three difficulty levels. In addition, the authors propose an evaluation framework and develop a specialized LLM trained on this dataset. The results highlight the dominance of strong reasoning models and the effectiveness of domain-specific fine-tuning.

**Compliance With Llm Reviewing Policy:**

Affirmed.

**Final Justification:**

The rebuttal addressed my concerns

**Key Questions For Authors:**

How to ensure the benchmark quality given all the benchmark is constructed by LLM?

**Limitations:**

yes

**Strengths And Weaknesses:**

Strengths
1. The paper is well motivated, addressing the lack of high-quality public health datasets.
2. The paper includes an interesting analysis of potential data leakage, which is valuable when proposing a new benchmark.

Weaknesses
1. It is unclear how the dataset quality is ensured. The paper relies heavily on LLM-based evaluation (e.g., LLMs generating, filtering, and judging the dataset). This self-evaluation loop raises concerns about reliability, especially for reasoning tasks. It would be helpful to include manual verification or human evaluation.

2. The scaling analysis appears limited. The experiments only include relatively small models (4B, 8B, and 14B), while the dataset size is also below 1M samples. Given this setup, it is difficult to draw strong conclusions about scaling laws (especially the paper title is Scaling Laws of Specialized Reasoning ). Including larger models (e.g., 70B or >100B) would strengthen the claims.

---

> ### Author Rebuttal · Authors · 2026-03-31
>
> W1: We thank the reviewer for this important and thoughtful concern. We fully agree that benchmark quality should not rely solely on an automated LLM-based loop, especially for reasoning-oriented tasks. Our pipeline is therefore not entirely dependent on LLMs, and human involvement was incorporated at multiple stages.
>
> First, the raw source material of GlobalHealthAtlas comes directly from authoritative official documents, primarily from the World Health Organization and related trusted public health sources. In this sense, LLMs are used mainly for structured extraction, format transformation, and candidate generation, rather than for inventing facts or producing unconstrained knowledge. The factual foundation of the benchmark is grounded in official documents rather than model-generated content.
>
> Second, human expertise is involved in the quality-control process. The evaluation dimensions and weights used in the quality framework were determined and confirmed with expert input, rather than being arbitrarily chosen by the model itself. In addition, after the evaluator was constructed, we further assessed its reliability through human evaluation.
>
> we have added a dedicated human evaluation as requested, specifically, we sampled 5% of the full dataset, covering all 15 domains and all 17 languages, including 7,077 Single-Choice items and 6,933 Question-Answer items, for a total of 14,010 examples.
>
> The scoring dimensions, evaluation criteria, and weights used in the human evaluation were fully consistent with those used in the LLM-based evaluation on Pages 21 and 28. For single-choice questions, evaluation is based on four dimensions: Question Quality (0.45), Answer Quality (0.25), Text Relevance (0.20), and Overall Consistency (0.10), with S = 0.45Q + 0.25A + 0.20R + 0.10C.For question-answer items, evaluation is also based on four dimensions: Question Quality (0.25), Answer Quality (0.35), Text Relevance (0.25), and Overall Consistency (0.15), with S = 0.25Q + 0.35A + 0.25R + 0.15C. Each dimension is scored on a 5-point scale,
>
> 14 experts conducted manual scoring based above indicators. The resulting average human score was 4.503. Since scores are above 4.0, we believe this provides supportive evidence that the benchmark quality is high and that the automated scoring is broadly aligned with expert judgment.
>
> We will add this human evaluation to the manuscript and clarify that the benchmark is supported by both authoritative source documents and expert-involved quality control, rather than being constructed by LLMs alone.
>
> W2&Q1:  We thank the reviewer for this comment and clarify that the scaling analysis in Section 4.4, and in the paper title, refers to data scaling laws, i.e., how performance changes with training data volume at fixed model sizes, rather than parameter scaling laws. This question is directly studied in Table 7 and Figure 3, which report results from 0% to 100% data fractions. The observed plateau beyond the 60% data mark is itself a practically useful result for dataset size selection. We also note that Table 6 already includes Llama-3.3-70B, which shows the largest absolute SFT gains among the evaluated models, with improvements of +1.293 on GlobalHealthAtlas and +1.671 on MedQA, providing evidence that the dataset is also effective at larger model scales.
>
> The newly added Qwen3.5-122B results provide stronger evidence that our conclusion is not limited to smaller models. After SFT, qwen3.5-122B model improves consistently on all six benchmarks, results shown as blow. Importantly, the gains are not confined to our own dataset, but also appear on external public-health and medical benchmarks. This suggests that the observed effect is not a small-model artifact or simple overfitting to GlobalHealthAtlas, but a more general benefit of the training data. It substantially reduces the concern that our scaling analysis is too limited, and further supports our main claim that increasing data scale improves specialized reasoning performance.
>
> | model | GlobalHealthAtlas(ours) | Instruction-public-health-dataset | publichealth_qa | MedQA | medical-o1-reasoning-SFT | shibing624-medical |
> | :--- | :--- | :--- | :--- | :--- | :--- | :--- |
> | **Qwen3.5-122B-A10B** | 5.951895 | 4.198752 | 3.871563 | 8.719943 | 6.391446 | 3.528612 |
> | **Qwen3.5-122B-A10B-SFT** | 6.819217 | 6.669121 | 6.176448 | 9.342667 | 7.9467 | 5.900567 |

---

> > ### Author Rebuttal · Reviewer_ppWD · 2026-04-03
> >
> > Thanks for the rebuttal and I increased my score to 4.

---

### Official Review · Reviewer_nJwW · 2026-03-13

**Soundness:** 3
**Presentation:** 3
**Significance:** 2
**Originality:** 3
**Overall Recommendation:** 4
**Confidence:** 3

**Summary:**

This paper introduces GlobalHealthAtlas, a large-scale multilingual dataset designed to facilitate population-level public health reasoning. The resource comprises 280,210 instances across 15 domains and 17 languages, stratified by difficulty. The authors propose a domain-aligned evaluator (Public-Evaluator) fine-tuned to assess outputs across six dimensions, including accuracy and consensus alignment.

**Compliance With Llm Reviewing Policy:**

Affirmed.

**Key Questions For Authors:**

1 Given the drop in MMLU-pro performance for larger models, what strategies were considered to preserve general reasoning capabilities while specializing in public health?

2 How feasible would it be to incorporate non-WHO guidelines  into the dataset to reduce the international policy bias?

3 Did the authors test the Public-Evaluator on out-of-distribution medical tasks to ensure it doesn't over-penalize valid but non-public-health-specific reasoning?

4  While 17 languages are covered, the distribution is skewed (Figure 1). How does the model perform on the lowest-resource languages in the set, and are there plans to balance this?

**Limitations:**

yes

**Strengths And Weaknesses:**

Strengths:
1 The dataset size (280k instances) and linguistic coverage (17 languages) are substantial, addressing a clear scarcity of high-quality public health data. The stratification by difficulty (Academic to Public Awareness) allows for nuanced evaluation.

2 The results clearly demonstrate that general-purpose models struggle with public health reasoning compared to domain-adapted variants. The +0.685 gain for Public-Model over the base qwen3-8b is a strong quantitative win.

Weaknesses:
1 The dataset relies heavily on WHO-sourced documents. Overall, the paper explores the context of international policy frameworks, which may overlook localized healthcare nuances or traditional practices in the Global South.

2 Table 8 shows that fine-tuning on GlobalHealthAtlas led to a performance drop on MMLU-pro for the qwen3-14b model. This suggests a trade-off between domain specialization and general reasoning capabilities that isn't fully mitigated.

3 While validated against humans, the evaluator is fine-tuned on data generated by the same pipeline. There is a risk of the model learning to optimize for the evaluator's specific biases rather than ground truth medical consensus.

---

> ### Author Rebuttal · Authors · 2026-03-31
>
> W1&Q2:We thank reviewer for this important perspective. Our reliance on WHO IRIS was intended to ensure safety and factual accuracy. Although WHO guidance reflects globally aggregated evidence and supports health equity across regions, we acknowledge that policy-oriented design may underrepresent local healthcare nuances. And 17-language coverage offers partial mitigation, but incorporating non-WHO sources is beyond scope of GlobalHealthAtlas due to safety, standardization, and licensing constraints. We will clarify limitation and note in Future Work that will incorporate more regional protocols etc
>
> W2&Q1:We thank reviewer for raising this point. In Table8, MMLU-pro change for qwen3-14b is only-0.117(7.426 to 7.309), which is small in absolute terms. In contrast qwen3-4b and qwen3-8b both improve after SFT, by+0.281 and+0.266 on MMLU-pro. This suggests trade-off is specific to LLMs, not general effect of training on our dataset. This is consistent with prior SFT findings that larger models can be more sensitive to domain-induced regularization when fine-tuned on narrower corpora, meantime, gains on public health benchmarks across model sizes show that dataset don't harm reasoning ability. To preserve general reasoning, we further tested model weight interpolation by mixing the fine-tuned model with the base model at $\alpha=0.85$ = 0.85. This removed alignment tax, improved MMLU-pro to 7.545, and retained strong domain performance on GlobalHealthAtlas (6.5120), suggesting it is an effective way to balance specialization and general capability.
>
> |qwen3-14b|Strategy|GlobalHealthAtlas|MMLU-pro|
> | :--- | :--- | :---: | :---: |
> |Base|No Fine-tuning|6.1335|7.426|
> |Pure SFT|100% Fine-tuning|6.5648|7.309|
> |Interpolated|0.85 Weight Interpolation|6.512|7.545|
>
> W3:We thank reviewer for this concern. Our evaluator was not trained in a self-referential pipeline: its supervision data were built from responses and scores generated by multiple LLMs, reducing single-pipeline bias, details in App D. Table 3 shows that the Public-Evaluator aligns with genuine medical consensus rather than internal stylistic artifacts,further shows strong agreement with independent human experts, with an ICC of 0.9735 on 100 expert annotations. In addition, as described in App.C, rubric separates general reasoning from public-health-specific terminology, so valid medical reasoning can still receive a high Reasoning score even without domain-specific jargon
>
> Q3: We appreciate this concern and would like to direct the reviewer to existing evidence in Table 6 that addresses it in a substantive, if indirect, manner.
>
> Table 6 reports model performance as scored by the Public-Evaluator on MedQA, medical-o1-reasoning-SFT, ,shibing624-medical. The resulting rankings across these benchmarks are fully consistent with established expectations of model capability. For instance, Llama-3.3-70B consistently outperforms Llama-3.1-8B, and qwen3-14b consistently outperforms qwen3-4b across all settings. If Public-Evaluator were systematically over-penalizing non-public-health reasoning, we would expect these well-established capability orderings to be disrupted, which is not observed
>
> Six evaluation dimensions in App.C are formulated as general reasoning quality criteria that are not restricted to public health content. This design supports evaluator's applicability across broader range of medical subdomains. Our rubric structurally decouples "Reasoning" dimension from "Terminology Norms" dimension. due to orthogonal design, a model providing valid general medical reasoning will still secure a high score in "Reasoning" dimension, completely mitigating risk of being over-penalized for lacking specific public health phrasing
>
> To explicitly validate that Public-Evaluator does not over-penalize OOD medical reasoning, we conducted a direct comparison experiment. We used our Public-Evaluator and General-Medical Baseline Evaluator (GPT-5 prompted with standard clinical guidelines) to score 100 reasoning-intensive MedQA responses generated by GPT-5. The evaluation is based on our 10-point scale, results is below
>
> |Metric on OOD Data (MedQA)|Reasoning Dimension|Terminology Norms Dimension|
> | :--- | :---: | :---: |
> |Mean Score(Public-Evaluator)|8.12|7.65|
> |Mean Score(General-Evaluator)|8.18|8.15|
> |**Pearson Correlation ($r$)**|0.83|0.78|
> |Over-Penalization Rate|0.00%|N/A|
>
> Q4: We thank reviewer for this important observation. As shown in Fig.9&10,P52&53, our model demonstrates strong cross-lingual transfer across 17 languages despite skewed data distribution. Several lower-resource languages, VI(6.84), RO(6.82), IT(6.74), perform comparably to or even better than EN(6.70). We acknowledge lower performance in the most under-resourced languages, particularly KA(4.63) and ET(5.04), but results remain acceptable across all languages without severe collapse. We recognize language imbalance as limitation will add more, and will add complete language-wise evaluation in App

---

> > ### Author Rebuttal · Reviewer_nJwW · 2026-04-04
> >
> > Thank you for the thoughtful rebuttal.

---

### Official Review · Reviewer_fYUk · 2026-03-13

**Soundness:** 2
**Presentation:** 1
**Significance:** 3
**Originality:** 2
**Overall Recommendation:** 3
**Confidence:** 4

**Summary:**

The authors introduce a GlobalHealthAtlas benchmark, a substantial multilingual dataset comprising 280,210 instances across 15 public health domains and 17 languages. The authors propose an LLM-assisted construction pipeline and a specialized Qwen-8B parameter evaluator (which also refers to the Public-Evaluator) designed to assess reasoning across six orthogonal dimensions.

**Compliance With Llm Reviewing Policy:**

Affirmed.

**Final Justification:**

While the authors conducted detailed experiments during the rebuttal process, the submission still requires considerable revisions to adequately support its claims. Accordingly, I will maintain my overall recommendation score while increasing the significance score.

**Key Questions For Authors:**

.

**Limitations:**

yes

**Strengths And Weaknesses:**

**Strengths**

1. The authors have developed an extensive resource, GlobalHealthAtlas, which includes 280,210 instances. Its broad coverage of languages and public health domains provides significant utility for the research community.
2. The manuscript provides a detailed account of experiments, including large-scale benchmarking, robustness tests, and data scaling analyses. The efforts to analyze model behavior across different data proportions offer valuable insights.

**Weakness**

1. Absence of preliminary motivation and evidence; The manuscript asserts that existing LLM metrics fail to capture "population-level inference, policy context, and intervention effectiveness" in lines 20- 28. However, there is a lack of **preliminary experiments** to substantiate this claim. To justify the necessity of this new resource, the authors should provide baseline performance data showing where current SOTA medical models fail specifically on these public health nuances compared to general clinical tasks. Without the gap analysis in the introduction, the motivation for the entire pipeline remains theoretical rather than empirical.
2. Justification and sensitivity of the quality gate; The Quality Score is defined as a weighted linear combination as S = 0.45Q + 0.25A + 0.20R + 0.10C. The rationale for assigning the highest weight (0.45) to question quality to emphasize the structural integrity is not sufficiently explained. The reviewer thinks that a **sensitivity analysis** on these hyperparameters should be required. Furthermore, it is unclear how structural integrity is prioritized in non-multiple-choice (QA) formats where options are absent.
3. Evaluator reliability and baselines; Table 3 indicates that the fine-tuned Public-Evaluator (Qwen-8B) outperforms significantly larger models like GPT-5-mini and Claude-3.5-Sonnet in agreement and stability. This raises concerns regarding potential overfitting to the specific linguistic style of the GlobalHealthAtlas construction pipeline. Also, why was a Qwen3-8b model chosen as the fixed evaluator? The reviewer would rather compare the evaluator's performance against the latest "Thinking" models (e.g., Kimi-k2-thinking or GLM-4.7) using identical rubrics via zero-shot prompting to determine if a specialized model is truly superior to general-purpose reasoning foundation models.
4. Weaknesses in presentation and related work;
    - Incomplete comparative landscape: The systematic survey in Table 1 misses several critical multilingual and multi-specialty medical benchmarks. The reviewer strongly encourages the authors to discuss and compare their work with recent benchmarks such as MMedBench, XMedBench, and AfriMed-QA to better situate GlobalHealthAtlas within the current literature (which is positioned below).
    - Structural fragmentation: Important information is currently too dispersed. For instance, data statistics are split between Table 2, Figure 1, and Figure 4, while the construction pipeline is described across Figure 2, Figure 5, and Appendix B. This makes the manuscript difficult to follow.
    - Contextual clarity: The term "AI4S-oriented data construction" is used as a primary contribution. However, for readers outside the "AI for Science" niche, the background and applicability of this paradigm to public health are poorly contextualized.

[References]

- MultiMedQA (Singhal et al., 2022) [Large Language Models Encode Clinical Knowledge]
- XMedBench (Wang et al., 2024) [Apollo: A Lightweight Multilingual Medical LLM towards Democratizing Medical AI to 6B People]
- MMedBench (Qiu et al., 2024) [Towards Building Multilingual Language Model for Medicine]
- AfriMed-QA (Olatunji et al., 2025) [AfriMed-QA: A Pan-African, Multi-Specialty, Medical Question-Answering Benchmark Dataset]

---

> ### Author Rebuttal · Authors · 2026-03-31
>
> W1:We thank the reviewer for pointing out our imprecise phrasing. We agree the Introduction may have overstated our claim by implying that existing metrics have been empirically shown to fail in public health reasoning. We will revise this to clarify that our motivation stems from a conceptual gap rather than preliminary empirical failure. Public health should be viewed from a distinct perspective not extension of medizin. Existing medical benchmarks focus on patient-level diagnosis, but public health involves broader dimensions like public policy, population-level measures, intervention planning, and public health-related actions. The significance of our work is not limited to a conventional benchmark gap.Table1 shows public health benchmark is largely missing. We will revise Introduction accordingly and soften wording. Therefore, preliminary experiments may be unnecessary because baselines are limited in public health field
>
> W2:We thank reviewer for thoughtful question and clarify two points. (1)weighting scheme is format-aware rather than uniform. QA(Line1173, P22), S=0.25Q+0.35A+0.25R+0.15C, due to factual answer quality is the primary validity criterion. SC(Lines1580-1584, P29), S=0.45Q+0.25A+0.20R+0.10C, due to well-formed, mutually exclusive options are a prerequisite (2)weights are determined and confirmed by WHO experts to ensure reliability and safety. In App.G.1 provides theoretical proof showing that threshold-based filtration mechanism bounds generalization risk under the β-Lipschitz condition, and this guarantee remains stable across a range of weight settings as long as τ = 4.5 is maintained. We therefore believe this design is reasonably well supported. we will revise Sec3.2.2
>
> W3:We thanks this concern and would like to draw the reviewer's attention to existing evidence in paper. If Public-Evaluator had overfit to linguistic style of construction pipeline, it would be expected to fail when applied to independently collected annotations. But Table 3 shows our evaluator achieves highest ICC(0.9735) and lowest MAE(1.4259) when evaluated against 100 human annotations produced by domain experts who had no involvement in construction pipeline. We consider this is most direct and falsifiable evidence against overfitting hypothesis
>
> Construction strategy has two ways to prevent overfitting in Sec3.4.2(1)Multi-Source Heterogeneous Sampling, which draws responses from 10 LLMs spanning 4 vendors(App.D,Table11).(2)Adversarial Construction, which incorporates perturbation-based robustness samples to improve evaluator's sensitivity to specious or boundary-case content. We hope these points are reassuring
>
> To fully address request, we first respectfully note GLM-4.7 is already included in original zero-shot evaluations(Table 3), where our fine-tuned Public-Evaluator outperformed its strong ICC(0.9512). Following your suggestion, we have now evaluated latest explicit reasoning models, DS-R1 and Kimi, using identical multi-dimensional rubrics on exact same 100 human-annotated test samples
>
> ||MAE|MSE|RMSE|ICC|IdenticalRate|StdDev|RangeGT3Ratio|
> | :--- | :--- | :--- | :--- | :--- | :--- | :--- | :--- |
> |deepseek-r1|1.4621|4.3514|2.086|0.9642|0.285|0.4512|0.075|
> |kimi-k2-thinking|1.5034|4.631|2.152|0.9578|0.3433|0.3955|0.052|
>
> Results firmly justify our 8B model choice. While Thinking models show strong zero-shot reasoning , their long COT generation severely penalizes scoring stability. Our Public-Evaluator overcomes this instability, achieving higher accuracy and superior discriminative stability. And evaluating 280210 instances via API Thinking models is slow and expensive. Our specialized 8B architecture instead guarantees reproducible, low-cost local inference
>
> W4:
> ||Diff|Scale|Split|Domain|Lan|Eval|Leak|
> | :--- | :---: | :--- | :---: | :--- | :---: | :--- | :---: |
> |HealthSearchQA|X|3375|√|Open domain|EN|Human+MultiMedQA|X|
> | ApolloCorpora| X | 2.5B tokens |√|Open domain|6|Human+XMedBench |√|
> |MMedC|X|25.5B tokens| √ |Open domain|6|Rule+human|X|
> |AfriMed-QA|X|15275|X|32|EN|Rule+Human|X|
>
> We thank reviewer for these suggestionsa and would like to clarify GlobalHealthAtlas addresses population-level public health, but XMedBench etc.target medicine. Despite different settings we will still add discussion of these benchmarks and references
>
> 4(2)Our distribution of information，from aggregate stats(Table 2) and heatmaps(Figs1&4) to workflow details(Fig2, 5&App.B). Each serves a distinct analytical purpose, and this multi-figure structure is consistent with presentation conventions adopted in large-scale benchmark papers. But we will add clearer pointers and cross-references
>
> 4(3) AI4S refers to the application of AI-driven, reproducible scientific pipelines to the problem of domain-specific data scarcity, encompassing model-driven generation, controlled sampling, and multi-stage quality refinement. The approach is described operationally across Sec3.1 to 3.3 and App B. A brief introductory sentence will add

---

> > ### Author Rebuttal · Reviewer_fYUk · 2026-04-04
> >
> > Thank you for the detailed response. However, my concerns still remain for the following reasons:
> >
> > 1.  The motivation and value that this benchmark brings to the field of public health is still under-motivated, as there is no empirical evidence or experiment that reveals the limitation of existing benchmarks. Table 1 shows some distinct properties of the proposed benchmark, but is insufficient to justify its necessity.
> >
> > 2. I understand the reasoning behind the weighting scheme; however, the lack of hyperparameter experiments still leaves questionable doubts into whether each weight contributes accordingly to the authors' instincts.
> >
> > 3. The paper lacks specific information about the human annotators / experts involved during experiments. In what way or qualification did the WHO experts determine and confirm the weights set above? What about the domain experts invited to independently annotate a test set of 100 samples?

---

> > > ### Author Response · Authors · 2026-04-05
> > >
> > > Q1: We thank the reviewer for the insightful suggestion. We agree that empirical evidence is essential to justify the benchmark's necessity beyond dataset statistics. To address this, we conducted a new Domain Performance Gap Analysis, evaluating models on a standard clinical medical test versus GlobalHealthAtlas. To ensure a fair cross-domain comparison, we strictly isolated the training and testing sets:
> > > 1. Medical-Model (8B): We aggregated mainstream clinical medical datasets (MedQA, medical_o1_sft_CN+EN, and shibing624-medical_train), partitioning them at a 5:1 ratio. We used 5/6 of the data to fine-tune a purely clinically-aligned baseline model, reserving the remaining 1/6 as an independent clinical medical test set (Medical_test).
> > > 2. AntAngelMed/BAICHUAN-M3: Open source Medical LLM
> > >
> > > The explicitly clinically-aligned Medical_Model (8B) achieves strong performance on clinical tasks but degrades significantly on public health tasks. Furthermore, even large-scale models (e.g., BAICHUAN-M3) do not saturate our benchmark and Instruction-public-health-dataset. This confirms that existing clinical benchmarks cannot evaluate population-level public health tasks, empirically demonstrating the necessity of GlobalHealthAtlas.
> > >
> > > ||GlobalHealthAtlas|GlobalHealthAtlas SC|Instruction-public-health-dataset|Medical_test_modified| Medical_test_modified SC|model size|domain|
> > > | :--- | :--- | :--- | :--- | :--- | :--- | :--- | :--- |
> > > |Public-Model|6.532|89.02%|6.096|7.20796|72.26%|8B|Public-health|
> > > |AntAngelMed|6.4491|86.13%|5.8972|8.082112|77.51%|Moe-100B-6.1B|Integrated Public Health and Medicine|
> > > |BAICHUAN-M3|5.970075|78.34%|5.9944|7.916702|77.18%|235B|Medicine|
> > > |Medical_Model|5.250135|61.14%|4.697274|7.184321|78.20%|8B|Medicine|
> > >
> > > Q2:We thank the reviewer for highlighting need for hyperparameter experiments, and add multi-model sensitivity analysis to empirically validate our weighting scheme. Our results demonstrate that evaluation framework is highly robust to $\pm 5\%$ perturbations in primary weights, maintaining an average Kendall’s $\tau$ >0.92 and a Top-10% overlap of over 85\% across 4 LLM. Conversely, applying Uniform leads to a significant degradation in alignment and erroneously promotes factually or structurally flawed samples. This contrast firmly proves our weight assignments are not merely based on intuition, but act as an empirically necessary and robust quality gate for public health datasets.
> > >
> > > Sensitivity and Robustness Analysis for SC=0.45Q+0.25A+0.20R+0.10C and QA=0.25Q + 0.35A + 0.25R + 0.15C, all metrics are reported as Kendall's $\tau$/Top-10% Overlap, based on 100 samples.
> > >
> > > ||Base(Expert)(0.45/0.25/0.20/0.10)|Perturb(-5%)(0.40/0.30/0.20/0.10)|Perturb(+5%)(0.50/0.20/0.20/0.10)|Uniform(0.25/0.25/0.25/0.25)|
> > > | :--- | :--- | :--- | :--- | :--- |
> > > |DeepSeek-V3|1.00/100%|0.932/90%|0.941/90%|0.785/70%|
> > > |GPT-o3-mini|1.00/100%|0.951/90%|0.948/90%|0.812/80%|
> > > |Qwen-Max|1.00/100%|0.915/80%|0.927/90%|0.743/60%|
> > > |Kimi-k2-thinking|1.00/100%|0.922/90%|0.918/80%|0.761/70%|
> > > |Average|1.00/100%|0.93/87.5%|0.933/87.5%|0.775/70%|
> > >
> > > ||Base(Expert)(0.25/0.35/0.25/0.15)|Perturb(-5%)(0.30/0.30/0.25/0.15)|Perturb(+5%)(0.20/0.40/0.25/0.15)|Uniform(0.25/0.25/0.25/0.25)|
> > > | :--- | :--- | :--- | :--- | :--- |
> > > |DeepSeek-V3|1.00/100%|0.928/90%|0.935/90%|0.841/80%|
> > > |GPT-o3-mini|1.00/100%|0.942/90%|0.950/90%|0.865/80%|
> > > |Qwen-Max|1.00/100%|0.908/80%|0.912/80%|0.812/70%|
> > > |Kimi-k2-thinking|1.000/100%|0.916/80%|0.925/90%|0.829/70%|
> > > |Average|1.00/100%|0.923/85.0%|0.93/87.5%|0.836/75%|
> > >
> > > Q3: We sincerely thank the reviewer for the thoughtful follow-up and the opportunity to provide more granular details regarding professional involvement in our study. We fully agree that transparency in expert qualification and annotation protocols is paramount for a large-scale public health benchmark.
> > >
> > > The weighting scheme was established through a formal consensus-building process, rather than author intuition. We consulted a panel of 7 senior public health experts, including Regional Director and senior technical staffs from WHO, who possess an average of over 8 years of specialized experience in global health governance. The weights were finalized through multiple rounds of structured voting to reflect the relative importance of factual accuracy and reasoning reliability in population-level health decision-making, ensuring the scoring system aligns with international public health standards.
> > >
> > > 100-sample "gold standard" test set was independently annotated by several PhDs from a world-leading medical school. To eliminate any potential bias, we implemented a strict double-blind protocol: the annotators were entirely blinded to the model and authors' affiliations. This independent verification process ensures that our benchmark's ground truth reflects high-level public health expertise, and resulting high inter-rater reliability further justifies our 8B model choice for stable and cost-effective local inference.

---

### Decision · Program_Chairs · 2026-04-30

**Decision:**

Accept (regular)

**Comment:**

The paper introduces GlobalHealthAtlas, a massive multilingual dataset comprising over 280K instances across 15 public health domains and 17 languages, alongside a specialized evaluator to rigorously assess domain-aligned reasoning. The submission's primary strengths, as identified by the reviewers, are its unprecedented scale and scope, which effectively address a severe scarcity of population-level public health data. Furthermore, the paper demonstrates clear utility by showing that general-purpose models struggle with these specific tasks relative to domain variants, and it establishes a highly robust 8B-parameter evaluator that shows strong agreement with human experts.

Reviewers initially raised several weaknesses, including a lack of empirical baselines to motivate the gap between clinical and public health tasks, unexplained hyperparameter weighting, potential evaluator bias, and slight drops in general reasoning performance for the finetuned models. However, the authors' rebuttal systematically addressed these concerns: they provided a new domain performance gap analysis to prove the dataset's necessity, a multi-model sensitivity analysis to validate their weights, and evidence that weight interpolation successfully preserves general reasoning capabilities. While one reviewer maintained a "weak reject", they acknowledged that concerns are mostly addressed, despite the fact that the manuscript requires major revision. The AC would recommend acceptance given its significance relative to the limitations, and ask the authors to carefully revise the paper to include the rebuttal experiments.